# Harnessing Multiple Correlated Networks for Exact Community Recovery

**Miklós Z. Rácz**
Northwestern University
Evanston, IL 60208
miklos.racz@northwestern.edu

**Jifan Zhang**
Northwestern University
Evanston, IL 60208
jifanzhang2026@u.northwestern.edu

## Abstract

We study the problem of learning latent community structure from multiple correlated networks, focusing on edge-correlated stochastic block models with two balanced communities. Recent work of Gaudio, Rácz, and Sridhar (COLT 2022) determined the precise information-theoretic threshold for exact community recovery using two correlated graphs; in particular, this showcased the subtle interplay between community recovery and graph matching. Here we study the natural setting of more than two graphs. The main challenge lies in understanding how to aggregate information across several graphs when none of the pairwise latent vertex correspondences can be exactly recovered. Our main result derives the precise information-theoretic threshold for exact community recovery using any constant number of correlated graphs, answering a question of Gaudio, Rácz, and Sridhar (COLT 2022). In particular, for every $K \geq 3$ we uncover and characterize a region of the parameter space where exact community recovery is possible using $K$ correlated graphs, even though (1) this is information-theoretically impossible using any $K - 1$ of them and (2) none of the latent matchings can be exactly recovered.

## 1 Introduction

Finding communities in networks—that is, groups of nodes that are similar—is one of the fundamental problems in machine learning. This task is crucially important for understanding the underlying structure and function of networks across diverse applications, including sociology and biology [23]. The increasing availability of network data sets offers the intriguing possibility of improving community recovery algorithms by synthesizing information across correlated networks. However, in many settings the graphs are not aligned—which may happen for a variety of reasons, including anonymization, missing or erroneous data, or simply the alignment being unknown—which presents a challenge. Thus graph matching—the task of recovering the latent vertex alignment between graphs—plays a central role in efforts to integrate data across networks. Our work follows an exciting recent line of work at the intersection of community recovery and graph matching.

Recently, Rácz and Sridhar [41] initiated the study of community recovery in correlated stochastic block models (SBMs), focusing on the simplest setting of two correlated graphs with two balanced communities. They determined the information-theoretic limits for exact graph matching, which has applications for community recovery. In particular, they uncovered a region of the parameter space where exact community recovery is possible using two correlated graphs even though it is information-theoretically impossible to do so using just a single graph. Subsequently, Gaudio, Rácz, and Sridhar [22] determined the information-theoretic limits for exact community recovery from two correlated SBMs. This required going beyond exact graph matching and understanding the subtle interplay between community recovery and graph matching.

38th Conference on Neural Information Processing Systems (NeurIPS 2024).

Gaudio, Rácz, and Sridhar [22] posed the question of understanding what happens in the case of more than two graphs, which arises naturally in all the motivating examples. For instance, people participate in numerous overlapping yet complementary social networks, and only by combining these can we fully understand and make inferences about society. Similarly, synthesizing information across protein-protein interaction networks from several related species can aid in inferring protein functions [44]. The main challenge lies in understanding how to optimally pass information across three or more graphs.

Our main contribution fully answers this open question by Gaudio, Rácz, and Sridhar [22]. Specifically, we precisely characterize the information-theoretic threshold for exact community recovery given $K$ correlated SBMs, for any constant $K$. This result highlights an intricate phase diagram and quantifies the value of each additional correlated graph for the task of community recovery. In particular, for every $K \geq 3$ we uncover and characterize a region of the parameter space where exact community recovery is possible using $K$ correlated graphs, even though (1) this is impossible using any $K-1$ of them and (2) none of the latent matchings can be exactly recovered. See Section 3 and Theorems 1 and 2 for details.

Along the way, we also precisely characterize the information-theoretic threshold for exact graph matching given $K$ correlated SBMs, for any constant $K$. In particular, we uncover and characterize a region of the parameter space where the latent matching between two correlated SBMs cannot be exactly recovered given just the two graphs, but it can be exactly recovered given $K > 3$ correlated SBMs. See Section 3 and Theorems 3 and 4 for details.

To prove our results, we study the so-called $k$-core matching between all pairs of graphs. Recent works have shown the $k$-core matching to be a flexible and successful tool in a variety of settings for two correlated graphs [13, 22, 43]. Our main technical contribution is to extend this analysis to more than two graphs. The main difficulty lies in understanding the size of *intersections* of "bad sets" for $k$-core matchings for different pairs of graphs. We refer to Section 4 for details.

## 2 Models and questions

**The stochastic block model (SBM).** The SBM is the most common probabilistic generative model for networks with latent community structure. First introduced by Holland, Laskey, and Leinhardt [25], it has garnered considerable attention and research. In particular, it can be employed as a natural testbed for evaluating and assessing clustering algorithms on average-case networks [1]. The SBM notably displays sharp information-theoretic phase transitions for various inference tasks, offering a detailed understanding of when community information can be extracted from network data. The phase transition thresholds were conjectured by Decelle et al. [15] and were proved rigorously in several papers [33, 34, 35, 36, 2, 3]. We refer to the survey [1] for a detailed overview of the SBM.

In this paper, we focus on the simplest setting, a SBM with two symmetric communities. Let $n$ be a positive integer and let $p, q \in [0, 1]$ be parameters representing probabilities. We construct a graph $G \sim \mathrm{SBM}(n, p, q)$ as follows. The graph $G$ has $n$ vertices, labeled by $[n] := \{1, 2, 3, \ldots, n\}$. Each vertex $i$ is assigned a community label $\sigma^*(i)$ from the set $\{+1, -1\}$; these are drawn i.i.d. uniformly at random across $i \in [n]$. Let $\boldsymbol{\sigma}^* := \{\sigma^*(i)\}_{i=1}^n$ denote the community label vector. The vertices are thus categorized into two communities: $V^+ := \{i \in [n] : \sigma^*(i) = +1\}$ and $V^- := \{i \in [n] : \sigma^*(i) = -1\}$. Given the community labels $\boldsymbol{\sigma}^*$, the edges of $G$ are drawn independently between pairs of distinct vertices. If $\sigma^*(i) = \sigma^*(j)$, then the edge $(i, j)$ is in $G$ with probability $p$; otherwise, it is in $G$ with probability $q$.

**Community recovery.** In the community recovery task, an algorithm takes as input the graph $G$ (without knowing $\boldsymbol{\sigma}^*$) and outputs an estimated community labeling $\widehat{\boldsymbol{\sigma}}$. Define the *overlap* between the estimated labeling and the ground truth as follows:

$$\mathrm{ov}(\boldsymbol{\sigma}^*, \widehat{\boldsymbol{\sigma}}) := \frac{1}{n} \left| \sum_{i=1}^n \sigma^*(i) \widehat{\sigma}(i) \right|.$$

The overlap measures how well the true community labels and the estimated labels of the algorithm match. Note that $\mathrm{ov}(\boldsymbol{\sigma}^*, \widehat{\boldsymbol{\sigma}}) \in [0, 1]$, where the larger the value is, the better performance the algorithm has. In particular, the algorithm succeeds in exactly recovering the partition into two communities (i.e., $\boldsymbol{\sigma}^* = \widehat{\boldsymbol{\sigma}}$ or $\boldsymbol{\sigma}^* = -\widehat{\boldsymbol{\sigma}}$) if and only if $\mathrm{ov}(\boldsymbol{\sigma}^*, \widehat{\boldsymbol{\sigma}}) = 1$. Our focus in this paper is achieving this goal, known as exact community recovery.

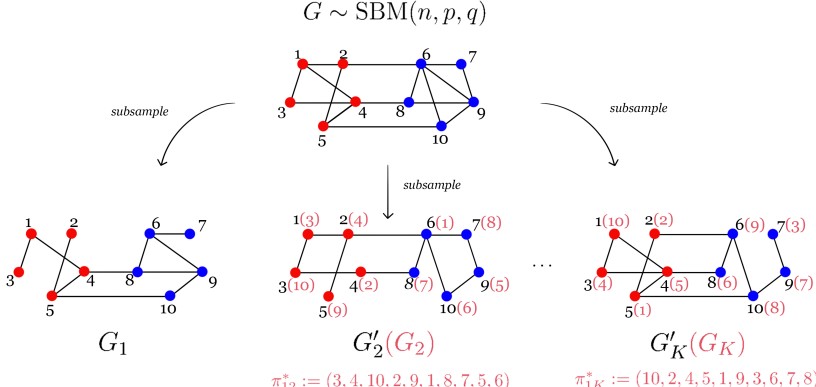

Figure 1: Schematic showing the construction of multiple correlated SBMs (see text for details).

It is well-known that exact community recovery is most challenging and interesting in the logarithmic average degree regime [1]. Accordingly, we focus on this regime: in most of the paper we assume that $p = a\frac{\log n}{n}$ and $q = b\frac{\log n}{n}$ for some constants $a, b > 0$. In this regime there is a sharp information-theoretic threshold for exact community recovery [2, 35, 3]. Let $D_+(a, b) := (\sqrt{a} - \sqrt{b})^2/2$ denote the so-called *Chernoff-Hellinger divergence*. Then the information-theoretic threshold is given by

$$D_+(a, b) = 1. \tag{2.1}$$

In other words, if $D_+(a, b) > 1$, then exact recovery is possible (and, in fact, efficiently). That is, there is a (polynomial-time) algorithm which outputs an estimator $\widehat{\sigma}$ with the guarantee that $\lim_{n\to\infty} \mathbb{P}(\mathsf{ov}(\sigma^*, \widehat{\sigma}) = 1) = 1$. On the other hand, if $D_+(a, b) < 1$ then exact recovery is impossible: for any estimator $\widetilde{\sigma}$, we have that $\lim_{n\to\infty} \mathbb{P}(\mathsf{ov}(\sigma^*, \widetilde{\sigma}) = 1) = 0$.

**Correlated SBMs.** The objective of our work is to understand how the sharp threshold for exact community recovery varies when the input data involves multiple correlated graphs. To do so, we first define a natural model of multiple correlated SBMs [29, 39, 28] (and see further discussion in Section 6 about alternative models).

We construct $(G_1, \ldots, G_K) \sim \mathrm{CSBM}(n, p, q, s)$ as follows, where the additional parameter $s \in [0, 1]$ reflects the degree of correlation between the graphs (and the number of graphs $K$ is dropped from the notation for ease of readability). First, generate a parent graph $G_0 \sim \mathrm{SBM}(n, p, q)$ with community labels $\sigma^*$. Subsequently, given $G_0$, construct $G_1', G_2', \ldots, G_K'$ by independent subsampling. Specifically, each edge of $G_0$ is included in $G_i'$ with probability $s$, independently of everything else, and non-edges of $G_0$ remain non-edges in $G_i'$. The graphs $G_i'$ inherit both the vertex labels and the community labels from the parent graph $G_0$. Finally, let $\pi_{12}^*, \ldots, \pi_{1K}^*$ be i.i.d. uniformly random permutations of $[n]$ and let $\pi^* := (\pi_{12}^*, \ldots, \pi_{1K}^*)$. Define $G_1 := G_1'$ and, for all $i \in \{2, \ldots, K\}$, define $G_i := \pi_{1i}^*(G_i')$. In other words, for every $i > 1$ and $j \in [n]$, vertex $j$ in $G_i'$ is relabeled to $\pi_{1i}^*(j)$ in $G_i$. This last relabeling step mirrors the real-world observation that vertex labels are often unaligned across graphs. This construction is shown in Figure 1.

An important property of the model is that marginally each graph $G_i$ is an SBM. Since the subsampling probability is $s$, we have that $G_i \sim \mathrm{SBM}(n, ps, qs)$. Thus, it follows from (2.1) that, in the logarithmic average degree regime where $p = a\frac{\log n}{n}$ and $q = b\frac{\log n}{n}$, the communities can be exactly recovered from $G_1$ alone precisely when $sD_+(a, b) = D_+(sa, sb) > 1$.

The key question in our work is how to improve the threshold by incorporating more information as $K$, the number of correlated SBMs, increases. This question was initiated by Rácz and Sridhar [41] and then solved by Gaudio, Rácz, and Sridhar [22] when $K = 2$.

An essential observation is that, to go beyond the threshold, one needs to combine information from the $K$ graphs $G_1, \ldots, G_K$ through graph matching. Then one can exactly recover the community labels using the combined information, even in regimes where it is information-theoretically impossible to exactly recover $\sigma^*$ given up to $K - 1$ graphs.

To be more specific, if $\pi^*$ were known, then one can reconstruct $G_j'$ from $G_j$ and then combine the graphs $G_1', \ldots, G_K'$ to obtain the union graph $H^*$, defined as follows: the edge $(i, j)$ is included in

$H^*$ if and only if $(i,j)$ is included in at least one of $G'_1, \ldots, G'_K$. Note that $H^*$ is also an SBM; specifically, $H^* \sim \text{SBM}\left(n, \left(1 - (1-s)^K\right)p, \left(1 - (1-s)^K\right)q\right)$, so (2.1) directly implies that the communities can be exactly recovered from the union graph $H^*$ if $\left(1 - (1-s)^K\right)\text{D}_+(a,b) > 1$.

**Graph matching.** In real applications, the permutations $\boldsymbol{\pi}^* = (\pi_{12}^*, \ldots, \pi_{1K}^*)$ are often not known. The arguments above highlight the importance of an intermediate task, known as graph matching: how can one recover the latent permutations $\boldsymbol{\pi}^*$ given the graphs $(G_1, G_2, \ldots, G_K)$? While here we regard graph matching as an important intermediate step, it is of great significance in its own right, with applications in social network privacy [40], machine learning [10], and more. For two correlated SBMs, this problem was resolved by Rácz and Sridhar [41], who proved that the information-theoretic threshold for (pairwise) exact graph matching is $s^2(a+b)/2 = 1$. Note that this is also the connectivity threshold for the intersection graph of $G_1$ and $G'_2$ (see [41]). Denote

$$\text{T}_c(a,b) := \frac{a+b}{2}. \tag{2.2}$$

With this notation, the (pairwise) exact graph matching threshold is given by $s^2\text{T}_c(a,b) = 1$. This directly implies (by a union bound) that if $s^2\text{T}_c(a,b) > 1$, then $\boldsymbol{\pi}^*$ can be exactly recovered given $(G_1, G_2, \ldots, G_K) \sim \text{CSBM}(n, a\frac{\log n}{n}, b\frac{\log n}{n}, s)$, for any constant $K$. By the discussion above, this also gives a sufficient condition for exact community recovery given $(G_1, G_2, \ldots, G_K) \sim \text{CSBM}(n, a\frac{\log n}{n}, b\frac{\log n}{n}, s)$:

$$s^2\text{T}_c(a,b) > 1 \qquad \text{and} \qquad \left(1 - (1-s)^K\right)\text{D}_+(a,b) > 1. \tag{2.3}$$

We will generalize the exact graph matching result of Rácz and Sridhar [41] and show (see Theorem 3 below) that $\boldsymbol{\pi}^*$ can be exactly recovered given $(G_1, G_2, \ldots, G_K) \sim \text{CSBM}(n, a\frac{\log n}{n}, b\frac{\log n}{n}, s)$ if

$$s\left(1 - (1-s)^{K-1}\right)\text{T}_c(a,b) > 1, \tag{2.4}$$

which (for $K > 2$) is weaker than the condition $s^2\text{T}_c(a,b) > 1$ implied by [41]. (Moreover, in Theorem 4 we show that the condition in (2.4) is tight for exact recovery of $\boldsymbol{\pi}^*$.) Thus, by the discussion above, this gives a sufficient condition for exact community recovery given $(G_1, G_2, \ldots, G_K) \sim \text{CSBM}(n, a\frac{\log n}{n}, b\frac{\log n}{n}, s)$:

$$s\left(1 - (1-s)^{K-1}\right)\text{T}_c(a,b) > 1 \qquad \text{and} \qquad \left(1 - (1-s)^K\right)\text{D}_+(a,b) > 1. \tag{2.5}$$

**The interplay between community recovery and graph matching.** The condition (2.5) is, however, not tight. To attain the sharp threshold for exact community recovery given $K$ graphs, we need to answer the following question: does there exist a parameter regime where exact community recovery is possible for $K$ graphs, even though (1) exact graph matching is impossible, and (2) exact community recovery is impossible using only $K-1$ graphs?

For $K = 2$ graphs, Gaudio, Rácz, and Sridhar [22] proved that the sharp threshold for exact community recovery given $K$ correlated SBMs is given by

$$s^2\text{T}_c(a,b) + s(1-s)\text{D}_+(a,b) > 1 \qquad \text{and} \qquad \left(1 - (1-s)^2\right)\text{D}_+(a,b) > 1. \tag{2.6}$$

The condition $\left(1 - (1-s)^2\right)\text{D}_+(a,b) > 1$ is necessary due to the work [41]. The first condition in (2.6) demonstrates the interplay between community recovery and graph matching. To be more specific, the first term $s^2\text{T}_c(a,b)$ is the threshold for exact graph matching given $(G_1, G_2)$, while the second term $s(1-s)\text{D}_+(a,b)$ comes from community recovery.

Our main contribution generalizes this result, determining the exact community recovery threshold for $K \geq 3$ graphs. If $\left(1 - (1-s)^K\right)\text{D}_+(a,b) > 1$, then the sharp threshold is given by

$$s\left(1 - (1-s)^{K-1}\right)\text{T}_c(a,b) + s(1-s)^{K-1}\text{D}_+(a,b) > 1. \tag{2.7}$$

The condition (2.7) also clearly exhibits the interplay between community recovery and graph matching. The first term comes from graph matching, while the second term comes from community recovery, as in the case of $K = 2$. We refer to Section 3 and Theorems 1 and 2 for details.

Despite the apparent similarity in results, when $K \geq 3$ the situation differs significantly from that of two graphs. The primary challenge lies in the existence of multiple methods for matching $K \geq 3$ graphs. When $K = 2$, there is only a single matching that needs to be recovered from $G_1$ and $G_2$. In contrast, with three or more graphs, the graphs can be matched pairwise, or to some anchor graph, or potentially in many other ways. Integrating information across different matchings requires substantial additional effort. We present the formal results in the next section.

## 3 Results

Our main contributions are to determine the precise information-theoretic thresholds for exact community recovery and for exact graph matching given $K$ correlated SBMs.

### 3.1 Threshold for exact community recovery

We first describe the precise information-theoretic threshold for exact community recovery, starting with the positive direction.

**Theorem 1** (Exact community recovery from $K$ correlated SBMs). *Fix constants $a, b > 0$ and $s \in [0, 1]$, and let $(G_1, G_2, \ldots, G_K) \sim \mathrm{CSBM}(n, a\frac{\log n}{n}, b\frac{\log n}{n}, s)$. Suppose that the following two conditions both hold:*

$$\left(1 - (1-s)^K\right) \mathrm{D}_+(a, b) > 1 \tag{3.1}$$

*and*

$$s\left(1 - (1-s)^{K-1}\right) \mathrm{T}_c(a, b) + s(1-s)^{K-1}\mathrm{D}_+(a, b) > 1. \tag{3.2}$$

*Then exact community recovery is possible. That is, there is an estimator $\widehat{\boldsymbol{\sigma}} = \widehat{\boldsymbol{\sigma}}(G_1, G_2, \ldots, G_K)$ such that $\lim_{n \to \infty} \mathbb{P}\left(\mathrm{ov}\left(\widehat{\boldsymbol{\sigma}}, \boldsymbol{\sigma}^*\right) = 1\right) = 1$.*

Combined with Theorem 2 below (which shows that Theorem 1 is tight), this result precisely answers an open problem of Gaudio, Rácz, and Sridhar [22]. The condition (3.1) is required for exact community recovery for $K$ graphs by [41]. We now focus on the condition (3.2). In the prior work [22], it is proved that the threshold for exact community recovery for two graphs is given by (2.6). The primary contribution of Theorem 1 is to go beyond this threshold as the number of graphs $K$ increases. In particular, this showcases that there exists a regime where (1) it is impossible to exactly recover $\boldsymbol{\sigma}^*$ from $(G_1, G_2, \ldots, G_{K-1})$ alone and (2) any exact graph matching is impossible, yet one can perform exact recovery of $\boldsymbol{\sigma}^*$ given $(G_1, G_2, \ldots, G_K)$. This requires developing novel algorithms that integrate information from $(G_1, G_2, \ldots, G_K)$ delicately and incorporate multiple graph matchings carefully.

Here we first provide a detailed discussion of the algorithms for three graphs $(G_1, G_2, G_3) \sim \mathrm{CSBM}(n, a\frac{\log n}{n}, b\frac{\log n}{n}, s)$, which is the simplest case with intriguing new phenomena and challenges as mentioned. This avoids complicated notations (which arise for general $K$) for easier understanding. The new techniques used for combining multiple matchings and integrating information with three graphs are subsequently generalized to $K > 3$ correlated SBMs. The high level idea of the algorithm for exact community recovery when $K = 3$ consists of five steps (in the following discussion we assume $a > b$; when $a < b$, change majority to minority everywhere):

1. Obtain an almost exact community labeling of $G_1$.
2. Obtain three pairwise partial almost exact graph matchings $\widehat{\mu}_{12}$, $\widehat{\mu}_{13}$, and $\widehat{\mu}_{23}$ between graph pairs $(G_1, G_2)$, $(G_1, G_3)$, and $(G_2, G_3)$, respectively.
3. For vertices in $G_1$ that are part of at least two matchings, refine the almost exact community labeling in Step 1 via majority vote in the (union) graph consisting of edges that appear at least once in $(G_1, G_2, G_3)$.
4. For vertices in $G_1$ that are part of only $\widehat{\mu}_{12}$ (resp. $\widehat{\mu}_{13}$), label them via majority vote of their neighbors' labels in the graph consisting of edges that appear only in $G_1$ and not in $G_2$ (resp. only in $G_1$ and not in $G_3$).
5. For vertices in $G_1$ that are not part of any of the three matchings or only part of $\widehat{\mu}_{23}$, label them via majority vote of their neighbors' labels in $G_1$.

Each step in the algorithm involves abundant technical details. See Section 4 for a detailed overview of the algorithms and proofs. Note that the threshold (3.2) captures the interplay between community recovery and graph matching, which we now discuss in more detail.

The first term in (3.2), which is $s\left(1 - (1-s)^2\right)\mathrm{T}_c(a, b)$ for $K = 3$, comes from graph matching. In [22] it is shown that for one matching, say $\widehat{\mu}_{12}$, the best possible almost exact graph matching makes $n^{1-s^2\mathrm{T}_c(a,b)+o(1)}$ errors. Here, we show that it is possible to obtain almost exact matchings $\widehat{\mu}_{12}$ and $\widehat{\mu}_{13}$ (namely, these will be $k$-core matchings; see Section 4 for details) such that the size

of the *intersection* of the two error sets is $n^{1-s(1-(1-s)^2)\mathrm{T_c}(a,b)+o(1)}$, which is a smaller power of $n$. This quantifies how synthesizing information across graph matchings can reduce errors and this exponent is precisely what shows up in the first term in (3.2). This observation is important and relevant for Steps 4 and 5 in the algorithm.

On the other hand, the second term in (3.2), which is $s(1-s)^2\mathrm{D_+}(a,b)$ for $K=3$, comes from community recovery. In fact, this term arises from the majority votes in Step 5, where we use only edges in $G_1$. Note that the nodes that are unmatched by $\widehat{\mu}_{12}$ and $\widehat{\mu}_{13}$ are, roughly speaking, the isolated nodes in the intersection graphs of $G_1$ and $G_2$, and $G_1$ and $G_3$, respectively. Thus, while we use all edges in $G_1$ in this step, the relevant edges are not present in $G_2$ nor in $G_3$, giving the "effective" factor of $s(1-s)^2$. By (2.1), the exact community recovery threshold for $\mathrm{SBM}(n, s(1-s)^2 a\frac{\log n}{n}, s(1-s)^2 b\frac{\log n}{n})$ is $s(1-s)^2\mathrm{D_+}(a,b) = 1$, giving the second term in (3.2).

The following impossibility result shows the tightness of Theorem 1.

**Theorem 2** (Impossibility of exact community recovery). *Fix constants $a, b > 0$ and $s \in [0,1]$, and let $(G_1, G_2, \ldots, G_K) \sim \mathrm{CSBM}(n, a\frac{\log n}{n}, b\frac{\log n}{n}, s)$. Suppose that either*

$$\left(1 - (1-s)^K\right)\mathrm{D_+}(a,b) < 1 \tag{3.3}$$

*or*

$$s\left(1 - (1-s)^{K-1}\right)\mathrm{T_c}(a,b) + s(1-s)^{K-1}\mathrm{D_+}(a,b) < 1. \tag{3.4}$$

*Then exact community recovery is impossible. That is, for any estimator $\widetilde{\boldsymbol{\sigma}} = \widetilde{\boldsymbol{\sigma}}(G_1, G_2, \ldots, G_K)$, we have that $\lim_{n\to\infty} \mathbb{P}\left(\mathrm{ov}\left(\widetilde{\boldsymbol{\sigma}}, \boldsymbol{\sigma}^*\right) = 1\right) = 0$.*

Impossibility of exact community recovery given $K$ graphs under the condition (3.3) is proved in [41]. Hence, Theorem 2 focuses on proving impossibility for exact community recovery given $K$ graphs under the condition (3.4). In particular, condition (3.4) reveals a parameter regime where exact community recovery from $(G_1, G_2, \ldots, G_K)$ is impossible, yet, if $\boldsymbol{\pi}^*$ were known, then exact community recovery would be possible based on the (correctly matched) union graph.

Theorems 1 and 2 combined give the tight threshold for exact community recovery for general $K$ correlated SBMs, see (2.7). Fig. 2 exhibits phase diagrams illustrating the results for three graphs.

## 3.2 Threshold for exact graph matching

The techniques that we develop in order to prove Theorems 1 and 2 also allow us to solve the question of exact graph matching, that is, exactly recovering $\boldsymbol{\pi}^* = (\pi_{12}^*, \ldots, \pi_{1K}^*)$ from $(G_1, G_2, \ldots, G_K)$. In the context of correlated SBMs and community recovery, exact graph matching can be thought of as an intermediate step towards exact community recovery. However, more generally, graph matching is a fundamental inference problem in its own right; see Section 5 for discussion of related work. We start with the positive direction in the following theorem.

**Theorem 3** (Exact graph matching from $K$ correlated SBMs). *Fix constants $a, b > 0$ and $s \in [0,1]$, and let $(G_1, G_2, \ldots, G_K) \sim \mathrm{CSBM}(n, a\frac{\log n}{n}, b\frac{\log n}{n}, s)$. Suppose that*

$$s\left(1 - (1-s)^{K-1}\right)\mathrm{T_c}(a,b) > 1. \tag{3.5}$$

*Then exact graph matching is possible. That is, there exists an estimator $\widehat{\boldsymbol{\pi}} = \widehat{\boldsymbol{\pi}}(G_1, G_2, \ldots, G_K) = (\widehat{\pi}_{12}, \ldots, \widehat{\pi}_{1K})$ such that $\lim_{n\to\infty} \mathbb{P}(\widehat{\boldsymbol{\pi}}(G_1, G_2, \ldots, G_K) = \boldsymbol{\pi}^*) = 1$.*

Since the condition in (3.5) is weaker than $s^2\mathrm{T_c}(a,b) > 1$ (which is the threshold for exact graph matching for $K = 2$, as shown in [41]), Theorem 3 implies that there exists a parameter regime where $\widehat{\pi}_{12}$ cannot be exactly recovered from $(G_1, G_2)$, but $\widehat{\boldsymbol{\pi}}$ can be exactly recovered from $(G_1, G_2, \ldots, G_K)$. In other words, it is necessary to combine information across all graphs in order to recover $\widehat{\boldsymbol{\pi}}$ (and even just to recover $\widehat{\pi}_{12}$).

The estimator $\widehat{\boldsymbol{\pi}}$ in Theorem 3 is based on pairwise $k$-core matchings (see Sec. 4 for further details). Roughly speaking, for each pairwise $k$-core matching the number of unmatched vertices is $n^{1-s^2\mathrm{T_c}(a,b)+o(1)}$; however, we shall show that the number of vertices which cannot be matched through some combination of pairwise $k$-core matchings is $n^{1-s\left(1-(1-s)^{K-1}\right)\mathrm{T_c}(a,b)+o(1)}$, which is of smaller order. So, when $s\left(1 - (1-s)^{K-1}\right)\mathrm{T_c}(a,b) > 1$, then exact graph matching is possible.

The following impossibility result shows the tightness of Theorem 3.

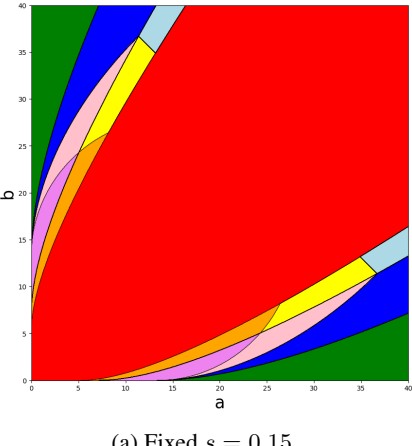

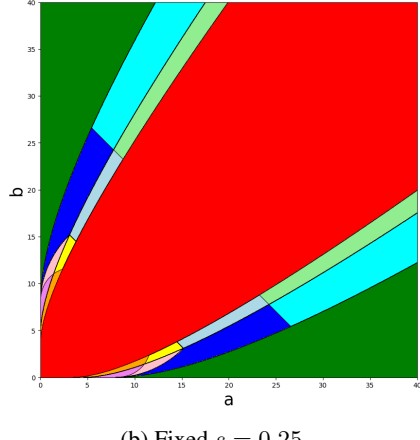

(a) Fixed $s = 0.15$.        (b) Fixed $s = 0.25$.

Figure 2: Phase diagram for exact community recovery for three graphs with fixed $s$, and $a \in [0, 40]$, $b \in [0, 40]$ on the axes. *Green region*: exact community recovery is possible from $G_1$ alone; *Cyan region*: exact community recovery is impossible from $G_1$ alone, but exact graph matching of $G_1$ and $G_2$ is possible, and subsequently exact community recovery is possible from $(G_1, G_2)$; *Dark Blue region*: exact community recovery is impossible from $G_1$ alone, exact graph matching is also impossible from $(G_1, G_2)$, yet exact community recovery is possible from $(G_1, G_2)$; *Pink region*: exact community recovery is impossible from $(G_1, G_2)$ (even though it would be possible if $\pi_{12}^*$ were known), yet exact community recovery is possible from $(G_1, G_2, G_3)$; *Violet region*: exact community recovery is impossible from $(G_1, G_2, G_3)$ (even though it would be possible from $(G_1, G_2)$ if $\pi_{12}^*$ were known); *Light Green region*: exact community recovery is impossible from $(G_1, G_2)$, but exact graph matching of graph pairs is possible, and subsequently exact community recovery is possible from $(G_1, G_2, G_3)$; *Grey region*: exact community recovery is impossible from $(G_1, G_2)$, exact graph matching is also impossible from $(G_1, G_2)$, but exact graph matching is possible from $(G_1, G_2, G_3)$, and subsequently exact community recovery is possible from $(G_1, G_2, G_3)$; *Yellow region*: exact community recovery is impossible from $(G_1, G_2)$, exact graph matching is impossible from $(G_1, G_2, G_3)$, yet exact community recovery is possible from $(G_1, G_2, G_3)$; *Orange region*: exact community recovery is impossible from $(G_1, G_2, G_3)$ (even though it would be possible from $(G_1, G_2, G_3)$ if $\pi^*$ were known); *Red region*: exact community recovery is impossible from $(G_1, G_2, G_3)$ (even if $\pi^*$ is known). The principal finding of this paper is the characterization of the Pink, Violet, Orange, Yellow, Grey, and Light Green regions.

**Theorem 4** (Impossibility of exact graph matching from $K$ correlated SBMs). *Fix constants $a, b > 0$ and $s \in [0, 1]$, and let $(G_1, G_2, \ldots, G_K) \sim \mathrm{CSBM}(n, \frac{a \log n}{n}, \frac{b \log n}{n}, s)$. Suppose that*

$$s \left(1 - (1-s)^{K-1}\right) \mathrm{T_c}(a, b) < 1. \tag{3.6}$$

*Then exact graph matching is impossible. That is, for any estimator $\widetilde{\boldsymbol{\pi}} = \widetilde{\boldsymbol{\pi}}(G_1, G_2, \ldots, G_K) = (\widetilde{\pi}_{12}, \ldots \widetilde{\pi}_{1K})$ we have that $\lim_{n \to \infty} \mathbb{P}\left(\widetilde{\boldsymbol{\pi}}(G_1, G_2, \ldots, G_K) = \boldsymbol{\pi}^*\right) = 0$.*

Theorems 3 and 4 combined give the tight threshold for exact graph matching for general $K$ correlated SBMs, see (2.4). We note that, in independent and concurrent work [5], Ameen and Hajek derived the threshold for exact graph matching from $K$ correlated Erdős–Rényi random graphs; in other words, they proved Theorems 3 and 4 in the special case of $a = b$.

Comparing Theorems 3 and 4 with Theorems 1 and 2, note that there exists a parameter regime where exact community recovery is possible even though exact graph matching is impossible.

## 4   Overview of algorithms and proofs

In this section we elaborate on the technical details of the community recovery algorithm, for which high-level ideas were presented in Section 3. We focus our discussion on the setting of $K = 3$ graphs, which already captures the main technical challenges; we highlight these and explain how

we overcome them. We subsequently explain the generalization from 3 graphs to $K$ graphs. The overview of the impossibility proof is discussed as well.

$k$**-core matching.** We now define a $k$-core matching [13, 22, 43], which is used for almost exact graph matching in Step 2. Given a pair of graphs $(G, H)$ with vertex set $[n]$, for any permutation $\pi$, we have the corresponding intersection graph $G \wedge_\pi H$, where $(i, j)$ is an edge in the intersection graph if and only if $(i, j)$ is an edge in $G$ and $(\pi(i), \pi(j))$ is an edge in $H$. The $k$-core estimator explores all possible permutations $\pi$ of $[n]$ to seek a permutation $\widehat{\pi}$ that maximizes the size of the $k$-core of the intersection graph $G \wedge_\pi H$; recall that the $k$-core of a graph is the maximal induced subgraph for which all vertices have degree at least $k$. The output of the $k$-core estimator is then a partial matching $\widehat{\mu}$, which is the restriction of $\widehat{\pi}$ to the vertex set of the $k$-core in $G \wedge_{\widehat{\pi}} H$.

One significant advantage of using $k$-core matchings is a certain optimality property in terms of performance. Specifically, if $(G_1, G_2) \sim \mathrm{CSBM}(n, a\frac{\log n}{n}, b\frac{\log n}{n}, s)$, then the $k$-core estimator between $G_1$ and $G_2$ fails to match at most $n^{1-s^2\mathrm{T_c}(a,b)+o(1)}$ vertices, which is the same order as the number of singletons of $G_1 \wedge_{\pi_{12}^*} G_2$ and any graph matching algorithm would fail to match these singletons [11, 41, 22]. Another significant benefit of utilizing $k$-core matchings is the correctness of the $k$-core estimator for correlated SBMs, as discussed in [22]. The $k$-core estimator might not be able to match all vertices under the parameter regime that we are interested in; however, every vertex that it does match is matched correctly with high probability.

**Community recovery subroutines.** The high-level summary of the algorithm is as follows. Since exact community recovery might be impossible in $G_1$ alone, we first obtain an initial estimate which gives almost exact community recovery in $G_1$, as described in Step 1. In Step 2, we use pairwise partial $k$-core matchings with $k = 13$ to obtain $\widehat{\boldsymbol{\mu}} := \{\widehat{\mu}_{12}, \widehat{\mu}_{13}, \widehat{\mu}_{23}\}$ (see Fig. 3b), which we will use to combine information across $(G_1, G_2, G_3)$ to recover communities. Note that each partial matching $\widehat{\mu}_{ij}$ only matches a subset of the vertices, denoted as $M_{ij}$; we denote the set of vertices not matched by $\widehat{\mu}_{ij}$ by $F_{ij} := [n] \setminus M_{ij}$. Subsequently, we split the vertices into two categories: "good" vertices and "bad" vertices, where "good" vertices are part of at least two matchings and "bad" vertices are part of at most one matching (see Fig. 3a). We conduct several majority votes among "good" and "bad" vertices to do the clean-up after the graph matching phase, where each subroutine is meticulously executed to disentangle the intricate dependencies among $(G_1, G_2, G_3)$ and $\widehat{\boldsymbol{\mu}}$.

**Exact community recovery for the "good" vertices.** The major distinction between being "good" and being "bad" is that "good" vertices can combine information from all three graphs via their union graph (which is denser), whereas "bad" vertices cannot. Suppose that vertex $i$ is part of $\widehat{\mu}_{12}$ and $\widehat{\mu}_{13}$ (i.e., $i \in M_{12} \cap M_{13}$). We can then identify the union graph $G_1 \vee_{\widehat{\mu}_{12}} G_2 \vee_{\widehat{\mu}_{13}} G_3$, which consists of edges $(i, j)$ such that $(i, j)$ is an edge in $G_1$ or $(\widehat{\mu}_{12}(i), \widehat{\mu}_{12}(j))$ is an edge in $G_2$ or $(\widehat{\mu}_{13}(i), \widehat{\mu}_{13}(j))$ is an edge in $G_3$, and $i$ is part of this union graph. Similarly, if $i \in M_{12} \cap M_{23}$, then $i$ is part of the union graph $G_1 \vee_{\widehat{\mu}_{12}} G_2 \vee_{\widehat{\mu}_{23} \circ \widehat{\mu}_{12}} G_3$ that also integrates information from all three graphs. On the union graph, we can refine the almost exact community labeling by reclassifying "good" vertices based on a majority vote among the labels of their neighbors that are also "good", and this reclassification will be correct as long as the condition (3.1) holds.

There are many underlying technical challenges and roadblocks in the theoretical analysis. The key difficulty arises from the structure of the union graph. It is statistically guaranteed that in $G_1 \vee_{\pi_{12}^*} G_2 \vee_{\pi_{13}^*} G_3$, all vertices have a community label which is the same as the majority community among their neighbors [35]. However, whether this is also the case for $G_1 \vee_{\widehat{\mu}_{12}} G_2 \vee_{\widehat{\mu}_{13}} G_3$ is unclear, since the latter graph is only defined on the "good" vertices $M_{12} \cap M_{13}$. One would like to demonstrate that the removal of "bad" vertices does not significantly affect the majority community among neighbors of "good" vertices. Prior work [22] addressed a similar problem for two graphs by employing a technique known as Łuczak expansion [27] to $F_{12}$ to ensure that the vertices inside the expanded set $\overline{F_{12}}$ are only weakly connected to the vertices outside of the expanded set $[n] \setminus \overline{F_{12}}$. Unfortunately, this method is no longer applicable for correlated SBMs with three or more graphs. Even though the size of the expanded set $\overline{F_{12}}$ is orderwise equal to the size of $F_{12}$, the size of the intersection of the expanded sets $\overline{F_{12}} \cap \overline{F_{13}}$ might not be orderwise equal to the size of $F_{12} \cap F_{13}$, which directly leads to the failure of the algorithm working down to the information-theoretic threshold. To overcome this challenge, we consider the graph $G\{[n] \setminus v\}$ to decouple the dependence of $v$ being connected to a vertex $w$ and $w$ being part of the $k$-core. Applying the Łuczak expansion on such a graph for any given $v$, and through a union bound, we prove that unmatched vertices are contained in the set of vertices whose degree is smaller than a constant, with high probability. This

allows us to quantify the size of $F_{12} \cap F_{13}$ and meanwhile directly ensure that "good" vertices within the $k$-core are only weakly connected with "bad" vertices.

Another hurdle needed to overcome, as stated in [22], concerns the almost exact community recovery in Step 1 which is subsequently used for majority votes. Therefore, it is of great importance to guarantee that the incorrectly-classified vertices are not well-connected and do not have a great impact on majority votes. Consequently, we utilize an algorithm originally developed by Mossel, Neeman, and Sly [35] which allows us to manage the geometry of the misclassified vertices and demonstrate that the vertices classified incorrectly are indeed only weakly connected.

**Exact community recovery for the "bad" vertices.** The remaining step is to label the "bad" vertices. The "bad" vertices can be further classified into three categories (see Fig. 3a): vertices in $F_{12} \cap F_{13}$, which are only matched by $\widehat{\mu}_{23}$ or are not matched by any of the three matchings; vertices in $F_{13} \cap F_{23} \setminus F_{12}$, which are only matched by $\widehat{\mu}_{12}$; and vertices in $F_{12} \cap F_{23} \setminus F_{13}$, which are only matched by $\widehat{\mu}_{13}$.

Consider the vertices in $F_{12} \cap F_{13}$ (the other cases are similar). First of all, as discussed above, we show that $|F_{12} \cap F_{13}| = n^{1-s(1-(1-s)^2)\mathrm{T}_c(a,b)+o(1)}$ with high probability. Consider the graph $G_1 \setminus_{\widehat{\pi}_{12}} G_2 \setminus_{\widehat{\pi}_{13}} G_3$, which consists of the edges $(i,j)$ in $G_1$ such that $(\widehat{\pi}_{12}(i), \widehat{\pi}_{12}(j))$ and $(\widehat{\pi}_{13}(i), \widehat{\pi}_{13}(j))$ are not edges in $G_2$ and $G_3$, respectively. Due to the approximate independence of $F_{12} \cap F_{13}$ and $G_1 \setminus_{\widehat{\pi}_{12}} G_2 \setminus_{\widehat{\pi}_{13}} G_3$, for a vertex $i \in F_{12} \cap F_{13}$ we can calculate the probability of the failure of the majority vote in the graph $G_1 \setminus_{\widehat{\pi}_{12}} G_2 \setminus_{\widehat{\pi}_{13}} G_3$ in a relatively straightforward manner, giving $n^{-s(1-s)^2\mathrm{D}_+(a,b)+o(1)}$. The factor $s(1-s)^2$ arises from the fact that the edges in this graph are subsampled in $G_1$ and are not subsampled in $G_2$ and $G_3$. Now since a vertex in $F_{12} \cap F_{13}$ can have at most 12 edges outside of $F_{12}$ in $G_1 \wedge_{\widehat{\mu}_{12}} G_2$, and also at most 12 edges outside of $F_{13}$ in $G_1 \wedge_{\widehat{\mu}_{13}} G_3$, the majority vote for $i \in F_{12} \cap F_{13}$ essentially does not change whether it is performed in $G_1$ or in $G_1 \setminus_{\widehat{\pi}_{12}} G_2 \setminus_{\widehat{\pi}_{13}} G_3$. Putting all this together, the probability that the majority vote fails is at most:

$$\mathbb{P}(\text{exists a vertex } i \in F_{12} \cap F_{13} \text{ such that the majority vote fails})$$

$$= |F_{12} \cap F_{13}| \times \mathbb{P}(\text{majority vote fails for a vertex}) = n^{1-s(1-(1-s)^2)\mathrm{T}_c(a,b)-s(1-s)^2\mathrm{D}_+(a,b)+o(1)}.$$

Thus, if (3.2) with $K = 3$ holds, then majority vote will correctly classify all vertices in $F_{12} \cap F_{13}$.

**Generalization to $K$ graphs.** For $K$ graphs, we have $\binom{K}{2}$ pairwise matchings to consider (see Fig. 3b). We again categorize the vertices as "good" and "bad". The "good" vertices can integrate information across all $K$ graphs through the pairwise partial $k$-core matchings $\{\widehat{\mu}_{ij} : i, j \in [K], i \neq j\}$, while "bad" vertices cannot. To illustrate this concept more vividly, for any vertex $v$, consider a new "metagraph" $\mathcal{MG}_v$ on $K$ nodes, defined as follows: there is an edge between $i$ and $j$ in $\mathcal{MG}_v$ if and only if $v$ can be matched through $\widehat{\mu}_{ij}$ (see Fig. 4). If the metagraph $\mathcal{MG}_v$ is connected, then there exists a path that can connect all of its $K$ nodes. Equivalently, there exists a set of matchings that allows us to combine information across all $K$ graphs. Subsequently, we quantify the number of "bad" vertices to be $n^{1-s(1-(1-s)^{K-1})\mathrm{T}_c(a,b)+o(1)}$. The remaining analysis for $K$ graphs can be derived by generalizing the analysis for three graphs.

**Impossibility proof.** As discussed in Section 3, we focus on the proof of (3.4) for impossibility. We compute the maximum a posterior (MAP) estimator for the communities in $G_1$. We show that, even with significant additional information provided, including all the correct community labels in $G_2$, the true matchings $\pi_{ij}^*$ for $i, j \in \{2, 3, \ldots, K\}$, and most of the true matching $\pi_{12}^*$ except for singletons in the graph $G_1 \wedge_{\pi_{12}^*} (G_2 \vee \ldots \vee G_K)$, the MAP estimator fails to exactly recovery communities with probability bounded away from 0 if (3.4) holds. The proof is adapted from the MAP analysis in [22]. The difference is that here we are considering $K$ graphs $G_1, G_2, \ldots, G_K$ with different additional information provided for the MAP estimator. Given that the MAP estimator is ineffective under this regime, all other estimators also fail.

**Exact graph matching threshold.** The proof of the exact graph matching threshold is implicitly present in the proof of the exact community recovery threshold. Essentially, since we show that the number of "bad" vertices is $n^{1-s(1-(1-s)^{K-1})\mathrm{T}_c(a,b)+o(1)}$, the condition (3.5) implies that there are no "bad" vertices with high probability. Since all vertices are "good", and "good" vertices can integrate information across all $K$ graphs, the latent matchings can be recovered exactly. For the impossibility result we analyze the MAP estimator and show that, even with significant additional information, including the true matchings $\{\pi_{ij}^* : i, j \in \{2, 3, \ldots, K\}\}$, it fails if (3.6) holds.

# 5 Related work

Our work generalizes—and solves an open question raised by—the work of Gaudio, Rácz, and Sridhar [22]. Just as [22], our work lies at the interface of the literatures on community recovery and graph matching[1]—two fundamental learning problems—which we briefly summarize here.

**Community recovery in SBMs.** A huge research literature exists on learning latent community structures in networks, and this topic is especially well understood for the SBM [25, 15, 34, 35, 33, 36, 2, 3, 7, 1]. Specifically, we highlight the work of [2, 35], which identify the precise threshold for exact community recovery for SBMs with two balanced communities. Our algorithm builds upon their analysis, taking particular care about dealing with the dependencies arising from the multiple inexact partial matchings between $K$ correlated graphs.

**Graph matching: correlated Erdős-Rényi random graphs.** The past decade has seen a plethora of research on average-case graph matching, focusing on correlated Erdős-Rényi random graphs [40]. The information-theoretic thresholds for recovering the latent vertex correspondence $\pi^*$ have been established for exact recovery [11, 46, 12], almost exact recovery [13], and weak recovery [20, 21, 24, 46, 16]. In parallel, a line of work has focused on algorithmic advances [37, 6, 18, 19, 30, 31, 32], culminating in recent breakthroughs that developed efficient graph matching algorithms in the constant noise setting [31, 32]. We particularly highlight the work of Cullina, Kiyavash, Mittal, and Poor [13], who introduced $k$-core matchings and showed their utility for partial matching of correlated Erdős–Rényi random graphs. Subsequent work has shown the power of $k$-core matchings as a flexible and successful tool for graph matching [22, 43, 4]. Our work both significantly builds upon these works, as well as further develops this machinery, which may be of independent interest. We also note the independent and concurrent work of Ameen and Hajek which determined the exact graph matching threshold for $K$ correlated Erdős–Rényi random graphs [5].

**Graph matching: beyond correlated Erdős-Rényi random graphs.** Motivated by real-world networks, a growing line of recent work studies graph matching beyond Erdős-Rényi graphs [8, 26, 9, 39, 42, 49, 41, 22, 45, 43, 17, 48, 47], including for correlated SBMs [29, 39, 28, 41, 22, 48, 47]. The works that are most relevant to ours are [41, 22], which have been discussed extensively above.

# 6 Discussion and Future Work

Our main contribution highlights the power of integrative data analysis for community recovery, yet many open questions still remain.

**Efficient algorithms.** Theorem 1 characterizes when exact community recovery is information-theoretically possible from $K$ correlated SBMs. Is this possible *efficiently* (i.e., in time polynomial in $n$)? The bottleneck in the algorithm that we use to prove Theorem 1, which makes it inefficient, is the $k$-core matching step; the other steps are efficient. Recent breakthrough results have developed efficient graph matching algorithms for correlated Erdős–Rényi random graphs [31, 32], which promisingly suggest that an efficient algorithm for exact community recovery may indeed exist in this regime. We refer to [22] for further discussion on this point.

**General block models.** We focused here on the simplest case of SBMs with two balanced communities. It would be interesting to extend these results to general block models with multiple communities. This is understood well in the single graph setting [1] and recent work has also characterized the threshold for exact graph matching for two correlated SBMs with $k$ symmetric communities [47].

**Alternative constructions of correlated graphs.** An exciting research direction is to study different constructions of correlated graph models. For general $K$, there are many ways that $K$ graphs can be correlated. In particular, the following is a natural alternative construction of multiple correlated SBMs. First, generate $G_0 \sim \mathrm{SBM}(n, p, q)$. Then, independently generate $H_i \sim \mathrm{SBM}(n, p', q')$ for $i \in [K]$. Construct $G_i' := G_0 \vee H_i$, and finally generate $G_i$ through an independent random permutation of the vertex indices in $G_i'$. This construction is equivalent to the one we studied in this paper for $K = 2$ and it is different when $K \geq 3$, and investigating it is interesting and valuable.

---

[1]We note that graph matching has both positive and negative societal impacts. In particular, it is well known that graph matching algorithms can be used to de-anonymize social networks, showing that anonymity is different, in general, from privacy [38]. At the same time, studying fundamental limits can aid in determining precise conditions when anonymity can indeed guarantee privacy, and when additional safeguards are necessary.

## Acknowledgements

We thank Taha Ameen, Julia Gaudio, Elchanan Mossel, and Anirudh Sridhar for helpful discussions. We also thank anonymous reviewers for constructive feedback.

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

# A   Organization

The rest of the paper is structured as follows. First, we elaborate on the recovery algorithm for three graphs in Section C. Section D includes some useful preliminary propositions, including some nice properties of almost exact community recovery on $G_1$. Section E discusses the $k$-core estimator. After these preparations, we are ready to prove the main theorems in the paper.

Section F proves Theorem 1 for three graphs, where we first validate the accuracy of the community labels for "good" vertices and then classify the remaining "bad" vertices. Section G presents the proof of the impossibility result (Theorem 2) for three graphs. Section H discusses the recovery algorithm for $K$ graphs and provides a general proof for $K$ graphs, with additional arguments on how to identify "good" and "bad" vertices. Section I discusses the proof of the impossibility result (Theorem 2) for $K$ graphs. Section J contains the proof of the threshold for exact graph matching given $K$ graphs, that is, the proofs of Theorems 3 and 4.

# B   Notation

We introduce here some notation that will be used in the rest of the paper. In most of the paper we focus on the setting of $K = 3$ graphs: $(G_1, G_2, G_3) \sim \mathrm{CSBM}(n, a\frac{\log n}{n}, b\frac{\log n}{n}, s)$, and this is the setting that we consider here as well.

Let $V := [n] = \{1, 2, 3, ..., n\}$ denote the vertex set of the parent graph $G_0$, and let $V^+ := \{i \in [n] : \sigma^*(i) = +1\}$ and $V^- := \{i \in [n] : \sigma^*(i) = -1\}$ denote the sets of vertices in the two communities. Let $\binom{[n]}{2} := \{(i,j) : i, j \in [n], i \neq j\}$ denote the set of all unordered vertex pairs. Given a community labeling $\sigma \in \{+1, -1\}^n$, we define the set of intra-community vertex pairs as $\mathcal{E}^+(\sigma) := \{(i,j) \in \binom{[n]}{2} : \sigma(i) = \sigma(j)\}$ and the set of inter-community vertex pairs as $\mathcal{E}^+(\sigma) := \{(i,j) \in \binom{[n]}{2} : \sigma(i) = -\sigma(j)\}$. Note that $\mathcal{E}^+(\sigma)$ and $\mathcal{E}^-(\sigma)$ form a partition of $\binom{[n]}{2}$.

Let $A$, $B$, and $C$ denote the adjacency matrices of $G_1$, $G_2$, and $G_3$, respectively. Let $B'$ and $C'$ denote the adjacency matrices of $G_2'$ and $G_3'$, respectively. Note that, by construction, we have for all $(i,j) \in \binom{[n]}{2}$ that $B_{i,j}' = B_{\pi_{12}^*(i), \pi_{12}^*(j)}$ and $C_{i,j}' = C_{\pi_{13}^*(i), \pi_{13}^*(j)}$. Observe that we have the following probabilities for every $(i,j) \in \binom{[n]}{2}$. If $a, b, c \in \{0, 1\}^3$ and $a + b + c > 0$, then

$$\mathbb{P}\left((A_{ij}, B_{ij}', C_{ij}') = (a, b, c)\right) = \begin{cases} s^{a+b+c}(1-s)^{3-a-b-c}p & \text{if } \sigma^*(i) = \sigma^*(j), \\ s^{a+b+c}(1-s)^{3-a-b-c}q & \text{if } \sigma^*(i) \neq \sigma^*(j). \end{cases}$$

Furthermore, we have that

$$\mathbb{P}\left((A_{ij}, B_{ij}', C_{ij}') = (0, 0, 0)\right) = \begin{cases} 1 - p + (1-s)^3 p & \text{if } \sigma^*(i) = \sigma^*(j), \\ 1 - q + (1-s)^3 q & \text{if } \sigma^*(i) \neq \sigma^*(j). \end{cases}$$

# C   The recovery algorithm for three graphs

Our recovery algorithm is based on discovering a matching between subsets of two graphs.

**Definition C.1.** *Let $G_i$ and $G_j$ be two graphs in vertex set $[n]$ with adjacency matrix $A, B$, respectively. The pair $(M_{ij}, \mu_{ij})$ is a matching between $G_i$ and $G_j$ if*

- *$M_{ij} \in [n]$,*

- *$\mu_{ij} : M_{ij} \to [n]$,*

- *$\mu_{ij}$ is injective.*

Given a matching $(M_{ij}, \mu_{ij})$, here are some related notations. Define $G_i \vee_{\mu_{ij}} G_j$ to be the union graph, whose vertex set is $M$, whose vertex index is the same as the vertex index of $G_i$ and whose edge set is $\{\{\ell, m\} : \ell, m \in M_{ij}, A_{\ell m} + B_{\mu_{ij}(\ell), \mu_{ij}(m)} \geq 1\}$. In other words, the edges are those that appear in either $G_i$ or $G_j$. Conversely, $G_i \wedge_{\mu_{ij}} G_j$ represents intersection graph, whose vertex set is $M_{ij}$, whose vertex index is the same as the vertex index of $G_i$ and whose edge set is $\{\{\ell, m\} : \ell, m \in M_{ij}, A_{\ell m} = B_{\mu_{ij}(\ell), \mu_{ij}(m)} = 1\}$. In other words, the edges are those that appear

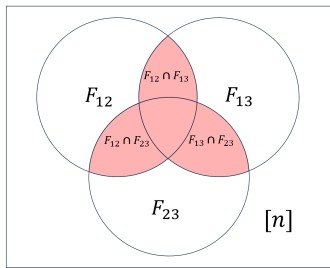
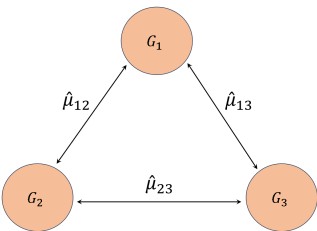

(a) Categorization of vertices for three graphs: vertices in the red regions are "bad" while vertices in the white regions are "good".

(b) Graph matchings for three graphs. For general $K$, consider $\binom{K}{2}$ partial graph matchings.

Figure 3: Schematic landscape of partial matchings over three graphs.

in both $G_i$ and $G_j$. Define $G_i \setminus_{\mu_{ij}} G_j$ to be the graph $G_i$ minus $G_j$, whose vertex set is $M_{ij}$, whose vertex index is the same as the vertex index of $G_i$ and the edges are those only appear on $G_i$ and not appear in $G_j$.

**Definition C.2.** *Let $G_i$ and $G_j, G_k$ be three graphs on vertex set $[n]$ with adjacency matrix $A, B, C$, respectively. The pair $(M_{ij}, \mu_{ij})$ is a matching between $G_i$ and $G_j$, while the pair $(M_{jk}, \mu_{jk})$ is a matching between $G_j$ and $G_k$. Denote $\mu_{jk} \circ \mu_{ij}$ as the composition matching between $G_i$ and $G_k$, defined on the vertex set $M_{ij} \cap M_{jk}$.*

For three graphs, we can define the additional notations in the same manner as in Definition C.1 and the core concepts remain consistent. $G_i \vee_{\mu_{ij}} G_j \vee_{\mu_{jk} \circ \mu_{ij}} G_k$ represents the union graph of $G_i, G_j, G_k$, whose vertex set is $M = M_{ij} \cap M_{jk}$, whose vertex index is the same as the vertex index of $G_i$ and whose edge set is $\{\{\ell, m\} : \ell, m \in M, A_{\ell m} + B_{\mu_{ij}(\ell), \mu_{ij}(m)} + C_{\mu_{jk} \circ \mu_{ij}(\ell), \mu_{jk} \circ \mu_{ij}(m)} \geq 1\}$. In other words, the edge set are the edges that appears in at least one graph out of $G_i, G_j, G_k$. Similarly, $G_i \wedge_{\mu_{ij}} G_j \wedge_{\mu_{jk}} G_k$ represents the intersection graph, the edge set is $\{\{\ell, m\} : \ell, m \in M; A_{\ell m} = B_{\mu_{ij}(\ell), \mu_{ij}(m)} = C_{\mu_{jk} \circ \mu_{ij}(\ell), \mu_{jk} \circ \mu_{ij}(m)} = 1\}$. Define $G_i \vee_{\mu_{ij}} G_j \setminus_{\mu_{jk}} G_k$ be the graph whose edge set is those edges that appear in either $G_i$ or $G_j$ and not appear in $G_k$. Similarly, we can define $G_i \wedge_{\mu_{ij}} G_i \setminus_{\mu_{jk}} G_k$, $G_i \setminus_{\mu_{ij}} (G_i \vee_{\mu_{jk}} G_k)$, and $G_i \setminus_{\mu_{ij}} (G_i \wedge_{\mu_{jk}} G_k)$ as well. Note that all the definitions above are defined on vertex set $M$ and use vertex index in $G_i$.

Introduce $d_{\min}(G) := \min_{i \in [n]} d(i)$, where $d(i)$ is the degree of vertex $i$.

**Definition C.3.** *A matching $(M_{ij}, \mu_{ij})$ is a $k$-core matching of $(G_i, G_j)$ if $d_{\min}(G_i \wedge_{\mu_{ij}} G_j) \geq k$. A matching $(M_{ij}, \mu_{ij})$ is called a maximal $k$-core matching if it involves the greatest number of vertices, among all $k$-core matchings.*

---

**Algorithm 1** $k$-core matching

---

**Input:** Pair of graphs $G_i, G_j$ on $n$ vertices, $k \in [n]$.
**Output:** A matching $(\widehat{M}_{ij}, \widehat{\mu}_{ij})$ of $G_i$ and $G_j$.

1: Enumerating all possible matchings, find the maximal $k$-core matching $(\widehat{M}_{ij}, \widehat{\mu}_{ij})$ of $G_i$ and $G_j$.

---

Let $(M_{ij}, \mu_{ij})$ be the matching found by Algorithm 1 with $k = 13$. $M_{ij}$ coincides with the maximal $k$-core of $G_i \wedge_{\pi_{ij}^*} G_j$, denote it as $M_{ij}^*$ while $\mu_{ij}$ coincides with the true permutation $\pi_{ij}^*$, with high probability (Lemma E.5).

The $k$-core matching is symmetric, i.e. $\mu_{ij}(M_{ij}) = M_{ji}$. Note that by Definition C.1, $M_{ij}$ uses the vertex index of $G_i$ while $M_{ji}$ uses the vertex index of $G_j$, they are equivalent and exchangeable through the 1-1 mapping. Now define $F_{ij} := [n] \setminus M_{ij}$ be the set of vertices which are excluded from the matching. Note that $F_{ij}$ use the vertice index same as $G_i$. We define $F_{ij}^* := [n] \setminus M_{ij}^*$ be the set of vertices which are outside the maximal $k$-core of $G_i \wedge_{\pi_{ij}^*} G_j$.

As briefly discussed in Section 4, we start with leveraging the "good" vertices in order to find the correct communities. The "good" vertices are those which are part of at least two matchings out of three partial matchings $\mu_{12}, \mu_{13}, \mu_{23}$. The details are shown in Algorithm 2.

---

**Algorithm 2** Labeling the good vertices

---

**Input:** Three graphs $G_1, G_2, G_3$ on $n$ vertices and three 13-core matchings $(M_{12}, \mu_{12}, M_{13}, \mu_{13}, M_{23}, \mu_{23})$, parameters $a, b, s, \epsilon$.
**Output:** A labeling of $(M_{13} \cap M_{32}) \cup (M_{12} \cap M_{13}) \cup (M_{23} \cap M_{12})$ given by $\widehat{\sigma}$.

1: Apply [35, Algorithm 1] to the graph $G_1$ and parameters $(sa, sb, \epsilon)$, obtaining a label $\widehat{\sigma_1}$.
2: Denote $F_{12} = [n] \setminus M_{12}, F_{13} = [n] \setminus M_{13}, F_{23} = [n] \setminus M_{23}$.
3: For $i \in M_{13} \cap M_{32}$, set $\widehat{\sigma}(i) \in \{-1, 1\}$ according to the neighborhood majority (resp., minority) of $\widehat{\sigma_1}(i)$ with respect to the graph $(G_1 \vee_{\mu_{32} \circ \mu_{13}} G_2 \vee_{\mu_{13}} G_3)\{M_{13} \cap M_{32}\}$ if $a > b$ (resp., $a < b$).
4: For $i \in M_{12} \cap M_{23}$, set $\widehat{\sigma}(i) \in \{-1, 1\}$ according to the neighborhood majority (resp., minority) of $\widehat{\sigma_1}(i)$ with respect to the graph $(G_1 \vee_{\mu_{12}} G_2 \vee_{\mu_{23} \circ \mu_{12}} G_3)\{M_{12} \cap M_{23}\}$ if $a > b$ (resp., $a < b$).
5: For $i \in M_{13} \cap M_{12}$, set $\widehat{\sigma}(i) \in \{-1, 1\}$ according to the neighborhood majority (resp., minority) of $\widehat{\sigma_1}(i)$ with respect to the graph $(G_1 \vee_{\mu_{12}} G_2 \vee_{\mu_{13}} G_3)\{M_{13} \cap M_{12}\}$ if $a > b$ (resp., $a < b$).
6: Return $\widehat{\sigma} : (M_{13} \cap M_{32}) \cup (M_{12} \cap M_{13}) \cup (M_{23} \cap M_{12}) \to \{-1, 1\}$.

---

The remaining step is to label the "bad" vertices which cannot utilize the combined information from three graphs. Hence, we classify the "bad" vertices according to the majority of neighborhood restricted to the corresponding "good" vertices. The detailed descriptions are shown Algorithm 3.

---

**Algorithm 3** Labeling the bad vertices

---

**Input:** Three graphs $(G_1, G_2, G_3)$ on $n$ vertices and three 13-core matching $(M_{12}, \mu_{12})$, $(M_{13}, \mu_{13})$, and $(M_{23}, \mu_{23})$, parameters $a, b, s$, a label on the "good" vertices $\widehat{\sigma}$.
**Output:** A labeling of $[n]$ given by $\widehat{\sigma}$.

1: For $i \in F_{12} \cap F_{13}$, set $\widehat{\sigma}(i) \in \{-1, 1\}$ according to the neighborhood majority (resp., minority) of $\widehat{\sigma}(i)$ with respect to the graph $G_1(M_{12} \cap M_{13} \cup \{i\})$ if $a > b$. (resp., $a < b$)
2: For $i \in F_{23} \cap F_{13} \setminus F_{12}$, set $\widehat{\sigma}(i) \in \{-1, 1\}$ according to the neighborhood majority (resp., minority) of $\widehat{\sigma}(i)$ with respect to the graph $G_1 \setminus_{\mu_{12}} G_2(M_{12} \cap M_{13} \cup \{i\})$ if $a > b$ (resp., $a < b$).
3: For $i \in F_{12} \cap F_{23} \setminus F_{13}$, set $\widehat{\sigma}(i) \in \{-1, 1\}$ according to the neighborhood majority (resp., minority) of $\widehat{\sigma}(i)$ with respect to the graph $G_1 \setminus_{\mu_{13}} G_3(M_{12} \cap M_{13} \cup \{i\})$ if $a > b$ (resp., $a < b$).
4: Return $\widehat{\sigma} : [n] \to \{-1, 1\}$.

---

The complete exact recovery algorithm is exhibited in Algorithm 4. First, the 13-core matchings are preformed. Next, the "good" vertices are labeled according to the union graph. Finally, the "bad" vertices are labeled according to neighborhood labels in $G_1$ or $G_1 \setminus_{\mu_{12}} G_2$ or $G_1 \setminus_{\mu_{13}} G_3$.

---

**Algorithm 4** Full Community Recovery

---

**Input:** Three graphs $(G_1, G_2, G_3)$ on $n$ vertices, $k = 13$, and $\epsilon > 0$.
**Output:** A labeling of $[n]$ given by $\widehat{\sigma}$.

1: Apply Algorithm 1 on input $(G_i, G_j, k)$, obtaining a matching $(\widehat{M}_{ij}, \widehat{\mu}_{ij}), i \neq j \in \{1, 2, 3\}$. Denote $\widehat{M} := (\widehat{M}_{13} \cap \widehat{M}_{32}) \cup (\widehat{M}_{12} \cap \widehat{M}_{13}) \cup (\widehat{M}_{23} \cap \widehat{M}_{12})$.
2: Apply Algorithm 2 on input $(G_1, G_2, G_3, \widehat{M}_{12}, \widehat{M}_{23}, \widehat{M}_{13}, \widehat{\mu}_{13}, \widehat{\mu}_{12}, \widehat{\mu}_{23})$, obtaining a labeling $\widehat{\sigma} : \widehat{M} \to \{-1, 1\}$.
3: Apply Algorithm 3 on input $(G_1, G_2, G_3, \widehat{M}_{12}, \widehat{M}_{23}, \widehat{M}_{13}, \widehat{\mu}_{13}, \widehat{\mu}_{12}, \widehat{\mu}_{23}, \widehat{\sigma})$, obtaining a labeling $\widehat{\sigma} : [n] \to \{-1, 1\}$.
4: Return $\widehat{\sigma} : [n] \to \{-1, 1\}$.

---

# D   Preliminaries

Here we provide some useful preliminary propositions.

### D.1 Binomial Probabilities

**Lemma D.1.** *Suppose that $a \geq b$. Let $Y \sim \text{Bin}(m^+, a \log(n)/n)$ and $Z \sim \text{Bin}(m^-, b \log(n)/n)$ be independent. If $m^+ = (1 + o(1))n/2, m^- = (1 + o(1))n/2$, then for any $\epsilon > 0$,*

$$\mathbb{P}(Y - Z \leq \epsilon \log n) \leq n^{-D_+(a,b) + \epsilon \log(a/b)/2 + o(1)}.$$

*Proof.* Proved by [22, Lemma 3.3]. $\qquad\square$

### D.2 A useful construction of three correlated stochastic block models

In this section, we elaborate on an alternative method for constructing three correlated SBMs, which emphasizes the independent regions of $G_1, G_2$ and $G_3$. we detail the construction for three graphs to maintain reasonable and manageable notation throughout our discussion. The extension of these ideas to the general case of $K$ graphs follows a similar structure where the key steps and arguments can be directly applied. This construction is analogous to the construction from [22, Section 3.2], generalizing the case from two graphs to three graphs.

Firstly, we construct a random partition $\{\mathcal{E}_{ijk}, i, j, k, \in \{0, 1\}\}$ of $\binom{[n]}{2}$. Independently, for each pair $\{i, j\} \in \binom{[n]}{2}$, we let $\{i, j\} \in \{\mathcal{E}_{ijk}\}$ with a probability of $(1 - s)^{3-i-j-k} s^{i+j+k}$. Subsequently, for each pair $\{i, j\} \in \binom{[n]}{2}$, an edge is constructed between $i$ and $j$ with probability $p$ if the two vertices are in the same community, and with probability $q$ if they are in different communities. Graph $G_1$ is constructed using the edges from $\cup_{j,k \in \{0,1\}, i=1} \mathcal{E}_{ijk}$, while the graph $G_2'$ is constructed using edges from $\cup_{i,k \in \{0,1\}, j=1} \mathcal{E}_{ijk}$. Graph $G_2$ is then generated from $G_2'$ and $\pi_{12}^*$ by relabeling the vertices of $G_2'$ according to $\pi_{12}^*$. Similarly, the graph $G_3'$ is constructed using edges from $\cup_{i,j \in \{0,1\}, k=1} \mathcal{E}_{ijk}$ and $G_3$ is obtained from $G_3'$ and $\pi_{13}^*$ by relabeling the vertices of $G_3'$ according to $\pi_{13}^*$. This construction offers an alternative method for generating multiple correlated SBMs and emphasizes regions of independence between the multiple graphs. The following lemma D.2 describes the idea formally.

**Lemma D.2.** *The random partition construction of correlated SBMs in Section D.2 is equivalent to the original construction shown in Figure 1. Moreover, conditioned on $\boldsymbol{\pi}^* := (\pi_{12}^*, \pi_{13}^*, \pi_{23}^*)$, $\sigma^*$, and $\boldsymbol{\mathcal{E}} := \{\mathcal{E}_{ijk}, i, j, k \in \{0, 1\}\}$, the graphs that are comprised of edges in disjoint $\mathcal{E}_{ijk}$ are mutually independent.*

*Proof.* Firstly we show that the distribution of $(A_{i,j}, B_{\pi_{12}^*(i)\pi_{12}^*(j)}, C_{\pi_{13}^*(i)\pi_{13}^*(j)})$ is the same under two constructions. Then by the indepence of vertex pairs, the equivalence follows. In the first construction,
If $a + b + c > 0$:

$$\mathbb{P}((A_{ij}, B_{\pi_{12}^*(i)\pi_{12}^*(j)}, C_{\pi_{13}^*(i)\pi_{13}^*(j)}) = (a, b, c) | \boldsymbol{\pi}^*, \sigma^*)$$

$$= \begin{cases} s^{a+b+c}(1-s)^{3-a-b-c} p & \text{if } \sigma^*(i) = \sigma^*(j), \\ s^{a+b+c}(1-s)^{3-a-b-c} q & \text{if } \sigma^*(i) \neq \sigma^*(j). \end{cases}$$

If $a + b + c = 0$:

$$\mathbb{P}((A_{ij}, B_{\pi_{12}^*(i)\pi_{12}^*(j)}, C_{\pi_{13}^*(i)\pi_{13}^*(j)}) = (0, 0, 0) | \boldsymbol{\pi}^*, \sigma^*)$$

$$= \begin{cases} 1 - p + (1-s)^3 p & \text{if } \sigma^*(i) = \sigma^*(j), \\ 1 - q + (1-s)^3 q & \text{if } \sigma^*(i) \neq \sigma^*(j). \end{cases}$$

Under the second construction, if $\sigma^*(i) = \sigma^*(j)$:

$$\mathbb{P}((A_{ij}, B_{\pi_{12}^*(i)\pi_{12}^*(j)}, C_{\pi_{13}^*(i)\pi_{13}^*(j)}) = (a, b, c) | \boldsymbol{\pi}^*, \sigma^*) = \mathbb{P}(\{i, j\} \in \mathcal{E}_{abc}) \times p$$

$$= \begin{cases} s^{a+b+c}(1-s)^{3-a-b-c} p & \text{if } a + b + c > 0, \\ 1 - p + (1-s)^3 p & \text{if } a + b + c = 0. \end{cases}$$

If $\sigma^*(i) \neq \sigma^*(j)$, the joint distribution is the same only with $p$ replaced by $q$. We can see that the joint distribution under two constructions is the same. To prove the second part of the lemma, note that conditioned on $\boldsymbol{\pi}^*, \sigma^*$, and the random partition $\{\mathcal{E}_{ijk}, i, j, k \in \{0, 1\}\}$, the edges in $\mathcal{E}_{ijk}$ form independently. Hence the graphs that are comprised of edges in disjoint $\mathcal{E}_{ijk}$ are mutually independent. $\qquad\square$

**Definition D.3.** *Define the constant:*

$$s_{abc} := \begin{cases} s^3 & (a,b,c) = (1,1,1), \\ s^2(1-s) & (a,b,c) \in \{(0,1,1),(1,0,1),(1,1,0)\}, \\ s(1-s)^2 & (a,b,c) \in \{(0,0,1),(1,0,0),(0,1,0)\}, \\ (1-s)^3 & (a,b,c) = (0,0,0). \end{cases}$$

*The event $\mathcal{F}$ holds if and only if*

$$n/2 - n^{3/4} \leq |V^+|, |V^-| \leq n/2 + n^{3/4},$$

*and the following conditions hold for all $a,b,c \in \{0,1\}, i \in [n]$:*

$$s_{abc}(|V^{\sigma^*(i)}| - n^{3/4}) \leq |\{j : j \in \mathcal{E}_{abc} \cap \mathcal{E}^+(\sigma^*(i))\}| \leq s_{abc}(|V^{\sigma^*(i)}| + n^{3/4}),$$
$$s_{abc}(|V^{-\sigma^*(i)}| - n^{3/4}) \leq |\{j : j \in \mathcal{E}_{abc} \cap \mathcal{E}^-(\sigma^*(i))\}| \leq s_{abc}(|V^{-\sigma^*(i)}| + n^{3/4}).$$

**Lemma D.4.** *Define $s_m := \min_{a,b,c \in \{0,1\}} s_{abc}$. We have $\mathbb{P}(\mathcal{F}^c) \leq 100 n \exp(-\frac{s_m^2 \sqrt{n}}{2})$.*

*Proof.* Denote $\mathcal{G}$ holds if and only if $n/2 - n^{3/4} \leq |V^+|, |V^-| \leq n/2 + n^{3/4}$. The event $\mathcal{G}$ is proved in Lemma 3.8 in [22]. $\mathbb{P}(\mathcal{G}^c) \leq 4e^{-\sqrt{n}}$.

Then, look at the remaining condition of event $\mathcal{F}$. Fix $i \in [n]$, condition on $\sigma_1^*, \pi_{12}^*, \pi_{13}^*$. Note that

$$k_{abc}^+(i) := |\{j : j \in \mathcal{E}_{abc} \cap \mathcal{E}^+(\sigma_1^*(i))\}| \sim \text{Bin}(|V^{\sigma^*(i)}| - 1, s_{abc}).$$

By Hoeffding inequality we have

$$\mathbb{P}(|k_{abc}^+(i) - s_{abc}(|V^{\sigma^*(i)}| - 1)| \geq \frac{s_{abc} n^{3/4}}{2} |\pi_{12}^*, \pi_{13}^*, \sigma_1^*) 1(\mathcal{G})$$
$$\leq 2\exp(-\frac{s_{abc}^2 n^{3/2}}{2|V^{\sigma^*(i)}|}) 1(\mathcal{G}) \leq 2\exp(-(1-o(1))s_{abc}^2 \sqrt{n}).$$

Then by a union bound,

$$\mathbb{P}(\exists i \in [n] : |k_{abc}^+(i) - s_{abc}|V^{\sigma^*(i)}|| \geq s_{abc} n^{3/4})$$
$$\leq \sum_{i=1}^{n} \mathbb{P}(|k_{abc}^+(i) - s_{abc}(|V^{\sigma^*(i)}| - 1)| \geq \frac{s_{abc} n^{3/4}}{2})$$
$$\leq \sum_{i=1}^{n} \mathbb{E}[\mathbb{P}(|k_{abc}^+(i) - s_{abc}(|V^{\sigma^*(i)}| - 1)| \geq \frac{s_{abc} n^{3/4}}{2} |\pi_{12}^*, \pi_{13}^*, \sigma_1^*)1(\mathcal{G})] + \mathbb{P}(\mathcal{G}^c)$$
$$\leq 2n \exp(-(1-o(1))s_{abc}^2 \sqrt{n}) + 4\exp(-\sqrt{n}) \leq 6n \exp(-(1-o(1))s_m^2 \sqrt{n}).$$

Similarly, we can define $k_{abc}^-(i)$ and through an identical proof we have

$$\mathbb{P}(\exists i \in [n] : |k_{abc}^-(i) - s_{abc}|V^{-\sigma^*(i)}|| \geq s_{abc} n^{3/4}) \leq 6n \exp(-(1-o(1))s_m^2 \sqrt{n}).$$

The conclusion then follows by a union bound. $\square$

### D.3 Almost exact recovery in a single SBM

**Lemma D.5.** *The algorithm (Algorithm 1, [22]) correctly classifies all vertices in $[n] \setminus I_\epsilon(G)$, where $I_\epsilon(G) := \{v \in [n] : \text{maj}_G(v) \leq \epsilon \log n \text{ or } N(v) > 100 \max\{1, a, b\} \log n\}$ if $a > b$.*

*Proof.* Directly proved by [22, Lemma 5.1], adapted from [35, Proposition 4.3]. $\square$

**Lemma D.6.** *Consider a $\text{SBM}(n, \alpha \log n/n, \beta \log n/n)$, denote $\gamma = \max(\alpha, \beta)$. Then for every $\sigma^*$ we have*
$$\mathbb{P}(\forall i \in [n], |N(i)| \leq 100 \max(1, \gamma) \log n | \sigma^*) \geq 1 - n^{-99}.$$

*Proof.* Directly proved by [22, Lemma 5.2], based on arguments of [35]. $\square$

**Lemma D.7.** *With the assumption of $D_+(a,b) < 99$, $E(|I_\epsilon(G)|) \leq 3n^{1-D_+(a,b)+\epsilon|\log(a/b)|}$.*

*Proof.* Proved by [22, Lemma 5.3]. □

**Lemma D.8.** *If $0 < \epsilon < \frac{D_+(a,b)}{2\log(a/b)}$, then*

$$\mathbb{P}(\forall i \in [n], |N(i) \cap I_\epsilon(G)| \leq 2\lceil D_+(a,b)^{-1}\rceil |\sigma^*) = 1 - o(1).$$

*Proof.* Proved by [22, Lemma 5.4]. □

# E    Analysis of the $k$-core estimator

In this section, we prove two important properties of $k$-core estimator. Lemma E.4 describes that all vertices have weak connections with those vertices who are not part of the $k$-core, in the logarithmic regime that we are interested in. Lemma E.1 argues that all the vertices with degree larger than a given constant will be part of the $k$-core.

**Lemma E.1.** *Fix $a, b > 0$. Consider the graph $G \sim \mathrm{SBM}(n, \frac{a\log n}{n}, \frac{b\log n}{n})$. For any integer $m$ satisfying $m > \frac{2}{a+b}$, all vertices whose degree is greater than $m + k$ are part of the $k$-core with high probability.*

*Proof.* For a given $m > \frac{2}{a+b}$, we would like to prove that any vertex $v$ with degree greater than $k + m$ will not be part of the $k$-core with probability $o(n^{-1})$. The lemma then follows by a union bound.

**Isolating vertex $v$ for independence.**

For a fixed $v \in [n]$, consider the graph $\widetilde{G} := G\{[n] \setminus v\}$. Now we look at the $k$-core of $\widetilde{G}$, denote it by $C_k(\widetilde{G})$. Since the $\deg_G(v) > k + 2/(a+b)$, we can suppose that $\deg_G(v) = m + k, m > 2/(a+b)$. If the vertex $v$ is not part of the $k$-core of $G$, it must has more than $m$ neighbors who are $\notin C_k(\widetilde{G})$. Note that the event $w \notin C_k(\widetilde{G})$ is independent of the event $w \in N_G(v)$, while the latter event is stochastically dominated by a binomial distribution with probability $\nu \log n/n, \nu = \max(a,b)$. Hence, by the tower rule,

$$\mathbb{P}(v \text{ is not part of } k \text{ - core in } G) \leq \mathbb{P}(|\{w \in N_G(v) : w \notin C_k(\widetilde{G})\}| > m)$$
$$\leq \mathbb{E}[\mathbb{1}_{\mathrm{Bin}(|\{w:w\notin C_k(\widetilde{G})\}|,\nu \log n/n)>m}]. \tag{E.2}$$

The size of $\{w : w \notin C_k(\widetilde{G})\}$ can not be directly quantified. Hence, we would like to find a set $\overline{U}$ based on $\widetilde{G}$ such that $\{w \in [n] \setminus v : w \notin C_k(\widetilde{G})\} \subset \overline{U}$, where we can bound the size of $\overline{U}$. Now we denote $\mu = |\overline{U}|\nu \log n/n$, then we have that

$$\mathbb{E}[\mathbb{1}_{\mathrm{Bin}(|\{w:w\notin C_k(\widetilde{G})\}|,\nu \log n/n)>m}] \leq \mathbb{E}[\mathbb{1}_{\mathrm{Bin}(|\overline{U}|,\nu \log n/n)>m}] \leq \mathbb{E}[\min_{t>0}\exp(\mu(e^t - 1) - tm)]. \tag{E.3}$$

To construct $\overline{U}$, the idea is motivated by Łuczak expansion in [27]. We consider a modified version of expanding the set in our setting.

**Quantify the set $U$.**

Define $U$ to be the set of vertices with degree at most $T$ in the graph $\widetilde{G}$. The choice of $T$ would be specified later. Denote $\mathcal{H} := \{n/2 - n^{3/4} \leq |V^+|, |V^-| \leq n/2 + n^{3/4}\}$. By Lemma D.4,

$\mathbb{P}(\mathcal{H}^c) = o(1/n)$.

$\mathbb{E}(|U|) \le \mathbb{E}(|U|1_{\mathcal{H}}) + \mathbb{P}(\mathcal{H}^c)n = \mathbb{E}(|U|1_{\mathcal{H}}) + o(1)$

$$\le n \sum_{i=0}^{T} \sum_{j=0}^{i} \binom{(1+o(1)\frac{n}{2})}{j} \binom{(1+o(1)\frac{n}{2})}{i-j} p^j (1-p)^{(1-o(1))\frac{n}{2}-j} q^{i-j} (1-q)^{(1-o(1))\frac{n}{2}-i+j}$$

$$\le 2n(1 - \frac{a\log n}{n})^{(1-o(1))\frac{n}{2}} (1 - \frac{b\log n}{n})^{(1-o(1))\frac{n}{2}} \sum_{i=0}^{T} \sum_{j=0}^{i} \left( \frac{(1+o(1))n}{2} \right)^i (\frac{\nu \log n}{n})^i$$

$$\le 2n^{1-\frac{a+b}{2}+o(1)} \sum_{i=0}^{T} (i+1) \left( \frac{(1+o(1))\nu \log n}{2} \right)^i$$

$$\le 2(T+1)^2 (\frac{\nu \log n}{2})^T n^{1-\frac{a+b}{2}+o(1)} = n^{1-\frac{a+b}{2}+o(1)}.$$

Consider the situation when $1 - \frac{a+b}{2} > 0$. Now we claim: for any constant $W$, $\mathbb{E}[|U|^W] \le n^{W-W\frac{a+b}{2}+o(1)}$. Suppose for $W-1$, it is true, then for $W$:

$$\mathbb{E}[|U|^W] = \sum_{i_1,\dots,i_W} \mathbb{P}(i_1 \in U, \dots, i_W \in U)$$

$$= \sum_{i_1 \ne i_2 \dots \ne i_W} \mathbb{P}(i_1 \in U, \dots, i_W \in U)$$

$$+ \sum_{i_1 \ne i_2 \dots \ne i_{W-1}} \mathbb{P}(i_1 \in U, \dots, i_{W-1} \in U) + \dots + \sum_{i_1 \in [n]} \mathbb{P}(i_1 \in U)$$

$$\le \sum_{i_1 \ne i_2 \dots \ne i_W} \mathbb{P}(i_1 \in U, \dots, i_W \in U) + \mathbb{E}(|U|^{W-1})$$

$$\le \sum_{i_1 \ne i_2 \dots \ne i_W} \mathbb{P}(i_1 \in U, \dots, i_W \in U) + \mathbb{E}[|U|^{W-1}].$$

The remaining thing is to show that $\sum_{i_1 \ne i_2 \dots \ne i_W} \mathbb{P}(i_1 \in U, \dots, i_W \in U) \le n^{W-W\frac{a+b}{2}+o(1)}$. We have

$$\sum_{i_1 \ne i_2 \dots \ne i_W} \mathbb{P}(i_1 \in U, \dots, i_W \in U)$$

$$\le \sum_{i_1 \ne i_2 \dots \ne i_W} \mathbb{P}(\cap_{j=1}^{W} \{i_j \text{ has at most T neighbours in } [n] \setminus i_1, \dots, i_W, v\})$$

$$= \sum_{i_1 \ne i_2 \dots \ne i_W} \mathbb{P}(\{i_1 \text{ has at most T neighbours in } [n] \setminus i_1, \dots, i_W, v\})^W$$

$$\le \mathbb{E}[|U|]^W \le n^{W-W\frac{a+b}{2}+o(1)}.$$

The first inequality is because that if $i_1 \in U$ in $\widetilde{G}$, then $i_1$ has at most $T$ neighbours in $[n] \setminus v$, then it implies that $\{i_1$ has at most $T$ neighbours in $[n] \setminus i_1, \dots, i_W, v\}$. The second inequality is due to the independence of the events. By induction, the claim follows. By choosing appropriate $m'$-th moment method of $|U|$, we can select a $\epsilon$ such that $\mathbb{P}(|U| \ge n^{1-\epsilon}) \le n^{-m'(a+b)/2+m'\epsilon+o(1)} = o(n^{-m\frac{a+b}{2}})$. Denote $\mathcal{D}_0$ as $|U| \le n^{1-\epsilon}$, then $\mathbb{P}(\mathcal{D}_0^c) = o(n^{-m\frac{a+b}{2}})$.

**The possibility of the existence of a well-connected small subgraph.**

Now we would like to bound the probability of the existence of a well-connected small subgraph. Define the event

$$\mathcal{D} := \{\text{there exists } S \in [n] \text{ such that } |S| < n^{1-\epsilon} \text{ and } G\{S\} \text{ has at least } N|S| \text{ edges }\}.$$

We would like to bound $\mathbb{P}(\mathcal{D}) = o(n^{-m(a+b)/2})$. Let $S$ be a $\kappa$-vertex subset of $[n]$. Let $X_S$ be the indicator variable that is 1 if the subgraph induced by $S$ has at least $N|S|$ edges. Denote

$\nu = \max(a, b)$, then we have

$$\mathbb{E}[\sum_{S \in [n], |S| = \kappa} X_S] \le \sum_{S \in [n], |S| = \kappa} \binom{\binom{\kappa}{2}}{N\kappa} (\frac{\nu \log n}{n})^{N\kappa} \le \sum_{S \in [n], |S| = \kappa} (\frac{\kappa e \nu \log n}{Nn})^{N\kappa}$$

$$\le \binom{n}{\kappa} (\frac{\kappa e \nu \log n}{Nn})^{N\kappa} \le ((\frac{e^{1+1/N} \nu \log n}{N})^N (\frac{\kappa}{n})^{N-1})^\kappa.$$

The second and the third inequality is because $\binom{n}{k} \le (\frac{en}{k})^k$. Under the assumption $|S| < n^{1-\epsilon}$,

$$(\frac{e^{1+1/N} \nu \log n}{N})^N (\frac{\kappa}{n})^{N-1} < n^{-\epsilon(N-1)+o(1)}.$$

Hence for $n$ sufficiently large we have:

$$\mathbb{E}[\sum_{S \in [n], |S| < n^{1-\epsilon}} X_S] \le \sum_{\kappa=1}^{n^{1-\epsilon}} (n^{-\epsilon(N-1)+o(1)})^\kappa \le n^{-\epsilon(N-1)+o(1)}.$$

Then by Markov's inequality, $\mathbb{P}(\mathcal{D}) \le n^{-\epsilon(N-1)+o(1)}$. If we want to bound $\mathbb{P}(\mathcal{D}) = o(n^{-m(a+b)/2})$, set $N > \frac{(a+b)m}{2\epsilon} + 1$. Hence,

$$\mathbb{P}(\mathcal{D}) \le \mathbb{E}[\sum_{S \in [n], |S| < n^{1-\epsilon}} X_S] \le n^{-\epsilon(N-1)+o(1)} < n^{-m(a+b)/2}.$$

**Identify the expansion set of $U$.**

Now we do the following expansion on $U$, the expansion process is adapted from Łuczak expansion first introduced in [27].

1. Define $U_0 := U$.

2. Given $U_t$, define $U_{t+1}^1$ to be the set of those vertices outside $U_t$ which have at least $N'$ neighbors in $U_t$. If $U_{t+1}^1$ is non-empty, set $U_{t+1} = U_t \cup \{u\}$, where $u$ is the first vertex in $U_{t+1}^1$. Otherwise, stop the expansion with the set $U_t$.

Suppose the expansion ends at the step $h$, hence we have an increasing sequence $\{U_s\}_{s=0}^h$. Denote $\overline{U} := U_h$ to be the set after expansion.

Now claim that on the event $\mathcal{D}^c \cap \mathcal{D}_0$, we can choose $N_1, N' > 0$, such that $|\overline{U}| \le N_1 |U|$. Suppose that $|\overline{U}| > N_1 |U|$, then there exists $\ell > 0$ s.t. $|U_\ell| = N_1 |U|$. On event $\mathcal{D}_0$, there exists $\epsilon > 0, |U| \le n^{1-\epsilon}$, hence $|U_\ell| = N_1 |U| \le n^{1-\epsilon+o(1)}$. Denote $e_l$ as the number of edges in $\widetilde{G}\{U_\ell\}$. Each step in the expanding process, at least $N'$ edges are added into the graph, hence $e_\ell \ge N' \ell \ge N'(|U_\ell| - |U|) = (N' - \frac{N'}{N_1})|U_\ell|$. We can choose $N', N_1$ such that $(N' - \frac{N'}{N_1}) \ge N$. However on the event $\mathcal{D}^c$, the set $|U_\ell| < n^{1-\epsilon}$ cannot have at least $N|U_\ell|$ edges, which is a contradiction. Therefore, on the event $\mathcal{D}^c \cap \mathcal{D}_0, |\overline{U}| \le N_1 |U|$. Subsequently,

$$\mathbb{E}[|\overline{U}|^m] \le N_1^m E[|U|^m 1_{\mathcal{D}^c \cap \mathcal{D}_0}] + n^m \mathbb{P}(\mathcal{D}) + n^m \mathbb{P}(\mathcal{D}_0)$$

$$\le N_1^m \mathbb{E}[|U|^m] + o(n^{m-m(a+b)/2}) \le n^{m-m\frac{a+b}{2}+o(1)}.$$

**Bound the probability of $v$ being in the $k$-core.**

Note that $\widetilde{G}\{[n] \setminus \overline{U}\}$ has minimum degree at least $T - N'$. If a vertex $i \in [n] \setminus \overline{U}$, then $i \notin U$, it follows that $i$ has degree at least $T$ in $\widetilde{G}$. However $i$ can have at most $N'$ neighbor in $\overline{U}$ by construction of expansion process, so $i$ must have at least $T - N'$ neighbors in $[n] \setminus \overline{U}$. We can set $T = k + N'$, then If $i \in [n] \setminus \overline{U}$, $i$ is part of $k$-core in $\widetilde{G}$.

Since the $\deg(v) \ge m + k$, it must has at least $m$ neighbors who are not part of $k$-core in $\widetilde{G}$.

Follow the equation (E.2), (E.3), denote $\mu = |\overline{U}|\nu \log n/n$, then we have

$$\mathbb{E}[1_{\text{Bin}(|\overline{U}|,\nu \log n/n)>m}] \le \mathbb{E}[\min_{t>0} \exp(\mu(e^t - 1) - tm)] \le \mathbb{E}[e\mu^m] + o(n^{-1})$$

$$= \frac{e\nu^m \log n^m}{n^m}\mathbb{E}[|\overline{U}|^m] + o(n^{-1}) \le n^{-(a+b)m/2+o(1)} + o(n^{-1})$$

$$= o(n^{-1}).$$

The second inequality follows by setting $t = \log(1/\mu)$. This is valid since $\mathbb{P}(|U| > n^{1-c}) \le \frac{\mathbb{E}[|U|^W]}{n^{(1-c)W}} \le n^{-((a+b)/2-c)W+o(1)}$. We can select $0 < c < \frac{a+b}{2}, W > 0$ such that $\mathbb{P}(|U| > n^{1-c}) = o(n^{-1})$. On the event $|U| \le n^{1-c}$, $\mu = o(1), 1/\mu > 1$. The third inequality follows by $\mathbb{E}[|\overline{U}|^m] \le n^{m-m\frac{a+b}{2}+o(1)}$ and the last equality is due to $m > \frac{2}{a+b}$.

If $1 - (a + b)/2 \le 0$, we can directly set $m = 1$. Similarly, we can prove $\mathbb{E}[|\overline{U}|] \le n^{1-\frac{a+b}{2}+o(1)}$, then through a similar calculation, $\mathbb{P}(\text{ v is not part of }k\text{-core in }G\}|) = o(n^{-1})$.

Hence, by a union bound, we can say that all vertices with degree larger than $m + k$ will be part of $k$-core with high probability. $\qquad\square$

**Lemma E.4.** *Fix* $a, b, \varepsilon > 0$. *Let* $G \sim \text{SBM}(n, \frac{a \log n}{n}, \frac{b \log n}{n})$. *Then, w.h.p., all vertices have at most* $\varepsilon \log n$ *neighbors who are not part of the* $k$-core.

*Proof.* For $v \in [n]$, consider the graph $\widetilde{G} := G\{[n]\backslash v\}$. Following the same arguments in Lemma E.1, we can obtain $\overline{U}$. We have

$$\mathbb{P}(|\{w \in N_G(v) : w \text{ is not part of }k\text{-core in }G\}| > \epsilon \log n)$$
$$\le \mathbb{P}(|\{w \in N_G(v) : w \notin C_k(\widetilde{G})\}| > \epsilon \log n)$$
$$\le \mathbb{P}(|\{w \in N_G(v) : w \notin C_k(\widetilde{G})\}| > m) \le o(n^{-1}).$$

Based on the proof of Lemma E.1, we can show the lemma follows immediately when $n$ is sufficiently large. Hence, all vertices have at most $\epsilon \log n$ neighbors who are not part of the $k$- core with probability $1 - o(1)$. $\qquad\square$

**Lemma E.5.** *Fix constants* $a, b > 0, s \in [0, 1]$. *Let* $(G_1, G_2) \sim \text{CSBM}(n, a\frac{\log n}{n}, b\frac{\log n}{n}, s)$. *Let* $M^*$ *be the set of vertices of the 13-core in the graph* $G_1 \wedge_{\pi^*} G_2$. $\pi^*$ *is the permutation of vertices from* $G_1$ *to* $G_2$. *Let* $(M_1, \mu_1)$ *be the output of the* $k$-core *match of* $(G_1, G_2), k = 13$. *Then* $\mathbb{P}((M_1, \mu_1) = (M^*, \pi^*)) = 1 - o(1)$.

**Remark:** We can replace $(M_k, \mu_k)$ by $(M_k^*, \pi_k^*\{M^*\})$ in any analysis.

*Proof.* Proved by [22, Lemma 4.8]. $\qquad\square$

**Lemma E.6.** *Let* $G \sim \text{SBM}(n, a\frac{\log n}{n}, b\frac{\log n}{n})$ *for fixed* $a, b > 0$. *Fix* $k \ge 1$. *With probability* $1 - o(1)$, *we have that* $|F| \le n^{1-\text{T}_c(a,b)+o(1)}$.

*Proof.* Proved by [22, Lemma 4.13]. $\qquad\square$

**Lemma E.7.** *Suppose* $G_1, G_2, G_3$ *are independently subsampled with probability* $s$ *from a parent graph* $G \sim \text{SBM}(n, a \log n/n, b \log n/n)$ *for* $a, b > 0$. *Let* $F_{ij}^*$ *be the set of vertices outside the* $k$-core *of* $G_i \wedge_{\pi_{ij}} G_j$ *(taking vertice index in i) with* $k = 13$. *Prove that with probability* $1 - o(1)$, $|F_{ij}^* \cap F_{jk}^*| \le n^{1-(2s^2-s^3)\text{T}_c(a,b)+\delta}$, *for any* $\delta > 0$.

*Proof.* $|F_{12}^* \cap F_{23}^*| = |F_{21}^* \cap F_{23}^*| = |F_{12}^* \cap F_{13}^*|$, by symmetricity of $G_1, G_2, G_3$.

Define $U_{ij}$ to be the set of vertices with degree at most $m + k$ in the intersection graph $G_i \wedge_{\pi_{ij}} G_j$ which marginally follows $\text{SBM}(n, \frac{as^2 \log n}{n}, \frac{bs^2 \log n}{n})$, and $m$ is an integer satisfying $m > \frac{2}{s^2(a+b)}$. Then by Lemma E.1, w.h.p., $F_{ij}^* \subset U_{ij}$. Hence $|F_{12}^* \cap F_{23}^*| \le |U_{12} \cap U_{23}|$ with high probability.

It thus remains to bound $|U_{12} \cap U_{13}|$. Firstly, we bound the expectation of $|U_{12} \cap U_{13}|$:

$$\mathbb{E}[|U_{12} \cap U_{13}|] = \sum_{v=1}^{n} \mathbb{E}[\mathbf{1}_{v \in U_{12} \cap U_{13}}] = n\mathbb{E}[\mathbf{1}_{v \in U_{12}} \mathbf{1}_{v \in U_{13}}].$$

Let $D_1$ denote the degree of vertex $v$ in the graph $G_1$. Let $X_a \sim \text{Bin}((1 + o(1))n/2, sa \log n/n)$, and $X_b \sim \text{Bin}((1 + o(1))n/2, sb \log n/n)$. On the event $\mathcal{F}$, $D_1 \stackrel{d}{=} X_a + X_b$, where $\mathcal{F}$ is defined in Definition D.3, $X_a, X_b$ are independent. Note that by Lemma D.4, $\mathbb{P}(\mathcal{F}^c) = o(\frac{1}{n^2})$. We have

$$\mathbb{E}[\mathbf{1}_{v \in U_{12}} \mathbf{1}_{v \in U_{13}}] = \mathbb{E}[\mathbb{E}[\mathbf{1}_{v \in U_{12}} \mathbf{1}_{v \in U_{13}} | D_1]] = \mathbb{E}\left[\left(\sum_{i=0}^{m+k} \binom{D_1}{i} s^i (1 - s)^{D_1 - i}\right)^2\right]$$

$$= \sum_{i=0}^{m+k} \sum_{j=0}^{m+k} C(i,j) \mathbb{E}[D_1^{i+j}(1 - s)^{2D_1}]. \tag{E.8}$$

Here $C(i,j)$ is a constant related to $i, j$. Now look at $\mathbb{E}[D_1^L(1 - s)^{2D_1}]$. In our regime, $L \le 2(m+k)$ are constant. Hence:

$$\mathbb{E}[D_1^L(1 - s)^{2D_1} \mathbf{1}_{\mathcal{F}}] = \mathbb{E}[(X_a + X_b)^L(1 - s)^{2X_a}(1 - s)^{2X_b} \mathbf{1}_{\mathcal{F}}]$$

$$= \sum_{t=0}^{K} C_t \mathbb{E}[X_a^t(1 - s)^{2X_a} \mathbf{1}_{\mathcal{F}}] \mathbb{E}[X_b^{L-t}(1 - s)^{2X_a} \mathbf{1}_{\mathcal{F}}]. \tag{E.9}$$

Here $C_t$ is constant related to $t$, the second equality is due to the independence of $X_a, X_b$. Now look at $\mathbb{E}[X_a^t(1 - s)^{2X_a} \mathbf{1}_{\mathcal{F}}]$.

$$\mathbb{E}[X_a^t(1 - s)^{2X_a} \mathbf{1}_{\mathcal{F}}] \le \mathbb{E}[X_a^t(1 - s)^{2X_a}]$$

$$= \sum_{\ell=0}^{(1+o(1))n/2} \ell^t(1 - s)^{2\ell} \left(\frac{sa \log n}{n}\right)^{\ell} \left(1 - \frac{sa \log n}{n}\right)^{(1+o(1))n/2 - \ell}$$

$$= \sum_{\ell=0}^{(\log n)^3} \ell^t \left(\frac{(1 - s)^2 sa \log n}{n}\right)^{\ell} \left(1 - \frac{sa \log n}{n}\right)^{(1+o(1))n/2 - \ell}$$

$$+ \sum_{\ell=(\log n)^3 + 1}^{(1+o(1))n/2} \ell^t \left(\frac{(1 - s)^2 sa \log n}{n}\right)^{\ell} \left(1 - \frac{sa \log n}{n}\right)^{(1+o(1))n/2 - \ell}.$$

We can bound the first part:

$$\sum_{\ell=0}^{(\log n)^3} \ell^t \left(\frac{(1 - s)^2 sa \log n}{n}\right)^{\ell} \left(1 - \frac{sa \log n}{n}\right)^{(1+o(1))n/2 - \ell}$$

$$\le (\log n)^{3t} \sum_{\ell=0}^{(\log n)^3} \left(\frac{(1 - s)^2 sa \log n}{n}\right)^{\ell} \left(1 - \frac{sa \log n}{n}\right)^{(1+o(1))n/2 - \ell}$$

$$\le (\log n)^{3t} \sum_{\ell=0}^{(1+o(1))n/2 - \ell} \left(\frac{(1 - s)^2 sa \log n}{n}\right)^{\ell} \left(1 - \frac{sa \log n}{n}\right)^{(1+o(1))n/2 - \ell}$$

$$= (\log n)^{3t} \left(1 - (1 - (1 - s)^2)\frac{sa \log n}{n}\right)^{(1+o(1))n/2} \le n^{-(1-(1-s)^2)sa/2 + o(1)}$$

Then we can bound the second part, note that

$$\sum_{\ell=(\log n)^3 + 1}^{(1+o(1))n/2} \ell^t \left(\frac{(1 - s)^2 sa \log n}{n}\right)^{\ell} \left(1 - \frac{sa \log n}{n}\right)^{(1+o(1))n/2 - \ell}$$

$$= \sum_{\ell=(\log n)^3 + 1}^{(1+o(1))n/2} \ell^t(1 - s)^{2\ell} \mathbb{P}(X_a = \ell) \le (\log n)^{3t}(1 - s)^{2(\log n)^3} \mathbb{P}(X_a > (\log n)^3)$$

$$\le n^{2(\log n)^2 \log(1-s) + o(1)} = o(n^{-(1-(1-s)^2)sa/2 + o(1)}).$$

The first inequality is true because $\ell^t(1-s)^{2\ell}$ decreases when $\ell > (\log n)^3$ for sufficiently large $n$. The last equality is true because $(\log n)^2 \log(1-s) < -(1-(1-s)^2)sa/2$ for sufficiently large $n$.

Hence, by summing up the two parts, $\mathbb{E}[X_a^t(1-s)^{2X_a}] \leq n^{-s(1-(1-s)^2)a/2+o(1)}$, similarly we can proof $\mathbb{E}[X_b^{L-t}(1-s)^{2X_b}] \leq n^{-s(1-(1-s)^2)b/2+o(1)}$. By (E.8) and (E.9),

$$\mathbb{E}[\mathbf{1}_{v \in U_{12}} \mathbf{1}_{v \in U_{13}}] \leq \mathbb{P}(\mathcal{F})^c + n^{-s(1-(1-s)^2)\mathrm{T}_c(a,b)+o(1)} = n^{-s(1-(1-s)^2)\mathrm{T}_c(a,b)+o(1)} + o\left(\frac{1}{n^2}\right).$$

By Markov's inequality we have

$$\mathbb{P}(|U_{12} \cap U_{13}| \geq \log n \mathbb{E}[|U_{12} \cap U_{13}|]) \leq \frac{1}{\log n} = o(1).$$

Hence we have $|F_{12}^* \cap F_{23}^*| \leq |U_{12} \cap U_{13}| \leq \log n \mathbb{E}[|U_{12} \cap U_{13}|] \leq n^{1-s(1-(1-s)^2)\mathrm{T}_c(a,b)+\delta}$, for any $\delta > 0$, with probability $1 - o(1)$. $\qquad\square$

# F   Proof of Theorem 1 for three graphs

## F.1   Exact recovery in $[n] \setminus (F_{12} \cup F_{23}) \cup [n] \setminus (F_{12} \cup F_{13}) \cup [n] \setminus (F_{13} \cup F_{23})$

**Definition F.1.** *For a vertex $i$ in $G$, define the quantity majority of $i$:*

$$\mathsf{maj}_G(i) := |N_G(i) \cap V^{\sigma^*(i)}| - |N_G(i) \cap V^{-\sigma^*(i)}|.$$

By Lemma D.1, we can directly deduce that if $(1-(1-s)^3)D_+(a,b) > 1 + \epsilon|\log(a/b)|$, then for all $i \in [n]$ we have that $\mathsf{maj}_{G_1 \vee_{\pi_{12}} G_2 \vee_{\pi_{13}} G_3}(i) \geq \epsilon \log n$ with probability $1 - o(1)$.

## F.2   Exact recovery in $[n] \setminus (F_{12} \cup F_{23})$

Now suppose that $i \in M_{12} \cap M_{23}$. Look at $\widetilde{G} := (G_1 \vee_{\mu_{12}} G_2 \vee_{\mu_{23} \circ \mu_{12}} G_3)([n] \setminus \{F_{12} \cup F_{23}\})$.

**Lemma F.2.** *Suppose that $(1-(1-s)^3)D_+(a,b) > 1 + 2\epsilon|\log(a/b)|$. Then with probability $1 - o(1)$, all vertices in $(M_{12} \cap M_{23})$ have an $\epsilon \log n$ majority in $\widetilde{G}\{M_{12} \cap M_{23}\}$.*

*Proof.* Denote $F_{ij}^*$ the set of vertices outside the 13-core of $G_i \wedge_{\pi_{ij}^*} G_j$. In light of Lemma E.5 and its remark, we can replace $\mu_{ij}$ with $\pi_{ij}^*$, $F_{ij}$ with $F_{ij}^*$ in Lemma F.2. Where we define

$$G^* := G_1 \vee_{\pi_{12}^*} G_2 \vee_{\pi_{13}^*} G_3,$$

$$H := G^*\{M_{12}^* \cap M_{23}^*\}.$$

To bound the neighborhood majority in $H$, for $i \in M_{12}^* \cap M_{23}^*$ note that:

$$\mathsf{maj}_H(i) = \sigma^*(i) \sum_{j \in N_H(i)} \sigma^*(j) \leq \mathsf{maj}_{G^*}(i) + |N_{G^*}(i) \cap \{F_{12}^* \cup F_{23}^*\}|,$$

$$\mathsf{maj}_H(i) = \sigma^*(i) \sum_{j \in N_H(i)} \sigma^*(j) \geq \mathsf{maj}_{G^*}(i) - |N_{G^*}(i) \cap \{F_{12}^* \cup F_{23}^*\}|.$$

To sum up, we have

$$|\mathsf{maj}_H(i) - \mathsf{maj}_{G^*}(i)| \leq |N_{G^*}(i) \cap \{F_{12}^* \cup F_{23}^*\}|. \tag{F.3}$$

Note that $\mathsf{maj}_{G^*}(i) > 2\epsilon \log n, i \in [n]$ with probability $1 - o(1)$, given that $(1-(1-s)^3)D_+(a,b) > 1 + 2\epsilon|\log(a/b)|$ by Lemma D.1. Now we prove that the right hand side of (F.3) can be bounded by $\epsilon \log n$. Look at $|N_{G^*}(i) \cap \{F_{12}^* \cup F_{23}^*\}|$,

$$|N_{G^*}(i) \cap \{F_{12}^* \cup F_{23}^*\}| \leq |N_{G_1 \wedge G_2}(i) \cap (F_{12}^*)| + |N_{G_2 \wedge G_3}(\pi_{12}^*(i)) \cap (F_{23}^*)|$$
$$+ |N_{G^* \setminus (G_1 \wedge G_2)}(i) \cap (F_{12}^*)| + |N_{G^* \setminus (G_2 \wedge G_3)}(i) \cap (F_{23}^*)|.$$

First, look at $|N_{G_1 \wedge G_2}(i) \cap F_{12}^*|$, by Lemma E.4, w.h.p.,

$$|N_{G_1 \wedge G_2}(i) \cap F_{12}^*| < \epsilon \log n/8.$$

Similarly, we have w.h.p.

$$|N_{G_2 \wedge G_3}(\pi_{12}^*(i) \cap F_{23}^*)| < \epsilon \log n / 8.$$

What is left is to bound $|N_{G^* \setminus (G_2 \wedge G_3)}(i) \cap (F_{23}^*)|, |N_{G^* \setminus (G_1 \wedge G_2)}(i) \cap (F_{12}^*)|$.

Note that conditioned on $\pi_{12}^*, \pi_{13}^*, \pi_{23}^*, \sigma^*, \boldsymbol{\mathcal{E}} := \{\mathcal{E}_{ijk}, i, j, k \in \{0, 1\}\}$, the graph $G^* \setminus (G_1 \wedge G_2)$ is independent of $F_{12}^*$ by Lemma D.2, since $F_{12}^*$ depends only on $G_1 \wedge G_2$. Thus we can stochastically dominate $|N_{G^* \setminus (G_1 \wedge G_2)}(i) \cap F_{12}^*|$ by a Poisson random variable X with mean

$$\lambda_n := \nu \frac{\log n}{n} |\{j \in F_{12}^* : \{i, j\} \in \mathcal{E}_{100} \cup \mathcal{E}_{101} \cup \mathcal{E}_{001} \cup \mathcal{E}_{010} \cup \mathcal{E}_{011}\}| \leq \nu \frac{\log n}{n} |F_{12}^*|, \nu := max(a, b).$$

For a fixed $\delta > 0$, define an event $\mathcal{Z} := \{|F_{12}^*| \leq n^{1 - s^2 \mathrm{T_c}(a,b) + \delta}\}$. On $\mathcal{Z}$, we have that $\lambda_n \leq n^{-s^2 \mathrm{T_c}(a,b) + \delta + o(1)}$. Hence, for any positive integer m:

$$\mathbb{P}(\{|N_{G^* \setminus (G_1 \wedge G_2)}(i) \cap (F_{12}^*)| \geq m\} \cap \mathcal{Z}) \leq \mathbb{P}(\{X \geq m\} \cap \mathcal{Z}) = \mathbb{E}[\mathbb{P}(X \geq m | F_{12}^*, \boldsymbol{\mathcal{E}}, \sigma^*, \boldsymbol{\pi^*}) \mathbf{1}_{\mathcal{Z}}]$$

$$\leq \mathbb{E}[(\inf_{\theta > 0} e^{-\theta m + \lambda_n(e^\theta - 1)}) \mathbf{1}_{\mathcal{Z}}] \leq \mathbb{E}[e \lambda_n^m \mathbf{1}_{\mathcal{Z}}] \leq n^{-m(s^2 \mathrm{T_c}(a,b) - \delta - o(1))}.$$

Above, the equality on the second line is due to the tower rule and since $\mathcal{Z}$ is measurable with respect to $|F_{12}^*|$, the inequality on the third line is due to a Chernoff bound; the inequality on the fourth line follows from setting $\theta = \log(1/\lambda_n)$ (which is valid since $\lambda_n = o(1)$ if $\mathcal{Z}$ holds). The final inequality uses the upper bound for $\lambda_n$ on $\mathcal{Z}$. Taking a union bound, we have

$$\mathbb{P}(\{\exists i \in [n], |N_{G^* \setminus (G_2 \wedge G_1)}(i) \cap F_{12}^*| \geq m\} \cap \mathcal{Z}) \leq n^{1 - m(s^2 \mathrm{T_c}(a,b) - \delta - o(1))}.$$

Here if we take $m > (s^2 \mathrm{T_c}(a, b))^{-1}$ and $\delta < s^2 \mathrm{T_c}(a, b) - m^{-1}$, the probability turns to $o(1)$. Thus, we can set $m = \lceil (s^2 \mathrm{T_c}(a, b))^{-1} \rceil + 1$. In light of Lemma E.6, $|F_{12}^*| \leq n^{1 - s^2 \mathrm{T_c}(a,b) + \delta}, \delta > 0$ w.h.p. Hence, the event $\mathcal{Z}$ happens with probability $1 - o(1)$. Hence we have

$$\mathbb{P}(\{\forall i \in [n], N_{G^* \setminus (G_2 \wedge G_1)}(i) \cap F_{12}^*| \leq \lceil (s^2 \mathrm{T_c}(a, b))^{-1} \rceil\}) = 1 - o(1).$$

By an identical proof, we have that

$$\mathbb{P}(\{\forall i \in [n], N_{G^* \setminus (G_2 \wedge G_3)}(i) \cap F_{23}^*| \leq \lceil (s^2 \mathrm{T_c}(a, b))^{-1} \rceil\}) = 1 - o(1).$$

Hence we have, with probability $1 - o(1)$, for $i \in M_{12}^* \cap M_{23}^*$,

$$|\mathsf{maj}_H(i) - \mathsf{maj}_{G_1 \vee_{\pi_{12}^*} G_2 \vee_{\pi_{13}^*} G_3}(i)| < \epsilon \log n,$$

and hence with probability $1 - o(1)$,

$$\mathsf{maj}_H(i) > \epsilon \log n.$$

Then by Lemma E.5, we can replace $H$ with $\widetilde{G}$, $F_{ij}^*$ with $F_{ij}$, the lemma follows. $\qquad \square$

Next, prove that each vertex in $G_2 \vee_{\pi_{23}^*} G_3 \setminus_{\pi_{12}^*} G_1$ has a small number of neighbors in $\pi_{12}^*(I_\epsilon(G_1))$

**Lemma F.4.** *If $0 < \epsilon \leq \frac{s \mathrm{D_+}(a,b)}{4 |\log(a/b)|}$, then*

$$\mathbb{P}(\forall i \in [n], |N_{G_2 \vee_{\pi_{23}^*} G_3 \setminus_{\pi_{12}^*} G_1}(i) \cap \pi_{12}^*(I_\epsilon(G_1))| \leq 2 \lceil (s \mathrm{D_+}(a, b))^{-1} \rceil) = 1 - o(1).$$

*Proof.* Since $I_\epsilon(G_1)$ depends on $G_1$ alone, it follows that $I_\epsilon(G_1)$ and $G_2 \vee_{\pi_{23}^*} G_3 \setminus_{\pi_{12}^*} G_1$ are conditionally independent given $\boldsymbol{\pi^*}, \sigma^*, \boldsymbol{\mathcal{E}}$. Hence we can stochastically dominate $|N_{G_2 \vee_{\pi_{23}^*} G_3 \setminus_{\pi_{12}^*} G_1}(i) \cap \pi_{12}^*(I_\epsilon(G_1))|$ by a Poisson random variable X with mean $\lambda_n$ given by

$$\lambda_n := \nu \log n / n |\{j \in I_\epsilon(G_1) : \{i, j\} \in \mathcal{E}_{011} \cup \mathcal{E}_{010} \cup \mathcal{E}_{001}\}| \leq \nu \log n / n |I_\epsilon(G_1)|.$$

Next, define the event $\mathcal{Z} := \{|I_\epsilon(G_1)| \leq n^{1 - s \mathrm{D_+}(a,b) + 2\epsilon |\log(a/b)|}\}$.

Notice that $P(\mathcal{Z}) = 1 - o(1)$ by Lemma D.7 and Markov's inequality, provided $s \mathrm{D_+}(a, b) < 99$. Following identical arguments as the proof of Lemma F.2, we arrive at

$$\mathbb{P}(\exists i \in [n], |N_{G_2 \vee_{\pi_{23}^*} G_3 \setminus_{\pi_{12}^*} G_1}(i) \cap \pi_{12}^*(I_\epsilon(G_1))| \geq m) = o(1),$$

when $m > \lceil (s \mathrm{D_+}(a, b) - 2\epsilon |\log a/b|)^{-1} \rceil$. If $\epsilon \leq \frac{s \mathrm{D_+}(a,b)}{4 |\log(a/b)|}$, it suffices to set $m = 2 \lceil (s \mathrm{D_+}(a, b))^{-1} \rceil + 1$. $\qquad \square$

**Lemma F.5.** *Suppose that $a, b, \epsilon > 0$ satisfy the following conditions:*

$$(1 - (1-s)^3)D_+(a,b) > 1 + 2\epsilon|\log a/b|, \qquad 0 < \epsilon \leq \frac{sD_+(a,b)}{4|\log a/b|}.$$

*With high probability, the algorithm correctly labels all vertices in $\{i \in [n] \setminus (F_{12}^* \cup F_{23}^*)\}$.*

*Proof.* Compare the neighborhood majority in $H$ corresponding to $\widehat{\sigma}_1$ with the true majority in $H$, where $H$ is defined in Lemma F.2:

$$\left|\sigma^*(i) \sum_{j \in N_H(i)} (\widehat{\sigma}_1(j) - \sigma^*(j))\right| \leq |N_H(i) \cap I_\epsilon(G_1)| \leq |N_{G^*}(i) \cap I_\epsilon(G_1)|$$

$$\leq |N_{G_2 \vee_{\pi_{23}^*} G_3 \setminus_{\pi_{12}^*} G_1}(i) \cap \pi_{12}^*(I_\epsilon(G_1))| + |N_{G_1}(i) \cap I_\epsilon(G_1)|$$

$$\leq 2\lceil D_+(a,b)^{-1}\rceil + 2\lceil(sD_+(a,b))^{-1}\rceil \leq \epsilon \log n/2.$$

The first inequality uses Lemma D.5 that the set of errors are contained in $I_\epsilon(G_1)$. The last inequality is due to Lemma D.8, F.4. Notice that $\mathsf{maj}_H(i) \geq \epsilon \log n$ for $i \in [n] \setminus (F_{12}^* \cup F_{23}^*)$. Hence, $\sigma^*(i) \sum_{j \in N_H(i)} \widehat{\sigma}_1(j) \geq \mathsf{maj}_H(i) - |\sigma^*(i) \sum_{j \in N_H(i)} (\widehat{\sigma}_1(j) - \sigma^*(j))| \geq \epsilon \log n/2 > 0$, which implies that the sign of neighborhood majorities are equal to the truth community label for any $i \in [n] \setminus (F_{12}^* \cup F_{23}^*)$, with probability $1 - o(1)$. Then we can convert $H$ to $\widetilde{G}\{[n] \setminus (F_{12} \cup F_{23})\}$, the vertices in $[n] \setminus (F_{12} \cup F_{23})$ are correctly labeled with probability $1 - o(1)$.

Using an identical proof, we can argue that the algorithm correctly label all vertices in $M_{13} \cap M_{32}$ and $M_{12} \cap M_{13}$. □

### F.3 Exact recovery in $[n] \setminus \{(M_{13} \cap M_{32}) \cup (M_{12} \cap M_{13}) \cup (M_{23} \cap M_{12})\}$

Define $M = (M_{13} \cap M_{32}) \cup (M_{12} \cap M_{13}) \cup (M_{23} \cap M_{12})$. Denote $F_b = (F_{12} \cap F_{13}) \cup (F_{12} \cap F_{23} \setminus F_{13}) \cup (F_{13} \cap F_{23} \setminus F_{12})$, note that $F_b = [n] \setminus M$.

**Lemma F.6.** *Suppose that $a, b, \epsilon > 0$ satisfy the following conditions:*

$$(1 - (1-s)^3)D_+(a,b) > 1 + 2\epsilon|\log a/b|, \qquad 0 < \epsilon \leq \frac{sD_+(a,b)}{4|\log a/b|},$$

$$s(1 - (1-s)^2)T_c(a,b) + s(1-s)^2D_+(a,b) > 1.$$

*With high probability, the algorithm correctly labels all vertices that are in $F_b$.*

*Proof.* For $i \in F_b$, define $H_i := (G_1 \setminus_{\pi_{13}^*} G_3 \setminus_{\pi_{12}^*} G_2)\{(M_{12} \cap M_{13}) \cup \{i\}\}$. Let $E_i$ be the event that i has a majority of at most $\epsilon' \log n$ in the graph $H_i$. Let $\widehat{\sigma}$ be the labeling after the step. For bervity, define a "nice" event based on the previous results. Define the event $\mathcal{H}$, which holds if and only if:

- $F_{ij} = F_{ij}^*$;

- $\widehat{H}_i = H_i$;

- $\widehat{\sigma}(i) = \sigma^*(i)$ for all $i \in M_{12} \cap M_{13}$;

- The event $\mathcal{F}$ holds;

- $|F_b| \leq n^{1-(2s^2-s^3)T_c(a,b)+\delta}$.

By Lemmas E.5, E.7, D.4, F.5, the event $\mathcal{H}$ holds with probability $1 - o(1)$. Furthermore, define $E_i^* := \mathsf{maj}_{H_i}(i) \leq \epsilon' \log n$, we have:

$$\mathbb{P}(\cup_{i \in [n]}(\{i \in F_b\} \cap E_i)) \leq \mathbb{P}((\cup_{i \in [n]}(\{i \in F_b^*\} \cap E_i^*)) \cap \mathcal{H}) + \mathbb{P}(\mathcal{H}^c)$$

$$\leq \sum_{i=1}^{n} \mathbb{P}(\{i \in F_b^*\} \cap E_i^* \cap \{F_b^* \leq n^{1-(2s^2-3s^3)T_c(a,b)+\delta}\} \cap \mathcal{F}) + o(1).$$

$$\tag{F.7}$$

By the tower rule, rewrite the term in the right hand side as:

$$\mathbb{E}[\mathbb{P}(E_i^*|\pi^*,\sigma^*,\mathcal{E},F_b^*)\mathbf{1}_{i\in F_b^*}\mathbf{1}_{\{|F_b^*|\le n^{1-(2s^2-s^3)\mathrm{T}_c(a,b)+\delta}\}\cap\mathcal{F}}]. \tag{F.8}$$

Now look at $\mathbb{P}(E_i^*|\pi^*,\sigma^*,\mathcal{E},F_b^*)$. Conditional on $\mathcal{E},\sigma^*,\pi^*,\mathsf{maj}_{H_i}(i):\overset{d}{=}Y-Z$, where $Y,Z$ are independent with:

$$Y\sim Bin(|j\in M_{12}^*\cup M_{13}^*:\{i,j\}\in\mathcal{E}_{100}\cap\mathcal{E}^+(\sigma^*)|,a\log n/n),$$

$$Z\sim Bin(|j\in M_{12}^*\cup M_{13}^*:\{i,j\}\in\mathcal{E}_{100}\cap\mathcal{E}^-(\sigma^*)|,b\log n/n).$$

By Definition D.3 of the event $\mathcal{F}$, we know that $|j\in M_{12}^*\cup M_{13}^*:\{i,j\}\in\mathcal{E}_{100}\cap\mathcal{E}^-(\sigma^*)|=(1-o(1))s(1-s)^2n/2$ and $|j\in M_{12}^*\cup M_{13}^*:\{i,j\}\in\mathcal{E}_{100}\cap\mathcal{E}^+(\sigma^*)|=(1-o(1))s(1-s)^2n/2$.

Lemma D.1 implies

$$\mathbb{P}(E_i^*|\pi^*,\sigma^*,\mathcal{E},F_b^*)\mathbf{1}_{i\in F_b^*}\mathbf{1}_{\{|F_b^*|\le n^{1-(2s^2-s^3)\mathrm{T}_c(a,b)+\delta}\}\cap\mathcal{F}}\le n^{-s(1-s)^2\mathrm{D}_+(a,b)+\epsilon'\log(a/b)/2+o(1)}.$$

Follow (F.8) and take a union bound, we have

$$\sum_{i=1}^n\mathbb{P}(\{i\in F_b^*\}\cap E_i^*\cap\{F_b^*\le n^{1-(2s^2-s^3)\mathrm{T}_c(a,b)+\delta}\}\cap\mathcal{F})+o(1)$$

$$\le n^{-s(1-s)^2\mathrm{D}_+(a,b)+\epsilon'\log(a/b)/2+o(1)}\mathbb{E}[|F_b^*|\mathbf{1}_{F_b^*\le n^{1-(2s^2-s^3)\mathrm{T}_c(a,b)+\delta}}]$$

$$\le n^{1-(2s^2-s^3)\mathrm{T}_c(a,b)-s(1-s)^2\mathrm{D}_+(a,b)+\epsilon'\log(a/b)/2+\delta+o(1)}.$$

Under the condition $(2s^2-s^3)\mathrm{T}_c(a,b)+s(1-s)^2\mathrm{D}_+(a,b)>1$, we can choose $\epsilon',\delta$ small enough so that the right hand side is $o(1)$. $\mathsf{maj}_{H_i}(i)>\epsilon'\log n$ for $i\in F_b^*$, by Lemma E.5, $\mathsf{maj}_{\widehat{H_i}}(i)>\epsilon'\log n$ for $i\in F_b$.

Suppose that $i\in F_{13}\cap F_{12}$, i has at most 12 neighbors in the graph $(G_1\wedge_{\pi_{12}^*}G_2)\{(M_{12}\cap M_{13})\cup\{i\}\}$, and in the graph $(G_1\wedge_{\pi_{13}^*}G_3)\{(M_{12}\cap M_{13})\cup\{i\}\}$. Therefore, i has an at least $(\epsilon'\log n-24)$ majority in $G_1\{(M_{12}\cap M_{13})\cup\{i\}\}$, with high probability. Then, Algorithm 3 correctly label all vertices in $F_{13}\cap F_{12}$.

Suppose that $i\in F_{12}\cap F_{23}\setminus F_{13}$, i has at most 12 neighbors in the graph $(G_1\wedge_{\pi_{12}^*}G_2)\{(M_{12}\cap M_{13})\cup\{i\}\}$ Therefore, i has an at least $(\epsilon'\log n-12)$ majority in $G_1\setminus_{\mu_{13}}G_3\{(M_{12}\cap M_{13})\cup\{i\}\}$, with high probability. Hence, Algorithm 3 correctly label all vertices in $F_{12}\cap F_{23}\setminus F_{13}$.

Suppose that $i\in F_{13}\cap F_{23}\setminus F_{12}$, i has at most 12 neighbors in the graph $(G_1\wedge_{\pi_{13}^*}G_3)\{(M_{12}\cap M_{13})\cup\{i\}\}$. Therefore, i has an at least $(\epsilon'\log n-12)$ majority in $G_1\setminus_{\mu_{12}}G_2\{(M_{12}\cap M_{13})\cup\{i\}\}$, with high probability. Algorithm 3 correctly label all vertices in $F_{13}\cap F_{23}\setminus F_{12}$. $\qquad\square$

# G Proof of impossibility for three graphs

In this section we prove that Theorem 2 when the exact community recovery is impossible. The impossibility under the condition $(1-(1-s)^3)\mathrm{D}_+(a,b)<1$ has been proved in [41]. Hence we focus on proving impossibility when

$$(2s^2-s^3)\mathrm{T}_c(a,b)+s(1-s)^2\mathrm{D}_+(a,b)<1. \tag{G.1}$$

To prove it, we study the MAP (maximum a posterior) estimator for the communities in $G_1$. Even with the additional information provided, including all the correct community labels in $G_2$, the true matching $\pi_{23}^*$ and most of the true matching $\pi_{12}^*$, the MAP estimator fails to exactly recovery communities with probability bounded away form 0 if the condition (G.1) holds. The proof is adapted from the MAP analysis in [22]. The difference is that we are considering three correlated SBM $G_1,G_2,G_3$. Since we know the true matching $\pi_{23}^*$, we can consider $H:=G_2\vee_{\pi_{23}^*}G_3\sim$ SBM$(n,(1-(1-s)^2)a\log n/n,(1-(1-s)^2)b\log n/n)$. Denote $R_{ij}$ the singleton in $G_i\wedge G_j$. Then $R=R_{12}\wedge R_{13}$ is the singleton set in $G_1\wedge H$.

## G.1  Notation

Here we review and introduce some notations in brief.
$\sigma_i^* :=$ the ground truth community labels in $G_i, i = 1, 2, 3$,
$V_i^+ := \{j \in [n] : \sigma_i^*(j) = +1\}, V_i^- := \{j \in [n] : \sigma_i^*(j) = -1\}$,
$\sigma_2^*(\pi_{12}^*(i)) = \sigma_1^*(i)$,
here we have $V_2^+ = \pi_{12}^*(V_1^+)$.

## G.2  The MAP estimator

First define the singleton set of a permutation $\pi$ with respect to the adjacency matrices $A$, $B$, and $C$ to be:
$$R(\pi, A, B, C) := \{i \in [n] : \forall j \in [n], A_{i,j} D_{\pi(i),\pi(j)} = 0\},$$
where $D_{ij} := \max\{B_{ij}, C_{\pi_{23}^*(i)\pi_{23}^*(j)}\}$. For brevity, write $R_\pi := R(\pi, A, B, C)$.

**Definition G.2.** *Define the set $S(\pi, A, B, C)$ as followings:*

1. *$i \in R(\pi, A, B, C)$;*

2. *$i$ is a singleton in $G_1\{R_\pi\}$;*

3. *If $j \in N_1(i)$, $\pi(j) \notin N_H(\pi(R_\pi))$.*

Where $A, B, C$ is the adjacency matrix of $G_1, G_2, G_3$ respectively and $D$ is the adjacency matrix in $H = G_2 \vee_{\pi_{23}^*} G_3$. Note that $D$ is the adjacency matrix of $H$, so $N_H(\pi(R_\pi)) = \{i \in [n] : \exists k \in \pi(R_\pi), D_{i,k} = 1\}$.

Define $\bar{R}_\pi := R_\pi \cup \pi^{-1}(N_H(\pi(R_\pi)))$. The condition 2 and 3 in Definition G.2 can be replaced by $A_{i,j} = 0$ for all $j \in \bar{R}_\pi$. Write $R^* = R(\pi_{12}^*, A, B, C)$, $S^* = S(\pi_{12}^*, A, B, C)$, and $\bar{R}^* = \bar{R}_{\pi_{12}^*}$ for brevity. We study the MAP estimate provided the additional knowledge $\sigma_2^*$, $\pi_{23}^*$, and $\pi_{12}^*\{[n] \setminus S^*\}$.

**Theorem 5.** *Let $A, B, C, \sigma_2^*, \pi_{23}^*, \pi_{12}^*\{[n] \setminus S^*\}, S^*$ be given. For $i \in [n] \setminus S^*, \widehat{\sigma}_{MAP}(i) = \sigma_2^*(\pi_{12}^*(i))$. For vertices in $S^*$, the MAP estimator depends on whether $a, b$ is larger:*
*1. If $a > b$, then the MAP estimator assigns the label +1 to the vertices corresponding to the largest $|S^* \cap V_1^+|$ values in the collection $\{\mathsf{maj}(i)\}_{i \in S^*}$ and assigns the label -1 to the remaining vertices in $S^*$.*
*2. If $a < b$, then the MAP estimator assigns the label +1 to the vertices corresponding to the smallest $|S^* \cap V_1^+|$ values in the collection $\{\mathsf{maj}(i)\}_{i \in S^*}$ and assigns the label -1 to the remaining vertices in $S^*$.*

Then the following corollary prove the potential failure of the MAP estimator.

**Corollary G.3.** *If $a > b$, there exists $i \in S^* \cap V_1^+, j \in S^* \cap V^-$ such that $\mathsf{maj}(i) < \mathsf{maj}(j)$, then the MAP estimator fails. Similarly, if $a < b$, there exists $i \in S^* \cap V_1^+, j \in S^* \cap V^-$ such that $\mathsf{maj}(i) > \mathsf{maj}(j)$, then the MAP estimator fails.*

*Proof.* Suppose $a > b$. If the MAP estimator classifies $i$ as $+1$ correctly. By Theorem 5 the MAP estimator classifies $j$ as $+1$ which is wrong. The argument for the case $a < b$ is similar. □

## G.3  The analysis of the failure of MAP estimator

**Definition G.4.** *The event $G_\delta$ holds if and only if*
$$n^{1-(2s^2-s^3)T_c(a,b)-\delta} \leq |R^* \cap V_1^+|, |R^* \cap V_1^-|, |\bar{R}^* \cap V_1^+|, |\bar{R}^* \cap V_1^+| \leq n^{1-(2s^2-s^3)T_c(a,b)+\delta}.$$

**Lemma G.5.** *For any fixed $\delta > 0$, $\mathbb{P}(G_\delta) = 1 - o(1)$.*

The proof of this lemma is straightforward but tedious and we defer it to Section G.6.

Now define the variable $W_i$:
$$W_i := \begin{cases} 1(i \in S^*, \mathsf{maj}(i) < 0), & i \in R^* \cap V_1^+, \\ 1(i \in S^*, \mathsf{maj}(i) > 0), & i \in R^* \cap V_1^-. \end{cases} \tag{G.6}$$

Denote $\mathcal{I}$ be the sigma algebra induced by the random variables

$$D = \max(B, C), \pi_{12}^*, \sigma_1^*, R^*, \{\mathcal{E}_{abc} : a, b, c \in \{0, 1\}\}.$$

Note that $\bar{R}^*, \bar{R}^* \cap V_1^+, \bar{R}^* \cap V_1^-$ are $\mathcal{I}$ measurable.

Now I'd like to show that $\sum_{i \in R^* \cap V_1^+} W_i > 0, \sum_{i \in R^* \cap V_1^-} W_i > 0$ with high probability, then it follows that $\exists i \in S^* \cap V_1^+, j \in S^* \cap V_1^-$ such that $\mathsf{maj}(i) < 0 < \mathsf{maj}(j)$. By Corollary G.3, the MAP estimator fails. Use the first and second method to analyze $\sum_{i \in R^* \cap V_1^+} W_i, \sum_{i \in R^* \cap V_1^-} W_i$:

**Lemma G.7.** *Fix $\delta > 0$ and denote $\theta := 1 - (2s^2 - s^3)\mathrm{T_c}(a, b) - s(1-s)^2 \mathrm{D_+}(a, b)$. We have*

$$\mathbb{E}[\sum_{i \in R^* \cap V_1^+} W_i | \mathcal{I}] 1(\mathcal{F} \cap \mathcal{G}_\delta) \geq (1 - n^{-(2s^2 - s^3)\mathrm{T_c}(a,b) + 2\delta}) n^{\theta - \delta - o(1)} 1(\mathcal{F} \cap \mathcal{G}_\delta)$$

*and*

$$\mathbb{E}[\sum_{i \in R^* \cap V_1^-} W_i | \mathcal{I}] 1(\mathcal{F} \cap \mathcal{G}_\delta) \geq (1 - n^{-(2s^2 - s^3)\mathrm{T_c}(a,b) + 2\delta}) n^{\theta - \delta - o(1)} 1(\mathcal{F} \cap \mathcal{G}_\delta).$$

**Lemma G.8.** *Fix $\delta > 0$ and denote $\theta := 1 - (2s^2 - s^3)\mathrm{T_c}(a, b) - s(1-s)^2 \mathrm{D_+}(a, b)$*

$$\mathrm{Var}(\sum_{i \in R^* \cap V_1^+} W_i | \mathcal{I}) 1(\mathcal{F} \cap \mathcal{G}_\delta) \leq n^{2\theta - 3\delta} 1(\mathcal{F} \cap \mathcal{G}_\delta)$$

*and*

$$\mathrm{Var}(\sum_{i \in R^* \cap V_1^-} W_i | \mathcal{I}) 1(\mathcal{F} \cap \mathcal{G}_\delta) \leq n^{2\theta - 3\delta} 1(\mathcal{F} \cap \mathcal{G}_\delta).$$

The proofs of these two lemmas are deferred to Section G.7. Using the lemmas above we can now prove Theorem 5.

*Proof of Theorem 2 when $K = 3$.* Firstly, show that $\sum_{i \in R^* \cap V_1^+} W_i > 0$ with high probability. Use the second moment method, we obtain

$$\mathbb{P}(\sum_{i \in R^* \cap V_1^+} W_i > 0 | \mathcal{I}) \geq \frac{\mathbb{E}[\sum_{i \in R^* \cap V_1^+} W_i | \mathcal{I}]^2}{\mathbb{E}[(\sum_{i \in R^* \cap V_1^+} W_i)^2 | \mathcal{I}]}$$

$$= \frac{\mathbb{E}[\sum_{i \in R^* \cap V_1^+} W_i | \mathcal{I}]^2}{\mathbb{E}[\sum_{i \in R^* \cap V_1^+} W_i | \mathcal{I}]^2 + \mathrm{Var}(\sum_{i \in R^* \cap V_1^+} W_i | \mathcal{I})}$$

$$\geq 1 - \frac{\mathrm{Var}(\sum_{i \in R^* \cap V_1^+} W_i | \mathcal{I})}{\mathbb{E}[\sum_{i \in R^* \cap V_1^+} W_i | \mathcal{I}]^2}.$$

Hence for unconditional probability, let $\delta$ small enough, $\delta < \min((2s^2 - s^3)\mathrm{T_c}(a, b)/8, \theta/4)$, then

$$\mathbb{P}(\sum_{i \in R^* \cap V_1^+} W_i > 0) \geq \mathbb{P}(\{\sum_{i \in R^* \cap V_1^+} W_i > 0\} \cap \mathcal{F} \cap \mathcal{G}_\delta) = \mathbb{E}[\mathbb{P}(\sum_{i \in R^* \cap V_1^+} W_i > 0 | \mathcal{I}) 1(\mathcal{F} \cap \mathcal{G}_\delta)]$$

$$\geq \mathbb{E}\left[\left(1 - \frac{\mathrm{Var}(\sum_{i \in R^* \cap V_1^+} W_i | \mathcal{I})}{\mathbb{E}[\sum_{i \in R^* \cap V_1^+} W_i | \mathcal{I}]^2}\right) 1(\mathcal{F} \cap \mathcal{G}_\delta)\right]$$

$$\geq \left((1 - o(1)) n^{2\theta - 3\delta - 2(\theta - \delta) + o(1)}\right) \mathbb{P}(\mathcal{F} \cap \mathcal{G}_\delta) = 1 - o(1).$$

The inequality on the last line is by Lemma G.7, G.8. The equality in the last line is by Lemma G.5, D.4. The proof for $\mathbb{P}(\sum_{i \in R^* \cap V_1^-} W_i > 0) = 1 - o(1)$ is identical. Hence, in light of Corollary G.3, the MAP estimator fails with probability $1 - o(1)$. $\qquad\square$

### G.4 Analysis of $S_\pi$

In this section we introduce the set $\mathcal{A}_\pi$ and some properties of $S_\pi$.

First, define the set

$$\mathcal{A}(S^*, \pi_{12}^*\{[n] \setminus S^*\}) := \{\pi \in S_n : S_\pi = S^*, \pi([n] \setminus S_\pi) = \pi_{12}^*([n] \setminus S^*)\}.$$

For brevity, sometimes write $\mathcal{A}^*$.

**Lemma G.9.** *For any $\pi \in \mathcal{A}^*$, $A_{i,j} B_{\pi(i),\pi(j)} C_{\pi_{23}^*(\pi(i)),\pi_{23}^*(\pi(j))} = A_{i,j} B_{\pi_{12}^*(i)\pi_{12}^*(j)} C_{\pi_{13}^*(i),\pi_{13}^*(j)}$. Moreover, if $i \in S^*$ or $j \in S^*$, $A_{i,j} B_{\pi(i),\pi(j)} C_{\pi_{23}^*(\pi(i)),\pi_{23}^*(\pi(j))} = A_{i,j} B_{\pi_{12}^*(i)\pi_{12}^*(j)} C_{\pi_{13}^*(i),\pi_{13}^*(j)} = 0, A_{i,j} C_{\pi_{23}^*(\pi(i)),\pi_{23}^*(\pi(j))} = A_{i,j} C_{\pi_{13}^*(i),\pi_{13}^*(j)} = 0, A_{i,j} B_{\pi(i),\pi(j)} = A_{i,j} B_{\pi_{12}^*(i)\pi_{12}^*(j)} = 0.$*

*Proof.* If $i, j \in [n] \setminus S^*$, then $\pi(i) = \pi_{12}^*(i)$ and $\pi(j) = \pi_{12}^*(j)$, hence $A_{i,j} B_{\pi(i),\pi(j)} C_{\pi_{23}^*(\pi(i)),\pi_{23}^*(\pi(j))} = A_{i,j} B_{\pi_{12}^*(i)\pi_{12}^*(j)} C_{\pi_{13}^*(i),\pi_{13}^*(j)}$. If $i \in S^*$ or $j \in S^*$, then by Definition G.2, $i \in R^*$ or $j \in R^*$, hence $A_{i,j} B_{\pi(i),\pi(j)} C_{\pi_{23}^*(\pi(i)),\pi_{23}^*(\pi(j))} = A_{i,j} B_{\pi_{12}^*(i)\pi_{12}^*(j)} C_{\pi_{13}^*(i),\pi_{13}^*(j)} = 0$, $A_{i,j} C_{\pi_{23}^*(\pi(i)),\pi_{23}^*(\pi(j))} = A_{i,j} C_{\pi_{13}^*(i),\pi_{13}^*(j)} = 0$, and $A_{i,j} B_{\pi(i),\pi(j)} = A_{i,j} B_{\pi_{12}^*(i)\pi_{12}^*(j)} = 0$. $\quad\square$

**Definition G.10.** *Let $\rho$ be a permutation of $S^*$ The permutation $P_{\pi,\rho}$ is given by*

$$P_{\pi,\rho} := \begin{cases} \pi(i) & i \in [n] \setminus S^*, \\ \pi(\rho(i)) & i \in S^*. \end{cases}$$

**Lemma G.11.** *Let $\rho$ be a permutation of $S^*$. Then $P_{\pi,\rho} \in \mathcal{A}^*$.*

*Proof.* The proof is identical to [22, Lemma 8.11]]. We only need to change $B$ to $B' = \max(B, C)$ in the argument. $\quad\square$

A useful corollary of Lemma G.11 is that the elements of $\mathcal{A}^*$ can be described by permutation of S.

**Corollary G.12.** *We have the following representation*

$$\mathcal{A}^* = \{P_{\pi^*,\rho} : \rho \text{ is a permutation of } S^*\}.$$

### G.5 Deriving the MAP estimator, proof of Theorem 5

#### G.5.1 The posterior distribution of $\pi_{12}^*$

First we define

$$\mu^+(\pi)_{abc} := \sum_{(\pi(i),\pi(j)) \in \mathcal{E}^+(\sigma_2^*)} \mathbb{1}((A_{i,j}, B_{\pi(i),\pi(j)}, C_{\pi_{23}^*(\pi(i)),\pi_{23}^*(\pi(j))}) = (a,b,c)), a,b,c \in \{0,1\},$$

$$\mu^-(\pi)_{abc} := \sum_{(\pi(i),\pi(j)) \in \mathcal{E}^-(\sigma_2^*)} \mathbb{1}((A_{i,j}, B_{\pi(i),\pi(j)}, C_{\pi_{23}^*(\pi(i)),\pi_{23}^*(\pi(j))}) = (a,b,c)), a,b,c \in \{0,1\},$$

$$\nu^+(\pi) := \sum_{(\pi(i),\pi(j)) \in \mathcal{E}^+(\sigma_2^*)} A_{i,j},$$

$$\nu^-(\pi) := \sum_{(\pi(i),\pi(j)) \in \mathcal{E}^-(\sigma_2^*)} A_{i,j}.$$

With these definitions, we can derive an exact expression for the posterior distribution of $\pi_{12}^*$ given $A, B, C, \sigma_2^*$.

**Lemma G.13.** *Let $\pi \in S_n$. There's a constant $D_1 = D_1(A,B,C,\sigma_2^*,\pi_{23}^*)$ such that*

$$\mathbb{P}(\pi_{12}^* = \pi | A, B, C, \sigma_2^*, \pi_{23}^*)$$

$$= D_1 \left(\frac{p_{111}p_{000}}{p_{011}p_{100}}\right)^{\mu^+(\pi)_{111}} \left(\frac{p_{100}}{p_{000}}\right)^{\nu^+(\pi)} \left(\frac{p_{000}p_{110}}{p_{100}p_{010}}\right)^{\mu^+(\pi)_{110}} \left(\frac{p_{000}p_{101}}{p_{001}p_{100}}\right)^{\mu^+(\pi)_{101}}$$

$$\times \left(\frac{q_{111}q_{000}}{q_{011}q_{100}}\right)^{\mu^-(\pi)_{111}} \left(\frac{q_{100}}{q_{000}}\right)^{\nu^-(\pi)} \left(\frac{q_{000}q_{110}}{q_{100}q_{010}}\right)^{\mu^-(\pi)_{110}} \left(\frac{q_{000}q_{101}}{q_{001}q_{100}}\right)^{\mu^-(\pi)_{101}}.$$

*Proof.* The proof is adapted from [22, Lemma 8.13]. By Bayes Rule,

$$\mathbb{P}(\pi_{12}^* = \pi | A, B, C, \sigma_2^*, \pi_{23}^*) = \frac{\mathbb{P}(A, B, C | \sigma_2^*, \pi_{12}^* = \pi, \pi_{23}^*)\mathbb{P}(\pi_{12}^* = \pi | \sigma_2^*, \pi_{23}^*)}{\mathbb{P}(A, B, C | \sigma_2^*, \pi_{23}^*)}.$$

In the construction of the multiple Correlated SBM, the permutation $\pi_{12}^*$ is chosen independently of everything else, including the community labeling $\sigma_2^*$ and the permutation $\pi_{23}^*$. Hence we can rewrite

$$\mathbb{P}(\pi_{12}^* = \pi | A, B, C, \sigma_2^*, \pi_{23}^*) = d_1(A, B, C, \sigma_2^*, \pi_{23}^*)\mathbb{P}(A, B, C | \sigma_2^*, \pi_{23}^*, \pi_{12}^* = \pi),$$

where $d_1(A, B, C, \sigma_2^*) = (n!\mathbb{P}(A, B, C | \sigma_2^*, \pi_{23}^*))^{-1}$. Look at $\mathbb{P}(A, B, C | \sigma_2^*, \pi^* = \pi, \pi_{23}^*)$, note that the edge formation process in $G_1, G_2$ and $G_3$ is mutually independent across all vertex pairs given $\sigma_2^*, \pi_{12}^*, \pi_{23}^*$. Hence we have

$$\mathbb{P}(A, B, C | \sigma_2^*, \pi^* = \pi, \pi_{23}^*) = \prod_{ijk \in \{0,1\}} p_{ijk}^{\mu^+(\pi)_{ijk}} q_{ijk}^{\mu^-(\pi)_{ijk}}.$$

In particular, the sums $\sum_{(\pi(i),\pi(j))\in\mathcal{E}^+(\sigma_2^*)} B_{\pi(i),\pi(j)}C_{\pi_{23}^*\pi(i),\pi_{23}^*\pi(j)}$, $\sum_{(\pi(i),\pi(j))\in\mathcal{E}^+(\sigma_2^*)} B_{\pi(i),\pi(j)}$, and $\sum_{(\pi(i),\pi(j))\in\mathcal{E}^+(\sigma_2^*)} C_{\pi_{23}^*\pi(i),\pi_{23}^*\pi(j)}, |\mathcal{E}^+(\sigma_2^*)|$ are measurable with respect to $B, C, \sigma_2^*, \pi_{23}^*$. Hence we do not care the relevant value and use $\Lambda$ to represent. Now, for simple notations we write $\sum ABC$ to represent $\sum_{(\pi(i),\pi(j))\in\mathcal{E}^+(\sigma_2^*)} A_{i,j}B_{\pi(i),\pi(j)}C_{\pi_{23}^*(\pi(i)),\pi_{23}^*(\pi(j))}$, $\sum AB$ to represent $\sum_{(\pi(i),\pi(j))\in\mathcal{E}^+(\sigma_2^*)} A_{i,j}B_{\pi(i),\pi(j)}$, and $\sum AC$ to represent $\sum_{(\pi(i),\pi(j))\in\mathcal{E}^+(\sigma_2^*)} A_{i,j}C_{\pi_{23}^*(\pi(i)),\pi_{23}^*(\pi(j))}$. Then, we can write

$$\mu^+(\pi)_{011} = \sum(1-A)BC = \sum BC - \mu^+(\pi)_{111} = \Lambda - \mu^+(\pi)_{111},$$

$$\mu^+(\pi)_{010} = \sum(1-A)B(1-C) = \Lambda - \mu^+(\pi)_{110},$$

$$\mu^+(\pi)_{001} = \sum(1-A)(1-B)C = \Lambda - \mu^+(\pi)_{101},$$

$$\mu^+(\pi)_{000} = \sum(1-A)(1-B)(1-C) = \Lambda - \mu^+(\pi)_{100} + \mu^+(\pi)_{101} + \mu^+(\pi)_{110} + \mu^+(\pi)_{111},$$

$$\mu^+(\pi)_{100} = \sum A(1-B)(1-C) = \Lambda - \mu^+(\pi)_{111} - \mu^+(\pi)_{101} - \mu^+(\pi)_{110} + \nu^+(\pi).$$

Hence we can write

$$\prod_{ijk \in \{0,1\}} p_{ijk}^{\mu^+(\pi)_{ijk}} = d_2^+ \left(\frac{p_{111}p_{000}}{p_{011}p_{100}}\right)^{\mu^+(\pi)_{111}} \left(\frac{p_{100}}{p_{000}}\right)^{\nu^+(\pi)} \left(\frac{p_{000}p_{110}}{p_{100}p_{010}}\right)^{\mu^+(\pi)_{110}} \left(\frac{p_{000}p_{101}}{p_{001}p_{100}}\right)^{\mu^+(\pi)_{101}}.$$

Here $d_2^+$ is some constant given the information $B, C, \sigma_2^*, \pi_{23}^*$ Replicating the arguments for $\mu^-(\pi)_{abc}$, we have that

$$\prod_{ijk \in \{0,1\}} q_{ijk}^{\mu^-(\pi)_{ijk}} = d_2^- \left(\frac{q_{111}q_{000}}{q_{011}q_{100}}\right)^{\mu^-(\pi)_{111}} \left(\frac{q_{100}}{q_{000}}\right)^{\nu^-(\pi)} \left(\frac{q_{000}q_{110}}{q_{100}q_{010}}\right)^{\mu^-(\pi)_{110}} \left(\frac{q_{000}q_{101}}{q_{001}q_{100}}\right)^{\mu^-(\pi)_{101}}.$$

Combining the two equations we prove the statement of lemma with $D_1 = d_1 d_2^+ d_2^-$. $\square$

**Lemma G.14.** *There is a constant $D_2 = D_2(A, B, C, \sigma_2^*, S^*, \pi_{23}^*, \pi_{12}^*\{[n] \setminus S^*\})$ such that*

$$\mathbb{P}(\pi_{12}^* = \pi | A, B, C, \sigma_2^*, \pi_{23}^*, S^*, \pi_{12}^*\{[n] \setminus S^*\}) = D_2(\sqrt{\frac{p_{100}q_{000}}{p_{000}q_{100}}})^{\nu^+\pi-\nu^-(\pi)} 1(\pi \in \mathcal{A}^*).$$

*Proof.* By Bayes Rule we have that

$$\mathbb{P}(\pi_{12}^* = \pi | A, B, C, \sigma_2^*, \pi_{23}^*, S^*, \pi_{12}^*\{[n] \setminus S^*\})$$

$$= \frac{\mathbb{P}(\pi_{12}^* = \pi | A, B, C, \sigma_2^*, \pi_{23}^*)\mathbb{P}(S^*, \pi_{12}^*\{[n] \setminus S^*\} | \pi_{12}^* = \pi, A, B, C, \sigma_2^*, \pi_{23}^*)}{\mathbb{P}(S^*, \pi_{12}^*\{[n] \setminus S^*\} | A, B, C, \sigma_2^*, \pi_{23}^*)}$$

$$= \frac{\mathbb{P}(\pi_{12}^* = \pi | A, B, C, \sigma_2^*, \pi_{23}^*)}{\mathbb{P}(S^*, \pi_{12}^*\{[n] \setminus S^*\} | A, B, C, \sigma_2^*, \pi_{23}^*)} 1(\pi \in \mathcal{A}^*).$$

The probability in the denominator is a function of $A, B, C, S^*, \pi^*\{[n] \setminus S^*\}, \pi_{23}^*, \sigma_2^*$. Furthermore, by Lemma G.9, $\mu^+(\pi)_{111}, \mu^+(\pi)_{110}, \mu^+(\pi)_{101}, \mu^-(\pi)_{111}, \mu^-(\pi)_{110}, \mu^-(\pi)_{101}$ are constant over $\pi \in \mathcal{A}^*$. By Lemma G.13, we can write

$$\mathbb{P}(\pi_{12}^* = \pi | A, B, C, \sigma_2^*, \pi_{23}^*, S^*, \pi_{12}^*\{[n] \setminus S^*\}) = d_2 \left(\frac{p_{100}}{p_{000}}\right)^{\nu^+(\pi)} \left(\frac{q_{100}}{q_{000}}\right)^{\nu^-(\pi)} 1(\pi \in \mathcal{A}^*),$$

where

$$d_2 = \frac{D_1}{\mathbb{P}(S^*, \pi_{12}^*\{[n] \setminus S^*\}|A, B, C, \sigma_2^*, \pi_{23}^*)} \left(\frac{q_{111}q_{000}}{q_{011}q_{100}}\right)^{\mu^-(\pi_{12}^*)_{111}} \left(\frac{q_{000}q_{110}}{q_{100}q_{010}}\right)^{\mu^-(\pi_{12}^*)_{110}}$$

$$\times \left(\frac{q_{000}q_{101}}{q_{001}q_{100}}\right)^{\mu^-(\pi_{12}^*)_{101}} \left(\frac{p_{111}p_{000}}{p_{011}p_{100}}\right)^{\mu^+(\pi)_{111}} \left(\frac{p_{000}p_{110}}{p_{100}p_{010}}\right)^{\mu^+(\pi)_{110}} \left(\frac{p_{000}p_{101}}{p_{001}p_{100}}\right)^{\mu^+(\pi)_{101}}.$$

$D_1$ is the same constant in Lemma G.13, $d_2$ is a constant given $A, B, C, \sigma_2^*, S^*, \pi_{23}^*, \pi_{12}^*\{[n] \setminus S^*\}$. To further simplifies the posterior distribution,

$$\mathbb{P}(\pi_{12}^* = \pi | A, B, C, \sigma_2^*, \pi_{23}^*, S^*, \pi_{12}^*\{[n] \setminus S^*\}) = d_2 \left(\frac{p_{100}}{p_{000}}\right)^{\nu^+(\pi)} \left(\frac{q_{100}}{q_{000}}\right)^{\nu^-(\pi)} 1(\pi \in \mathcal{A}^*)$$

$$= d_2 (\sqrt{\frac{p_{100}q_{100}}{p_{000}q_{100}}})^{\nu^-(\pi)+\nu^+(\pi)} (\sqrt{\frac{p_{100}q_{100}}{p_{000}q_{100}}})^{\nu^+(\pi)-\nu^-(\pi)} 1(\pi \in \mathcal{A}^*).$$

Note that $\nu^-(\pi) + \nu^+(\pi) = \sum_{(i,j)\in\binom{[n]}{2}} A_{i,j}$ that only depends on $A$. The results follows then

$$\mathbb{P}(\pi_{12}^* = \pi | A, B, C, \sigma_2^*, \pi_{23}^*, S^*, \pi_{12}^*\{[n] \setminus S^*\}) = D_2 (\sqrt{\frac{p_{100}q_{100}}{p_{000}q_{100}}})^{\nu^+(\pi)-\nu^-(\pi)} 1(\pi \in \mathcal{A}^*),$$

where $D_2 = d_2 (\sqrt{\frac{p_{100}q_{100}}{p_{000}q_{100}}})^{\sum_{(i,j)\in\binom{[n]}{2}} A_{i,j}}.$ $\qquad\square$

### G.5.2  The posterior distribution of $\sigma^*$

Now we can study the posterior distribution of the community labeling $\sigma^*$. For a community partition $X = (X^+, X^-)$ of $[n]$ in $G_1$, define the set

$$B(X) := \{\pi \in \mathcal{A}^* : \pi(X^+) = V_2^+, \pi(X^-) = V_2^-\}.$$

In particular, if $\sigma_X$ denotes the community memberships associated with $X$, the following must hold:

- $\sigma_X(i) = \sigma_2^*(\pi_{12}^*(i))$ for $i \in [n] \setminus S^*$;
- $|S^* \cap X^+| = |S^* \cap V_1^+|, |S^* \cap X^-| = |S^* \cap V_1^-|$.

The first condition must hold since we know the true vertex correspondence and the true community labels outside of the $S^*$. The second condition must hold since the number of vertices of each community in $S^*$ can be deduced by examining the community labels of $\pi^*(S^*)$ with respect to $\sigma_2^*$.

**Lemma G.15.** *If $|B(X)|$ is not empty, then $|B(X)| = |S^* \cap X^-|!|S^* \cap X^+|!$.*

*Proof.* The proof is almost identical to [22, Lemma 8.15]. Suppose that $\pi_0, \pi_1 \in B(X)$, by Corollary G.12, there exists $\rho$ such that $\pi_1 = P_{\pi_0,\rho}$. Claim: if $i \in S^* \cap X^+, \rho(i) \in S^* \cap X^+$, if $i \in S^* \cap X^-, \rho(i) \in S^* \cap X^-$. If $\exists i \in S^* \cap X^+, \rho(i) \in S^* \cap X^-$, then $\sigma_2^*(\pi_0(i)) = 1, \sigma_2^*(\pi_1(i)) = \sigma_2^*(\pi_0(\rho(i))) = -1$. This violates the definition of $B(X)$. The claim is proved. Hence we can decomposition $\rho$ into two disjoint permutations $\rho^+, \rho^-.\rho^+$ is a permutation of $S^* \cap X^+$ while $\rho^-$ is a permutation of $S^* \cap X^-$. Hence $|B(X)| = (\#$ of choices of $\rho^+) \times (\#$ of choices of $\rho^-) = |S^* \cap X^+|!|S^* \cap X^-|! = |S^* \cap V_1^+|!|S^* \cap V^-|!.$ $\qquad\square$

Then we look at $\nu^+(\pi) - \nu^-(\pi)$.

**Lemma G.16.** *For all $\pi \in B(X)$, we have that $\nu^+(\pi) - \nu^-(\pi) = D_3 + \sum_{i \in S^*} \mathsf{maj}(i)\sigma_X(i)$ where $D_3$ is a constant depending on $A, B, C, \sigma_2^*, \pi_{23}^*, S^*, \pi_{12}^*\{[n] \setminus S^*\}$ but not on $X$.*

*Proof.* Note that $\sigma_X(i)\sigma_X(j) = 1$ if $(\pi(i), \pi(j)) \in \mathcal{E}^+(\sigma_2^*)$, $\sigma_X(i)\sigma_X(j) = -1$ if $(\pi(i), \pi(j)) \in \mathcal{E}^-(\sigma_2^*)$. We have

$$\nu^+(\pi) - \nu^-(\pi) = \sum_{(i,j)\in\binom{[n]}{2}} \sigma_X(i)\sigma_X(j)A_{i,j}$$

$$= \sum_{(i,j)\in\{[n]\setminus S^*\}} \sigma_X(i)\sigma_X(j)A_{i,j} + \sum_{i,j\in S^*} \sigma_X(i)\sigma_X(j)A_{i,j} + \sum_{i\in S^*, j\in\{[n]\setminus S^*\}} \sigma_X(i)\sigma_X(j)A_{i,j}.$$

We should note that if $(i, j) \in \{[n] \setminus S^*\}$, then $\sigma_X(i)\sigma_X(j)A_{i,j} = \sigma_2^*(\pi_{12}(i))\sigma_2^*(\pi_{12}(j))A_{i,j} = \sigma_2^*(\pi_{12}^*(i))\sigma_2^*(\pi_{12}^*(j))A_{i,j}$. Denote

$$D_3 := \sum_{(i,j)\in\{[n]\setminus S^*\}} \sigma_X(i)\sigma_X(j)A_{i,j}.$$

Clearly $D_3$ depends only on $A, \sigma_2^*, S^*, \pi_{12}^*\{[n] \setminus S^*\}$. If $i, j \in S^*$, $A_{i,j} = 0$ by Definition G.2.

$$\sum_{i\in S^*, j\in\{[n]\setminus S^*\}} \sigma_X(i)\sigma_X(j)A_{i,j} = \sum_{i\in S^*, j\in\{[n]\setminus S^*\}} \sigma_X(i)\sigma_1^*(j)A_{i,j}$$

$$= \sum_{i\in S^*} \sigma_X(i) \sum_{j\in\{[n]\setminus S^*\}} \sigma_1^*(j)A_{i,j}$$

$$= \sum_{i\in S^*} \sigma_X(i)\mathsf{maj}(i).$$

The last equality is because if $i \in S^*$, $\mathsf{maj}(i) = \sum_{j\in[n]} A_{i,j}\sigma_1^*(j) = \sum_{j\in\{[n]\setminus S^*\}} A_{i,j}\sigma_1^*(j)$, since $A_{i,j} = 0$ if $i, j \in S^*$. Then the statement follows,

$$\nu^+(\pi) - \nu^-(\pi) = D_3 + \sum_{i\in S^*} \sigma_X(i)\mathsf{maj}(i). \qquad \square$$

**Lemma G.17.** *If $B(X)$ is nonempty, then*

$$\mathbb{P}((V_1^+, V_2^+) = (X^+, X^-)|A, B, C, \sigma_2^*, \pi_{23}^*, S^*, \pi_{12}^*\{[n] \setminus S^*\})$$

$$= D_4 \left(\sqrt{\frac{p_{100}q_{100}}{p_{000}q_{000}}}\right)^{\sum_{i\in S^*} \sigma_X(i)\mathsf{maj}(i)},$$

*where $D_4$ is a constant depending on $A, B, C, \sigma_2^*, \pi_{23}^*, S^*, \pi_{12}^*\{[n] \setminus S^*\}$, but not on $X$.*

*Proof.* We have that

$$\mathbb{P}((V_1^+, V_2^+) = (X^+, X^-)|A, B, C, \sigma_2^*, \pi_{23}^*, S^*, \pi_{12}^*\{[n] \setminus S^*\})$$

$$= \sum_{\pi\in B(X)} \mathbb{P}(\pi_{12}^* = \pi|A, B, C, \sigma_2^*, \pi_{23}^*, S^*, \pi_{12}^*\{[n] \setminus S^*\})$$

$$= \sum_{\pi\in B(X)} D_2 \left(\sqrt{\frac{p_{100}q_{000}}{p_{000}q_{100}}}\right)^{\nu^+(\pi)-\nu^-(\pi)}$$

$$= D_2 \left(\sqrt{\frac{p_{100}q_{000}}{p_{000}q_{100}}}\right)^{D_3} \sum_{\pi\in B(X)} \left(\sqrt{\frac{p_{100}q_{000}}{p_{000}q_{100}}}\right)^{\sum_{i\in S^*} \sigma_X(i)\mathsf{maj}(i)}$$

$$= D_2 \left(\sqrt{\frac{p_{100}q_{000}}{p_{000}q_{100}}}\right)^{D_3} |B(X)| \left(\sqrt{\frac{p_{100}q_{000}}{p_{000}q_{100}}}\right)^{\sum_{i\in S^*} \sigma_X(i)\mathsf{maj}(i)}$$

$$= D_4 \left(\sqrt{\frac{p_{100}q_{000}}{p_{000}q_{100}}}\right)^{\sum_{i\in S^*} \sigma_X(i)\mathsf{maj}(i)},$$

where $D_4 = D_2 \left(\sqrt{\frac{p_{100}q_{000}}{p_{000}q_{100}}}\right)^{D_3} |S^* \cap V_1^+||S^* \cap V_1^-|$. The statement follows. $\qquad \square$

*Proof of Theorem 5.* Given $A, B, C, \sigma_2^*, \pi_{23}^*, S^*, \pi_{12}^*\{[n] \setminus S^*\}$, it's obivious that $\widehat{\sigma}_{MAP}(i) = \sigma_2^*(\pi_{12}^*(i))$ for $i \in [n] \setminus S^*$. For vertices in $S^*$, note that

$$\frac{p_{100}q_{000}}{p_{000}q_{100}} = \frac{s(1-s)^2 p(1-(1-(1-s)^3)q)}{s(1-s)^2 q(1-(1-(1-s)^3)q)} = (1+o(1))\frac{p}{q} = (1+o(1))\frac{a}{b}.$$

Thus, by Lemma G.17, if $a > b$, the MAP estimator maximizes $\sum_{i \in S^*} \sigma_X(i)\mathsf{maj}(i)$ while the MAP estimator minimizes $\sum_{i \in S^*} \sigma_X(i)\mathsf{maj}(i)$ if $a < b$. Suppose $a > b$, while satisfying the condition $|S^* \cap X^+| = |S^* \cap V^+|, |S^* \cap X^-| = |S^* \cap V^-|$, the maximum of $\sum_{i \in S^*} \sigma_X(i)\mathsf{maj}(i)$ is obtained by seting $\sigma_X(i) = +1$ to the vertices $i \in S^*$ corresponding to the largest $|S^* \cap V_1^+|$ values in the collection $\{\mathsf{maj}(i)\}_{i \in S^*}$ and assigns the label -1 to the remaining vertices in $S^*$. The proof is the same suppose $a < b$. Then Theorem 5 follows. $\qquad\square$

## G.6  Proof of Lemma G.5

Denote $E_i$ as the event that $i$ is a singleton in $G_1 \wedge_{\pi_{12}^*} H$, in other words that $i \in R^*$. We assume that the communities are approximately balanced. More precisely, we assume that the event $\mathcal{G} = \{n/2 - n^{3/4} \leq |V^+|, |V^-| \leq n/2 + n^{3/4}\}$ holds. By Lemma D.4, we have that $\mathbb{P}(\mathcal{G}) = 1 - o(1)$.

Conditioning on $\sigma_1^*$, if $i \in V_1^+$, then we have

$$\mathbb{P}(E_i|\sigma_1^*)1(\mathcal{G}) = (1 - s(1 - (1-s)^2)a\log n/n)^{|V_1^+|-1}(1 - s(1-(1-s)^2)b\log n/n)^{|V_1^-|}1(\mathcal{G})$$
$$\leq \exp(-s(2s-s^2)\log n/n(a(|V_1^+|-1) + b|V_1^-|))1(\mathcal{G})$$
$$= \exp(-(1-o(1))(2s^2 - s^3)\mathrm{T_c}(a,b)\log n)1(\mathcal{G}) = n^{-(2s^2-s^3)\mathrm{T_c}(a,b)+o(1)}1(\mathcal{G}).$$

The first inequality uses the fact $1 - x \leq e^{-x}$. Hence

$$\mathbb{E}[|R^* \cap V_1^+||\sigma_1^*]1(\mathcal{G}) = \sum_{i \in V^+} \mathbb{P}(E_i|\sigma_1^*)1(\mathcal{G}) \leq |V_1^+|n^{-(2s^2-s^3)\mathrm{T_c}(a,b)+o(1)}1(\mathcal{G})$$
$$\leq n^{1-(2s^2-s^3)\mathrm{T_c}(a,b)+o(1)}.$$

By Markov's inequality,

$$\mathbb{P}(|R^* \cap V_1^+| \geq n^{1-(2s^2-s^3)\mathrm{T_c}(a,b)+\delta}|\sigma_1^*)1(\mathcal{G}) \leq n^{-\delta+o(1)} = o(1).$$

Hence

$$\mathbb{P}(|R^* \cap V_1^+| \geq n^{1-(2s^2-s^3)\mathrm{T_c}(a,b)+\delta})$$
$$\leq \mathbb{P}(\mathcal{G}^c) + \mathbb{E}[\mathbb{P}(|R^* \cap V_1^+| \geq n^{1-(2s^2-s^3)\mathrm{T_c}(a,b)+\delta}|\sigma_1^*)1(\mathcal{G})] = o(1).$$

Now we derive a lower bound for $|R^* \cap V_1^+|$. For $\epsilon = \epsilon_n$ sufficiently small:

$$(\log(\mathbb{P}(E_i|\sigma_1^*)))1(\mathcal{G}) = ((|V_1^+|-1)\log(1 - s(1-(1-s)^2)a\log n/n)$$
$$+ |V_1^-|\log(1 - s(1-(1-s)^2)b\log n/n))1(\mathcal{G})$$
$$\geq -(1+\epsilon)(2s^2 - s^3)\log n/n(a|V_1^+| + b|V_1^-|)1(\mathcal{G})$$
$$= (1 - o(1))(1+\epsilon)(2s^2 - s^3)\mathrm{T_c}(a,b)(\log n)1(\mathcal{G}).$$

The first inequality uses that $\log(1-x) \geq -(1+\epsilon)x$ provided $0 < x < \epsilon/(1+\epsilon)$. Setting $\epsilon = n^{-0.5}$, we have that

$$\mathbb{E}[|R^* \cap V_1^+||\sigma_1^*]1(\mathcal{G}) = \sum_{i \in V^+} \mathbb{P}(E_i|\sigma_1^*)1(\mathcal{G}) \geq n^{1-(2s^2-s^3)\mathrm{T_c}(a,b)-o(1)}1(\mathcal{G}).$$

Then we bound the variance of $|R^* \cap V_1^+|$. For $i, j \in V_1^+, i \neq j$:

$$\mathrm{Cov}(1(E_i), 1(E_j)|\sigma_1^*) = \mathbb{P}(E_iE_j|\sigma_1^*) - \mathbb{P}(E_i|\sigma_1^*)^2$$
$$= (1-s(2s-s^2)a\log n/n)^{2|V_1^+|-3}(1-s(2s-s^2)b\log n/n)^{2|V_1^-|}(1-(1-s(2s-s^2)a\log n/n)).$$

On event $\mathcal{G}$, $2|V_1^+| - 3 = (1+o(1))n$, $2|V_1^-| = (1+o(1))n$. Hence using the equality $1 - x \le e^{-x}$, we have that

$$\mathrm{Cov}(1(E_i), 1(E_j)|\sigma_1^*)1(\mathcal{G}) \le s(2s - s^2)a\log n/n \exp(-(1-o(1))s(2s-s^2)(a+b)\log n)1(\mathcal{G})$$
$$= n^{-1-(2s^2-s^3)(a+b)+o(1)}1(\mathcal{G}).$$

Then

$$\mathrm{Var}(|R^* \cap V_1^+||\sigma_1^*)1(\mathcal{G})$$
$$= \mathrm{Var}(\sum_{i\in V_1^+} 1(E_i)|\sigma_1^*)1(\mathcal{G}) = \sum_{i,j\in V_1^+} \mathrm{Cov}(1(E_i), 1(E_j)|\sigma_1^*)1(\mathcal{G})$$
$$\le \sum_{i\in V_1^+} \mathbb{P}(E_i|\sigma_1^*)1(\mathcal{G}) + \sum_{i\ne j\in V_1^+} \mathrm{Cov}(1(E_i), 1(E_j)|\sigma_1^*)1(\mathcal{G})$$
$$\le (n^{1-(2s^2-s^3)(a+b)+o(1)} + n^{1-(2s^2-s^3)(a+b)/2+o(1)})1(\mathcal{G}) = n^{1-(2s^2-s^3)(a+b)/2+o(1)}1(\mathcal{G}).$$

By the Paley-Zygmund inequality,

$$\mathbb{P}(|R^* \cap V_1^+| \ge n^{1-(2s^2-s^3)\mathrm{T}_c(a,b)-\delta}|\sigma_1^*)1(\mathcal{G})$$
$$\ge (1 - n^{-\delta+o(1)})^2 \frac{\mathbb{E}[|R^* \cap V_1^+||\sigma_1^*]^2}{\mathbb{E}[|R^* \cap V_1^+|^2|\sigma_1^*]}1(\mathcal{G})$$
$$\ge (1 - n^{-\delta+o(1)})^2 (1 - \frac{\mathrm{Var}[|R^* \cap V_1^+||\sigma_1^*]}{\mathbb{E}[|R^* \cap V_1^+||\sigma_1^*]^2})1(\mathcal{G})$$
$$\ge (1 - n^{-\delta+o(1)})^2 (1 - n^{-(1-(2s^2-s^3)\mathrm{T}_c(a,b))+o(1)})1(\mathcal{G}) = (1-o(1))1(\mathcal{G}).$$

Together with $\mathbb{P}(\mathcal{G}) = 1 - o(1)$, we have

$$\mathbb{P}(n^{1-(2s^2-s^3)\mathrm{T}_c(a,b)-\delta} \le |R^* \cap V_1^+| \le n^{1-(2s^2-s^3)\mathrm{T}_c(a,b)+\delta}) = 1 - o(1).$$

The proof for $|R^* \cap V_1^-|$ is the same. Now we study $|\bar{R}^* \cap V_1^+|$. Since $R^* \subset \bar{R}^*$, we have the lower bound $\mathbb{P}(|\bar{R}^* \cap V_1^+| \ge |R^* \cap V_1^+| \ge n^{1-(2s^2-s^3)\mathrm{T}_c(a,b)-\delta}) = 1 - o(1)$. For the upper bound,

$$|\bar{R}^* \cap V_1^+| \le |\bar{R}^*| \le \sum_{i\in R^*} (1 + |N_H(\pi_{12}^*(i))|) \le |R^*|(1 + \max_{j\in[n]} N_H(j)).$$

By Lemma D.6, since $H \sim \mathrm{SBM}(n, (1-(1-s)^2)a\log n/n, (1-(1-s)^2)b\log n/n)$, we have that $\max_{j\in[n]} N_H(j) \le 100\max a, b(1-(1-s)^2)\log n$ with probability $1 - o(1)$. Hence we have that with high probability

$$|\bar{R}^* \cap V_1^+| \le (1 + 100\max a, b(1-(1-s)^2)\log n)|R^*|$$
$$\le 2(1 + 100\max a, b(1-(1-s)^2)\log n)n^{1-(2s^2-s^3)\mathrm{T}_c(a,b)+\delta}$$
$$\le n^{1-(2s^2-s^3)\mathrm{T}_c(a,b)+\delta}.$$

### G.7 Proof of Lemma G.7, G.8

Firstly, we use some useful conditional independence properties given the sigma algebra $\mathcal{I}$.

**Lemma G.18.** *Let $i \in R^*$, conditioned on $\mathcal{I}$, $\{A_{ij} : \{i,j\} \in \mathcal{E}_{100}\}$ is a collection of mutually independent random variables where*

$$A_{ij} \sim \begin{cases} \mathrm{Ber}(a\log n/n) & \sigma_1^*(i) = \sigma_1^*(j), \\ \mathrm{Ber}(b\log n/n) & \sigma_1^*(i) = -\sigma_1^*(j). \end{cases}$$

*The random variables $1(i \in S^*)$ and $\sum_{j\in[n]\setminus\bar{R}^*} A_{i,j}\sigma_1^*(j)$ are conditionally independent given $\mathcal{I}$.*

*Proof.* Note that the sets $R^*, \bar{R}^*$ only depend on $\pi_{12}^*, H, G_1 \wedge_{\pi_{12}^*} H$. $G_1 \setminus_{\pi_{12}^*} H$ is comprised of edges in $\mathcal{E}_{100}$, the graph $H$ is comprised of edges in $\cup_{j+k>0}\mathcal{E}_{ijk}$, the graph $G_1 \wedge_{\pi_{12}^*} H$ is comprised of edges in $\cup_{j+k>0}\mathcal{E}_{1jk}$. Thus by lemma D.2, $G_1 \setminus_{\pi_{12}^*} H$ is conditionally independent of $R^*, \bar{R}^*$

given $\boldsymbol{\pi}^*, \sigma_1^*$ and the partition $\mathcal{E}$. In particular, $\{A_{i,j} : \{i,j\} \in \mathcal{E}_{100}\}$ is conditionally independent of $\mathcal{I}$.

Note that $1(i \in S^*) = 1(i \in R^*)1(A_{i,j} = 0, \forall j \in \bar{R}^*)$. Hence $1(i \in S^*)$ is measurable with respect to the sigma algebra generated by $\mathcal{I}$ and the collection $C_1 := \{A_{i,j} : j \in \bar{R}^*, \{i,j\} \in \mathcal{E}_{100}\}$. On the other hand, since $\bar{R}^*$ is $\mathcal{I}-$ measurable, $\sum_{j \in [n]\setminus\bar{R}^*} A_{i,j}\sigma_1^*(j)$ is measurable with respect to the sigma algebra generated by $\mathcal{I}$ and the collection $C_2 := \{A_{i,j} : j \in [n] \setminus \bar{R}^*, \{i,j\} \in \mathcal{E}_{100}\}$. $C_1 \cap C_2 = \emptyset$, the independence of $A_{i,j}$ implies that the two random variables are conditionally independent given $\mathcal{I}$. $\qquad\square$

**Lemma G.19.** *For any $\delta > 0$, $i \in R^*$,it holds for sufficiently large $n$ that*

$$\mathbb{P}(i \in S^*|\mathcal{I})1(\mathcal{G}_\delta) \geq (1 - n^{-(2s^2-s^3)\mathrm{T_c}(a,b)+2\delta})1(\mathcal{G}_\delta).$$

*Proof.* For $i \in R^*$, define the following random sets that are $\mathcal{I}-$ measurable:

$$C^+(i) := \{j \in \bar{R}^* : \{i,j\} \in \mathcal{E}_{100} \cap \mathcal{E}^+(\sigma_1^*)\},$$
$$C^-(i) := \{j \in \bar{R}^* : \{i,j\} \in \mathcal{E}_{100} \cap \mathcal{E}^-(\sigma_1^*)\}.$$

Note that $i \in S^*$ if and only if $i \in R^*$ and $A_{i,j} = 0, \forall j \in C^+(i) \cup C^-(i)$ conditioned on $\mathcal{I}$. Hence by Lemma G.18:

$$\begin{aligned}
\mathbb{P}(i \in S^*|\mathcal{I}) = \mathbb{P}(A_{i,j} = 0, \forall j \in C^+(i) \cup C^-(i)|\mathcal{I}) &= (1 - a\log n/n)^{|C^+(i)|}(1 - b\log n/n)^{|C_{(i)}|} \\
&\geq (1 - a|C^+(i)|\log n/n)(1 - b|C^-(i)|\log n/n) \\
&\geq 1 - (a|C^+(i)| + b|C^-(i)|)\log n/n.
\end{aligned}$$

The first inequality is because of Bernoulli's inequality. Note that on event $G_\delta$, $|C^+(i)|, |C^-(i)| \leq |\bar{R}^*| \leq 2n^{1-(2s^2-s^3)\mathrm{T_c}(a,b)+\delta}$. Thus

$$\mathbb{P}(i \in S^*|\mathcal{I})1(\mathcal{G}_\delta) \geq (1 - 2(a+b)n^{-(2s^2-s^3)\mathrm{T_c}(a,b)+\delta}\log n)1(\mathcal{G}_\delta).$$

Since $2(a+b)\log n \leq n^\delta$, the lemma follows. $\qquad\square$

Now, define the random variable

$$X_i := \begin{cases} \mathbb{P}(\sum_{j\in[n]\setminus\bar{R}^*} A_{i,j}\sigma_1^*(j) < 0|\mathcal{I}) & i \in R^* \cap V_1^+, \\ \mathbb{P}(\sum_{j\in[n]\setminus\bar{R}^*} A_{i,j}\sigma_1^*(j) > 0|\mathcal{I}) & i \in R^* \cap V_1^-. \end{cases}$$

Then we will study $X_i$ on the event $\mathcal{F} \cap G_\delta$.

**Lemma G.20.** *For $i \in R^*$,*

$$X_i 1(\mathcal{F} \cap G_\delta) = n^{-s(1-s)^2 \mathrm{D_+}(a,b)+o(1)}1(\mathcal{F} \cap G_\delta).$$

*Proof.* Suppose $i \in R^* \cap V_1^+$. Note that $\bar{R}^*, \sigma_1^*$ are $\mathcal{I}-$ measurable. By Lemma G.18:

$$\sum_{j\in[n]\setminus\bar{R}^*} A_{i,j}\sigma_1^*(j) \stackrel{d}{=} Y - Z$$

$$Y \sim \mathrm{Bin}(|\{j \in [n] \setminus \bar{R}^* : \{i,j\} \in \mathcal{E}_{100} \cap \mathcal{E}^+(\sigma_1^*)|, a\log n/n),$$
$$Z \sim \mathrm{Bin}(|\{j \in [n] \setminus \bar{R}^* : \{i,j\} \in \mathcal{E}_{100} \cap \mathcal{E}^-(\sigma_1^*)|, b\log n/n).$$

Where $Y, Z$ are independent. For brevity, suppose $Y \sim \mathrm{Bin}(y,p)$, $Z \sim \mathrm{Bin}(z,q)$. On the event $\mathcal{F} \cap G_\delta$, the upper bound for $y$:

$$y \leq |\{j \in [n] : \{i,j\} \in \mathcal{E}_{100} \cap \mathcal{E}^+(\sigma_1^*)| \leq s(1-s)^2(n/2 + 2n^{3/4}) = (1 + o(1))s(1-s)^2 n/2.$$

The lower bound for $y$:

$$\begin{aligned}
y &\geq |\{j \in [n] : \{i,j\} \in \mathcal{E}_{100} \cap \mathcal{E}^+(\sigma_1^*)| - |\bar{R}^*| \\
&\geq s(1-s)^2(n/2 - 2n^{3/4}) - n^{1-(2s^2-s^3)\mathrm{T_c}(a,b)+\delta} = (1 - o(1))s(1-s)^2 n/2.
\end{aligned}$$

Same calculation for $z$ and we can derive:
$$(1 - o(1))s(1 - s)^2 n/2 \le z \le (1 + o(1))s(1 - s)^2 n/2.$$

We can rewrite $p, q$ as $p = (1 + o(1)as(1 - s)^2 \log(s(1 - s)^2 n)/(s(1 - s)^2 n))$, $q = (1 + o(1)bs(1 - s)^2 \log(s(1 - s)^2 n)/(s(1 - s)^2 n))$, hence by Lemma D.1, we have

$$\mathbb{P}(\sum_{j \in [n] \setminus \bar{R}^*} A_{i,j} \sigma_1^*(j) < 0 | \mathcal{I}) 1(\mathcal{F} \cap G_\delta) = \mathbb{P}(Y - Z < 0 | \mathcal{I}) 1(\mathcal{F} \cap G_\delta) \le n^{-s(1-s)^2 \mathrm{D}_+(a,b) + o(1)}.$$

The same argument works for $i \in R^* \cap V_1^-$. $\qquad\qquad\qquad\qquad\qquad\qquad\qquad\qquad\square$

*Proof of Lemma G.7.* For $i \in R^* \cap V_1^+$, we have that

$$\mathbb{E}[W_i | \mathcal{I}] 1(\mathcal{F} \cap \mathcal{G}_\delta) = \mathbb{P}(i \in S^*, \sum_{j \in [n] \setminus \bar{R}^*} A_{i,j} \sigma_1^*(j) < 0 | \mathcal{I}) 1(\mathcal{F} \cap \mathcal{G}_\delta)$$

$$= \mathbb{P}(i \in S^* | \mathcal{I}) \mathbb{P}(\sum_{j \in [n] \setminus \bar{R}^*} A_{i,j} \sigma_1^*(j) < 0 | \mathcal{I}) 1(\mathcal{F} \cap \mathcal{G}_\delta).$$

The first equality holds because $A_{i,j} = 0$ if $i \in S^*, j \in \bar{R}^*$. The second equality exits because the conditional independence in Lemma G.18. By Lemm G.19 and G.20, we have that

$$\mathbb{E}[W_i | \mathcal{I}] 1(\mathcal{F} \cap \mathcal{G}_\delta) \ge (1 - n^{-(2s^2 - s^3) \mathrm{T_c}(a,b) + 2\delta}) X_i 1(\mathcal{F} \cap \mathcal{G}_\delta)$$

$$\ge (1 - n^{-(2s^2 - s^3) \mathrm{T_c}(a,b) + 2\delta}) n^{-s(1-s)^2 \mathrm{D}_+(a,b) + o(1)} 1(\mathcal{F} \cap \mathcal{G}_\delta).$$

Summing over $i \in R^* \cap V^+$, we have that

$$\mathbb{E}[\sum_{i \in R^* \cap V^+} W_i | \mathcal{I}] 1(\mathcal{F} \cap \mathcal{G}_\delta)$$

$$\ge |R^* \cap V^+|(1 - n^{-(2s^2 - s^3) \mathrm{T_c}(a,b) + 2\delta}) n^{-s(1-s)^2 \mathrm{D}_+(a,b) + o(1)} 1(\mathcal{F} \cap \mathcal{G}_\delta)$$

$$\ge (1 - n^{-(2s^2 - s^3) \mathrm{T_c}(a,b) + 2\delta}) n^{1 - (2s^2 - s^3) \mathrm{T_c}(a,b) - s(1-s)^2 \mathrm{D}_+(a,b) - \delta + o(1)} 1(\mathcal{F} \cap \mathcal{G}_\delta).$$

The second inequality is because on $\mathcal{G}_\delta$, $|R^* \cap V^+| \ge n^{1 - (2s^2 - s^3) \mathrm{T_c}(a,b) - \delta}$. The proof of lower bound for $i \in R^* \cap V_1^-$ is the same. $\qquad\qquad\qquad\qquad\qquad\qquad\square$

*Proof of Lemma G.8.* Suppose $i \ne j \in R^* \cap V_1^+$, we have

$$\mathbb{E}[W_i W_j | \mathcal{I}] = \mathbb{P}(i, j \in S^*, \sum_{k \in [n] \setminus \bar{R}^*} A_{i,k} \sigma_1^*(k) < 0, \sum_{k \in [n] \setminus \bar{R}^*} A_{j,k} \sigma_1^*(k) < 0 | \mathcal{I})$$

$$\le \mathbb{P}(\sum_{k \in [n] \setminus \bar{R}^*} A_{i,k} \sigma_1^*(k) < 0, \sum_{k \in [n] \setminus \bar{R}^*} A_{j,k} \sigma_1^*(k) < 0 | \mathcal{I})$$

$$= \mathbb{P}(\sum_{k \in [n] \setminus \bar{R}^*} A_{i,k} \sigma_1^*(k) < 0 | \mathcal{I}) \mathbb{P}(\sum_{k \in [n] \setminus \bar{R}^*} A_{j,k} \sigma_1^*(k) < 0 | \mathcal{I}) = X_i X_j.$$

The second inequality is because by Lemma G.18, $\{A_{i,k}\}_{k \in [n] \setminus \bar{R}^*}\}$ and $\{A_{j,k}\}_{k \in [n] \setminus \bar{R}^*}\}$ are conditionally independent given $\mathcal{I}$. Consider the case $i = j$,

$$\mathbb{E}[W_i^2 | \mathcal{I}] = \mathbb{P}(i \in S^*, \sum_{j \in [n] \setminus \bar{R}^*} A_{i,j} \sigma_1^*(j) < 0 | \mathcal{I}) \le X_i.$$

Summing over $i \in R^* \cap V_1^+$:

$$\mathbb{E}[(\sum_{i \in R^* \cap V^+} W_i)^2 | \mathcal{I}] = \sum_{i \in R^* \cap V^+} \mathbb{E}[W_i | \mathcal{I}] + \sum_{i \ne j \in R^* \cap V^+} \mathbb{E}[W_i W_j | \mathcal{I}]$$

$$\le \sum_{i \in R^* \cap V^+} X_i + \sum_{i \ne j \in R^* \cap V^+} X_i X_j \le \sum_{i \in R^* \cap V^+} X_i + (\sum_{i \in R^* \cap V^+} X_i)^2.$$

Note that

$$\mathbb{E}[\sum_{i\in R^*\cap V_1^+} W_i|\mathcal{I}]^2 1(\mathcal{F}\cap\mathcal{G}_\delta) = (\sum_{i\in R^*\cap V_1^+} \mathbb{P}(i\in S^*, \sum_{j\in[n]\setminus\bar{R}^*} A_{i,j}\sigma_1^*(j)<0|\mathcal{I}))^2 1(\mathcal{F}\cap\mathcal{G}_\delta)$$
$$\geq (1-n^{-(2s^2-s^3)\mathrm{T_c}(a,b)+2\delta})^2 (\sum_{i\in R^*\cap V_1^+} X_i)^2 1(\mathcal{F}\cap\mathcal{G}_\delta).$$

Hence we have

$$\mathrm{Var}(\sum_{i\in R^*\cap V_1^+} W_i|\mathcal{I})1(\mathcal{F}\cap\mathcal{G}_\delta)$$

$$= \left(\mathbb{E}[(\sum_{i\in R^*\cap V_1^+} W_i)^2|\mathcal{I}] - \mathbb{E}[\sum_{i\in R^*\cap V_1^+} W_i|\mathcal{I}]^2\right) 1(\mathcal{F}\cap\mathcal{G}_\delta)$$

$$\leq \left(\sum_{i\in R^*\cap V_1^+} X_i + (\sum_{i\in R^*\cap V_1^+} X_i)^2 \left(1-(1-n^{-(2s^2-s^3)\mathrm{T_c}(a,b)+2\delta})^2\right)\right) 1(\mathcal{F}\cap\mathcal{G}_\delta)$$

$$\leq \left(\sum_{i\in R^*\cap V_1^+} X_i + 2(\sum_{i\in R^*\cap V_1^+} X_i)^2 n^{-(2s^2-s^3)\mathrm{T_c}(a,b)+2\delta}\right) 1(\mathcal{F}\cap\mathcal{G}_\delta)$$

$$\leq \left(|R^*\cap V_1^+|n^{-s(1-s)^2\mathrm{D_+}(a,b)+o(1)} + 2\left(|R^*\cap V_1^+|n^{-s(1-s)^2\mathrm{D_+}(a,b)+o(1)}\right)^2 n^{-(2s^2-s^3)\mathrm{T_c}(a,b)+2\delta}\right) 1(\mathcal{F}\cap\mathcal{G}_\delta)$$

$$\leq (n^{\theta+\delta+o(1)} + 2n^{2\theta+2\delta-(2s^2-s^3)\mathrm{T_c}(a,b)+2\delta+o(1)})1(\mathcal{F}\cap\mathcal{G}_\delta).$$

The second inequality uses the fact that $1-(1-n^{-x})^2 \leq 2n^{-x}$. The third inequality exists by Lemma G.5, G.20. Now, for further simplying the upper bound, note that if $\delta$ is small enough (specifically, $\delta < \min((2s^2-s^3)\mathrm{T_c}(a,b)/8, \theta/4)$), then $\theta+\delta+o(1) < 2\theta-3\delta$, and $2\theta+4\delta-(2s^2-s^3)\mathrm{T_c}(a,b)+o(1) < 2\theta-3\delta$. Hence

$$\mathrm{Var}(\sum_{i\in R^*\cap V_1^+} W_i|\mathcal{I})1(\mathcal{F}\cap\mathcal{G}_\delta) \leq n^{2\theta-3\delta}1(\mathcal{F}\cap\mathcal{G}_\delta).$$

The proof for $\mathrm{Var}(\sum_{i\in R^*\cap V_1^-} W_i|\mathcal{I})1(\mathcal{F}\cap\mathcal{G}_\delta)$ is the same. $\qquad\square$

## H  Proof of Theorem 1 for $K$ graphs

### H.1  Categorization of vertices

We start with the definition of categorizing vertices as "good" and "bad" vertices. To begin with, we define a metagraph for each vertex. Then we categorize each vertex as "good" and "bad" according to the connectivity of the metagraph for the vertex.

**Definition H.1.** *Given* $(G_1,\ldots,G_K) \sim \mathrm{CSBM}(n, a\frac{\log n}{n}, b\frac{\log n}{n}, s)$, *and* $\binom{K}{2}$ *partial k-core matchings* $\widehat{\mu} := \{\widehat{\mu}_{ij} : i\neq j \in [K]\}$. *For any vertex* $v$, *define the following graph matching metagraph for the vertex* $v$, *denoted it as* $\mathcal{MG}_v$. $\mathcal{E}(\mathcal{MG}_v)$ *denotes the edge set in the graph* $\mathcal{MG}_v$. *There are* $K$ *nodes in* $\mathcal{MG}_v$, *where node* $i$ *represents the graph* $G_i$, *and an edge exists between* $(i,j)$, *that is,* $(i,j) \in \mathcal{E}(\mathcal{MG}_v)$ *if and only if vertex* $v$ *can be matched in the partial matching* $\widehat{\mu}_{ij}$ *between* $G_i$ *and* $G_j$.

Note that $\mathcal{MG}_v$ is an undirected graph. This is because of an inherent symmetry in the definition of a $k$-core matching, which looks at the $k$-core of the intersection graph of the two matched graphs. Thus for $k$-core matchings, a vertex is matched by $\widehat{\mu}_{ij}$ if and only if it is matched by $\widehat{\mu}_{ji}$. This property does not necessarily hold for other graph matching algorithms.

**Definition H.2.** *Given* $(G_1,\ldots,G_K) \sim \mathrm{CSBM}(n, a\frac{\log n}{n}, b\frac{\log n}{n}, s)$, *and* $\binom{K}{2}$ *partial k-core matchings* $\widehat{\mu} := \{\widehat{\mu}_{ij} : i\neq j \in [K]\}$, *we define a vertex* $v$ *to be "good" if and only if* $\mathcal{MG}_v$ *is connected. Conversely, a vertex* $v$ *is "bad" if and only if* $\mathcal{MG}_v$ *is disconnected.*

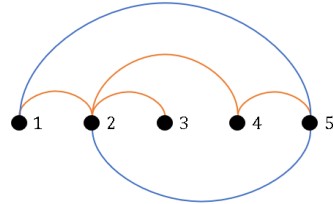

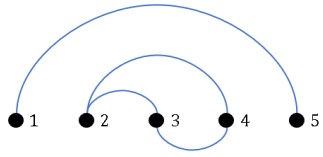

(a) $\mathcal{MG}_v$ for a "good" vertex $v$: $\mathcal{MG}_v$ is connected.

(b) $\mathcal{MG}_v$ for a "bad" vertex $v$: $\mathcal{MG}_v$ is disconnected.

Figure 4: Schematic showing the meta graph $\mathcal{MG}_v$ when $K = 5$.

For a "bad" vertex $v$, since $\mathcal{MG}_v$ is disconnected, the metagraph has at least two disjointed components. Hence, there must exist two sets $\Gamma_g(v), \Gamma_b(v)$ satisfying $\Gamma_g(v) \cap \Gamma_b(v) = \emptyset, \Gamma_g(v) \cup \Gamma_g(b) = [K]$ and for any $i \in \Gamma_g(v), j \in \Gamma_b(v), (i, j)$ is not an edge in $\mathcal{MG}_v$. In other words, $v$ cannot be matched for any matching between the graph $G_i, i \in \Gamma_g(v)$ and the graph $G_j, j \in \Gamma_b(v)$. Heuristically, the definition implies that "bad" vertices cannot utilize the combined information for all $K$ graphs.

Otherwise, the "good" vertices can utilize the combined information for all $K$ graphs, as shown in Lemma H.3.

**Lemma H.3.** *For a "good" vertex $v$ defined in Definition H.2, for any two node i,j (represents two graphs $G_i$, $G_j$), there exists a path $i := \ell_0 - \ell_1 - \ell_2 \ldots - \ell_d := j$ such that $v$ can be matched for $\widehat{\mu}_{\ell_m \ell_{m+1}}, m \in \{0, 1, ..., k-1\}$. Define $\widehat{\pi}_{ij}(v) := \widehat{\mu}_{\ell_{d-1}\ell_d} \circ \widehat{\mu}_{\ell_{d-2}\ell_{d-1}} \circ \ldots \circ \widehat{\mu}_{\ell_1 \ell_2} \circ \widehat{\mu}_{\ell_0 \ell_1}(v).$*

*Proof.* For a "good" vertex $v$, since $\mathcal{MG}_v$ is connected, for any two nodes $i, j \in [K]$, there exists a path $\psi_{ij}(v) := \{\ell_0 - \ell_1 - \ldots - \ell_{d-1} - \ell_d, \ell_0 = i, \ell_d = j, (\ell_m, \ell_{m+1}) \in \mathcal{E}(\mathcal{MG}_v)\}$. We can use the path $\psi_{ij}(v)$ to define the $\widehat{\pi}_{ij}(v)$ □

**Remark:** Note that such a path is not unique. However, the choice of path does not matter whp. By Lemma E.5, for $k$-core parital matching, $\widehat{\mu}_{ij} = \pi_{ij}^*, i, j \in [K]$, with high probability. Hence, for two different paths with endpoints being $i, j \in [K]$, denoted by $\psi_{ij}^1, \psi_{ij}^2$, we can define $\widehat{\pi}_{ij}^1, \widehat{\pi}_{ij}^2$ based on the two paths $\psi_{ij}^1, \psi_{ij}^2$ separately. Then, with high probability $\widehat{\pi}_{ij}^1 = \pi_{ij}^* = \widehat{\pi}_{ij}^2$.

By Lemma H.3 and its remark, we can have the union graph of $K$ graphs $(G_1 \vee G_2 \vee G_3 \vee \ldots \vee G_K)$ for the "good" vertices, using the matchings $\widehat{\boldsymbol{\pi}} := (\widehat{\pi}_{12}, \widehat{\pi}_{13}, \ldots, \widehat{\pi}_{1K})$ where $\widehat{\pi}_{1K}$ is defined in Lemma H.3. If there are multiple paths existing, pick the shortest one, and break ties in lexicographic order.

## H.2 Exact community recovery algorithm for $K$ graphs

The key steps of the exact community recovery algorithm for $K$ graphs are essentially the same as the algorithm for three graphs. Based on a given almost exact community recovery $\widehat{\sigma}_1$ and $\binom{K}{2}$ $k$-core matchings pairwise, we divide the vertices into two categories:"good" and "bad" vertices according to Definition H.1, H.2. Then refine the community label according to the majority votes for the "good" and "bad" vertices sequentially, to obtain the exact community recovery label under the given conditions. The full algorithm for exact community recovery is given in Algorithm 5:

## H.3 Analysis of the $k$-core estimator

Here we quantify the size of the"bad" vertices. Suppose that for vertex $v$, $\Gamma_g = \{G_1, G_2, \ldots, G_L\}$ and $\Gamma_b = \{G_{L+1}, G_{L+2}, \ldots, G_K\}$, here $1 < L < K - 1$, $v$ cannot be matched for all the matchings between one graph from $\Gamma_a$ and another graph from $\Gamma_b$. Apparently, $v$ cannot be matched for $L(K - L)$ matchings. Heuristically, as the number of mathings that cannot be matched for vertices increases, the size of such vertices decreases, since every matching would match $(1 - o(1))n$ vertices and only a very small fraction of vertices that cannot be matched. The following lemma demonstrated the claim.

---

**Algorithm 5** Community Recovery for $K$ graphs

---

**Input:** $K$ graphs $(G_1, G_2, \ldots, G_K)$ on $n$ vertices, $k = 13$, and $\epsilon > 0$.
**Output:** A labeling of $[n]$ given by $\widehat{\sigma}$.

1: Apply [35, Algorithm 1] to the graph $G_1$ and parameters $(sa, sb, \epsilon)$, obtaining a label $\widehat{\sigma_1}$.
2: Apply Algorithm 1 on input $(G_i, G_j, k)$, obtaining a matching $(\widehat{M}_{ij}, \widehat{\mu}_{ij}), i \neq j \in \{1, 2, .., K\}$.
3: For "good" vertices $v$, look at the set $\Psi := \{\widehat{\mu}_{ij}, i, j \in [K] : (i, j) \in \mathcal{E}(\mathcal{MG}_v)\}$, by Lemma H.3, we can define the $\widehat{\boldsymbol{\pi}} := (\widehat{\pi}_{12}, \widehat{\pi}_{13}, \ldots, \widehat{\pi}_{1K})$ to obtain the union graph of $K$ graphs based on the matchings from $\Psi$, denote it as $(G_1 \vee G_2 \ldots \vee G_K)_\Psi$. Denote $M := \cap M_{ij}$, where $(i, j)$ satisfying $\widehat{\mu}_{ij} \in \mathcal{MG}_v$. Set $\widehat{\sigma}(v) \in \{-1, 1\}$ according to the neighborhood majority (resp., minority) of $\widehat{\sigma_1}(v)$ with respect to the graph $(G_1 \vee G_2 \ldots \vee G_K)_\Psi \{M\}$ if $a > b$ (resp., $a < b$).
4: For "bad" vertices $v$, denote $\phi := \{j \in [K] : (1, j) \in \mathcal{E}(\mathcal{MG}_v)\}$. Denote $M := \cap_{i \in [K]} M_{1i}$, set $\widehat{\sigma}(v) \in \{-1, 1\}$ according to the neighborhood majority (resp., minority) of $\widehat{\sigma}(v)$ with respect to the graph $G_1 \backslash_{j \notin \phi} G_j(M \cup \{v\})$ if $a > b$ (resp., $a < b$).
5: Return $\widehat{\sigma} : [n] \to \{-1, 1\}$.

---

**Lemma H.4.** *Suppose that $G_1, G_2, \ldots, G_K$ are independently subsampled with probability $s$ from a parent graph $G \sim \mathrm{SBM}(n, a \log n/n, b \log n/n)$ for $a, b > 0$. Let $F_{ij}^*$ be the set of vertices outside the $k$-core of $G_i \wedge_{\pi_{ij}} G_j$ with $k = 13$. For $1 \leq L \leq K - 1$ and every $\delta > 0$, with probability $1 - o(1)$ we have that $|\cap_{1 \leq i \leq L, L+1 \leq j \leq K} F_{ij}^*| \leq n^{1 - s(1 - (1-s)^{K-1})\mathrm{T_c}(a,b) + \delta}$.*

*Proof.* Define $U_{ij}$ to be the set of vertices with degree at most $m + k$ in $G_i \wedge_{\pi_{ij}} G_j$, where $m > \frac{2}{(a+b)s^2}$, as defined in Lemma E.1. Then by Lemma E.1, w.h.p. we have $F_{ij}^* \subset U_{ij}$ and $|\cap_{1 \leq i \leq L, L+1 \leq j \leq K} F_{ij}^*| \leq |\cap_{1 \leq i \leq L, L+1 \leq j \leq K} U_{ij}|$. Now we look at the expectation:

$$\mathbb{E}\left[|\cap_{1 \leq i \leq L, L+1 \leq j \leq K} U_{ij}\right] = n\mathbb{E}\left[\prod_{i=1}^{L} \prod_{j=L+1}^{K} \mathbf{1}_{v \in U_{ij}}\right].$$

Let $D$ denote the degree of vertex $v$ in the graph $G_1 \vee G_3 \ldots \vee G_L$.

$$\mathbb{E}\left[\prod_{i=1}^{L} \prod_{j=L+1}^{K} \mathbf{1}_{v \in U_{ij}}\right] = \mathbb{E}\left[\mathbb{E}\left[\prod_{j=L+1}^{K} \prod_{i=1}^{L} \mathbf{1}_{v \in U_{ij}} \,\bigg|\, D\right]\right]$$

$$\leq \mathbb{E}\left[\left(\sum_{i=0}^{(m+k)L} \binom{D}{i} s^i(1-s)^{D-i}\right)^{K-L}\right]$$

The first equality is by the tower rule. The second inequality is due to two observations. Firstly, $\{\prod_{i=1}^{L} \mathbf{1}_{v \in U_{ij}}\} \subseteq \{\deg(v) \leq L(m+k)$ in the graph $(G_1 \vee G_3 \ldots \vee G_L) \wedge G_j, L+1 \leq j \leq K\}$. To be more detailed, if the degree of $v$ is at most $m + k$ in the graph $G_i \wedge G_j, 1 \leq i \leq L$, then the degree of $v$ is at most $L(m + k)$ in the graph $(G_1 \vee G_3 \ldots \vee G_L) \wedge G_j$. The second observation is, given $D$, for $j_1 \neq j_2$, the events $\{\deg(v) \leq L(m + k)$ in the graph $(G_1 \vee G_3 \ldots \vee G_L) \wedge G_{j_1}\}, \{\deg(v) \leq L(m + k)$ in the graph $(G_1 \vee G_3 \ldots \vee G_L) \wedge G_{j_2}\}$ are independent.

Similar to the proof of Lemma E.7, let $X_a \sim \mathrm{Bin}((1 + o(1))n/2, (1 - (1-s)^L)a \log n/n)$, and $X_b \sim \mathrm{Bin}((1 + o(1))n/2, (1 - (1-s)^L)b \log n/n)$. On the event $\mathcal{F}, D_1 \overset{d}{=} X_a + X_b$, where $\mathcal{F}$ is defined in Definition D.3, $X_a, X_b$ are independent. Note that by Lemma D.4, $\mathbb{P}(\mathcal{F}^c) = o(\frac{1}{n^2})$. We have

$$\mathbb{E}\left[\left(\sum_{i=0}^{m+k} \binom{D}{i} s^i(1-s)^{D-i}\right)^{K-L}\right]$$

$$= \sum_{i_1, i_2, \ldots, i_{K-L}=0}^{(m+k)L} C(i_1, i_2, \ldots, i_{K-L})\mathbb{E}[D^{\sum_{j=0}^{K-L} i_j}(1-s)^{(K-L)D}]. \quad \text{(H.5)}$$

Here $C(i_1, i_2, \ldots, i_{K-L})$ is a constant given $i_1, i_2, \ldots, i_{K-L}$. Now look at $\mathbb{E}[D^N(1-s)^{(K-L)D}]$. In our regime, $N \leq (m+k)(K-L)L$ are constant. Hence:

$$\mathbb{E}[D^N(1-s)^{(K-L)D}\mathbf{1}_\mathcal{F}] = \mathbb{E}[(X_a + X_b)^N(1-s)^{(K-L)X_a}(1-s)^{(K-L)X_b}\mathbf{1}_\mathcal{F}]$$

$$= \sum_{t=0}^N C_t \mathbb{E}[X_a^t(1-s)^{(K-L)X_a}\mathbf{1}_\mathcal{F}]\mathbb{E}[X_b^{N-t}(1-s)^{(K-L)X_a}\mathbf{1}_\mathcal{F}].$$

Here $C_t$ is constant related to $t$, the second equality is due to the independence of $X_a, X_b$. Now look at $\mathbb{E}[X_a^t(1-s)^{(K-L)X_a}\mathbf{1}_\mathcal{F}]$.

$$\mathbb{E}[X_a^t(1-s)^{(K-L)X_a}\mathbf{1}_\mathcal{F}] \leq \mathbb{E}[X_a^t(1-s)^{(K-L)X_a}]$$

$$= \sum_{\ell=0}^{(1+o(1))n/2} \ell^t(1-s)^{(K-L)\ell}\left(\frac{(1-(1-s)^L)a\log n}{n}\right)^\ell\left(1 - \frac{(1-(1-s)^L)a\log n}{n}\right)^{(1+o(1))n/2-\ell}$$

$$= \sum_{\ell=0}^{(\log n)^3} \ell^t\left(\frac{(1-s)^{K-L}(1-(1-s)^L)a\log n}{n}\right)^\ell\left(1 - \frac{(1-(1-s)^L)a\log n}{n}\right)^{(1+o(1))n/2-\ell}$$

$$+ \sum_{\ell=(\log n)^3+1}^{(1+o(1))n/2} \ell^t\left(\frac{(1-s)^{K-L}(1-(1-s)^L)a\log n}{n}\right)^\ell\left(1 - \frac{(1-(1-s)^L)a\log n}{n}\right)^{(1+o(1))n/2-\ell}.$$

Similarly, we can bound the first part:

$$\sum_{\ell=0}^{(\log n)^3} \ell^t\left(\frac{(1-s)^{K-L}(1-(1-s)^L)a\log n}{n}\right)^\ell\left(1 - \frac{(1-(1-s)^L)a\log n}{n}\right)^{(1+o(1))n/2-\ell}$$

$$\leq (\log n)^{3t}\sum_{\ell=0}^{(\log n)^3}\left(\frac{(1-s)^{K-L}(1-(1-s)^L)a\log n}{n}\right)^\ell\left(1 - \frac{(1-(1-s)^L)a\log n}{n}\right)^{(1+o(1))n/2-\ell}$$

$$\leq (\log n)^{3t}\sum_{\ell=0}^{(1+o(1))n/2}\left(\frac{(1-s)^{K-L}(1-(1-s)^L)a\log n}{n}\right)^\ell\left(1 - \frac{(1-(1-s)^L)a\log n}{n}\right)^{(1+o(1))n/2-\ell}$$

$$= (\log n)^{3t}\left(1 - (1-(1-s)^{K-L})(1-(1-s)^L)\frac{sa\log n}{n}\right)^{(1+o(1))n/2}$$

$$\leq n^{-(1-(1-s)^{K-L})(1-(1-s)^L)a/2+o(1)}.$$

Then we can bound the second part, using the similar arguments in Lemma E.7, we can show that

$$\sum_{\ell=(\log n)^3+1}^{(1+o(1))n/2} \ell^t\left(\frac{(1-s)^{K-L}(1-(1-s)^L)a\log n}{n}\right)^\ell\left(1 - \frac{(1-(1-s)^L)a\log n}{n}\right)^{(1+o(1))n/2-\ell}$$

$$= o(n^{-(1-(1-s)^{K-L})(1-(1-s)^L)a/2+o(1)}).$$

Hence, by summing up the two parts, $\mathbb{E}[X_a^t(1-s)^{(K-L)X_a}] \leq n^{-(1-(1-s)^{K-L})(1-(1-s)^L)a/2+o(1)}$. This is also true for $X_b$. Then, $\mathbb{E}[D^N(1-s)^{(K-L)D}] \leq n^{-(1-(1-s)^{K-L})(1-(1-s)^L)\mathrm{T}_c(a,b)+o(1)}$.

For $f(x) = (1-s)^x + (1-s)^{K-x}, x \in [1, K-1]$, the function $f(x)$ obtains its maximum at $x = 1, K-1$. Hence $n^{((1-s)^{K-L}-1)((1-(1-s)^L)\mathrm{T}_c(a,b)+o(1)} \leq n^{-s(1-(1-s)^{K-1})\mathrm{T}_c(a,b)+o(1)}$. Similar to Lemma E.7, by Markov inequality, the lemma follows immediately. $\square$

**Lemma H.6.** *The size of "bad" vertices defined in Definition H.2 can be upper bounded by* $n^{-s(1-(1-s)^{K-1})\mathrm{T}_c(a,b)+o(1)}$.

*Proof.* By definition H.2, the meta graph $\mathcal{MG}_v$ for "bad" vertex $v$ are disconnected. Hence, there exists at least two components of $\mathcal{MG}_v$, where there are no edges between the nodes from two components. $\square$

With all the preparations for "bad" vertices and "good" vertices, we are now ready to prove Theorem 1 for general $K$ graphs.

## H.4 Exact recovery for "good" vertices

By Lemma D.1, we can directly deduce that if $(1-(1-s)^K)D_+(a,b) > 1+\epsilon|\log(a/b)|$, then for all $i \in [n]$ we have that $\mathsf{maj}_{G_1 \vee G_2 \ldots \vee G_K}(i) \geq \epsilon \log n$ w.h.p.

Now for a "good" vertex $i$, there exists a corresponding matching set $\Psi$ that $i$ can be matched for all the matchings in $\Psi$ and there is an union graph $\widetilde{G} := (G_1 \vee G_2 \vee \ldots \vee G_3)$ that can be derived from the matchings in $\Psi$. Let $M$ be the set of vertices that can be matched for all matchings in $\Psi$.

**Lemma H.7.** *Suppose that* $(1-(1-s)^3)D_+(a,b) > 1+2\epsilon|\log(a/b)|$. *Then with probability* $1-o(1)$, *all the "good" vertices in $M$ have an $\epsilon \log n$ majority in $\widetilde{G}\{M\}$.*

*Proof.* Denote $F_{ij}^*$ the set of vertices outside the 13-core of $G_i \wedge_{\pi_{ij}^*} G_j$. In light of Lemma E.5 and its remark, we can replace $\mu_{ij}$ with $\pi_{ij}^*$, $F_{ij}$ with $F_{ij}^*$, $M$ with $M^*$ in Lemma F.2. Where we define

$$G^* := G_1 \vee_{\pi_{12}^*} G_2 \vee \ldots \vee_{\pi_{1K}^*} G_K, \qquad H := G^*\{M^*\}.$$

To bound the neighborhood majority in $H$, for $i \in M^*$ we have:

$$\mathsf{maj}_H(i) = \sigma^*(i) \sum_{j \in N_H(i)} \sigma^*(j) \leq \mathsf{maj}_{G^*}(i) + \sum_{\widehat{\mu}_{j\ell} \notin \Psi} |N_{G^*}(i) \cap F_{j\ell}^*|,$$

$$\mathsf{maj}_H(i) = \sigma^*(i) \sum_{j \in N_H(i)} \sigma^*(j) \geq \mathsf{maj}_{G^*}(i) - \sum_{\widehat{\mu}_{j\ell} \notin \Psi} |N_{G^*}(i) \cap F_{j\ell}^*|.$$

To sum up, we have

$$|\mathsf{maj}_H(i) - \mathsf{maj}_{G^*}(i)| \leq \sum_{\widehat{\mu}_{j\ell} \notin \Psi} |N_{G^*}(i) \cap F_{j\ell}^*|. \tag{H.8}$$

Note that $\mathsf{maj}_{G^*}(i) > 2\epsilon \log n$, $i \in [n]$ with probability $1-o(1)$, given that $(1-(1-s)^K)D_+(a,b) > 1+2\epsilon|\log(a/b)|$ by Lemma D.1. Now we would like to prove that the right hand side of (H.8) can be bounded by $\epsilon \log n$.

Note that $|N_{G^*}(i) \cap F_{j\ell}^*| \leq |N_{G_j \wedge G_\ell}(i) \cap (F_{j\ell}^*)| + |N_{G^* \setminus (G_j \wedge G_\ell)}(i) \cap (F_{j\ell}^*)|$.

First, look at $|N_{G_j \wedge G_\ell}(i) \cap (F_{j\ell}^*)|$, by Lemma E.4, w.h.p.,

$$|N_{G_j \wedge G_\ell}(i) \cap F_{j\ell}^*| < \epsilon \log n / 2K^2.$$

The remaining thing is to bound $|N_{G^* \setminus (G_j \wedge G_\ell)}(i) \cap (F_{j\ell}^*)|$. Note that conditioned on $\pi^*, \sigma^*, \mathcal{E} := \{\mathcal{E}_{i_1 i_2 \ldots i_K}, i_1, i_2, \ldots, i_K \in \{0,1\}\}$, the graph $G^* \setminus (G_j \wedge G_\ell)$ is independent of $F_{j\ell}^*$ by Lemma D.2, since $F_{j\ell}^*$ depends only on $G_j \wedge G_\ell$. Thus we can stochastically dominate $|N_{G^* \setminus (G_j \wedge G_\ell)}(i) \cap F_{j\ell}^*|$ by a Poisson random variable X with mean

$$\lambda_n := \nu \frac{\log n}{n} |\{j \in F_{j\ell}^* : \{i,j\} \in G^* \setminus (G_j \wedge G_\ell)| \leq \nu \frac{\log n}{n} |F_{j\ell}^*|, \nu := max(a,b).$$

For a fixed $\delta > 0$, define an event $\mathcal{Z} := \{|F_{j\ell}^*| \leq n^{1-s^2 \mathrm{T_c}(a,b)+\delta}\}$. On $\mathcal{Z}$, $\lambda_n \leq n^{-s^2 \mathrm{T_c}(a,b)+\delta+o(1)}$. Hence, for any positive integer $m$:

$$\mathbb{P}(\{|N_{G^* \setminus (G_j \wedge G_\ell)}(i) \cap (F_{j\ell}^*)| \geq m\} \cap \mathcal{Z} \leq \mathbb{P}(\{X \geq m\} \cap \mathcal{Z}) = \mathbb{E}[\mathbb{P}(X \geq m | F_{j\ell}^*, \mathcal{E}, \sigma^*, \pi^*) \mathbf{1}_{\mathcal{Z}}]$$

$$\leq \mathbb{E}[(\inf_{\theta > 0} e^{-\theta m + \lambda_n(e^\theta - 1)}) \mathbf{1}_{\mathcal{Z}}] \leq \mathbb{E}[e \lambda_n^m \mathbf{1}_{\mathcal{Z}}] \leq n^{-m(s^2 \mathrm{T_c}(a,b)-\delta-o(1))}.$$

Above, the equality on the second line is due to the tower rule and since $\mathcal{Z}$ is measurable with respect to $|F_{j\ell}^*|$, the inequality on the third line is due to a Chernoff bound; the inequality on the fourth line follows from setting $\theta = \log(1/\lambda_n)$ (which is valid since $\lambda_n = o(1)$ if $\mathcal{Z}$ holds). The final inequality uses the upper bound for $\lambda_n$ on $\mathcal{Z}$. Taking a union bound, we have

$$\mathbb{P}(\{\exists i \in [n], |N_{G^* \setminus (G_j \wedge G_\ell)}(i) \cap F_{j\ell}^*| \geq m\} \cap \mathcal{Z}) \leq n^{1-m(s^2 \mathrm{T_c}(a,b)-\delta-o(1))}.$$

Here if we take $m > (s^2 \mathrm{T_c}(a,b))^{-1}$ and $\delta < s^2 \mathrm{T_c}(a,b) - m^{-1}$, the probability turns to $o(1)$. Thus, we can set $m = \lceil (s^2 \mathrm{T_c}(a,b))^{-1} \rceil + 1$. In light of Lemma E.6, $|F_{j\ell}^*| \leq n^{1-s^2 \mathrm{T_c}(a,b)+\delta}$, $\delta > 0$ w.h.p. Hence, the event $\mathcal{Z}$ happens with probability $1-o(1)$. Hence we have

$$\mathbb{P}(\{\forall i \in [n], N_{G^* \setminus (G_j \wedge G_\ell)}(i) \cap F_{j\ell}^* \leq \lceil (s^2 \mathrm{T_c}(a,b))^{-1} \rceil\}) = 1-o(1).$$

Hence we have, with probability $1 - o(1)$, for $i \in M^*$

$$|\mathsf{maj}_H(i) - \mathsf{maj}_{G^*}(i)| < \sum_{\widehat{\mu}_{j\ell} \notin \Psi} (\epsilon \log n / 2K^2 + \lceil (s^2 \mathrm{T}_{\mathrm{c}}(a,b))^{-1} \rceil) < \epsilon \log n,$$

and hence with probability $1 - o(1)$,

$$\mathsf{maj}_H(i) > \epsilon \log n.$$

Then by Lemma E.5, we can replace $H$ with $\widetilde{G}$, $F_{ij}^*$ with $F_{ij}$, the lemma follows. $\qquad \square$

Next, prove that each vertex in $G^* \setminus_{\pi_{12}^*} G_1$ has a small number of neighbors in $I_\epsilon(G_1)$.

**Lemma H.9.** *If* $0 < \epsilon \leq \frac{s \mathrm{D}_+(a,b)}{4|\log(a/b)|}$, *then*

$$\mathbb{P}(\forall i \in [n], |N_{G^* \setminus_{\pi_{12}^*} G_1}(i) \cap I_\epsilon(G_1)| \leq 2\lceil (s\mathrm{D}_+(a,b))^{-1} \rceil) = 1 - o(1).$$

*Proof.* Since $I_\epsilon(G_1)$ depends on $G_1$ alone, it follows that $I_\epsilon(G_1)$ and $G^* \setminus_{\pi_{12}^*} G_1$ are conditionally independent given $\pi^*, \sigma^*, \mathcal{E}$. Hence we can stochastically dominate $|N_{G^* \setminus_{\pi_{12}^*} G_1}(i) \cap I_\epsilon(G_1)|$ by a Poisson random variable X with mean $\lambda_n$ given by

$$\lambda_n := \nu \log n / n |\{j \in I_\epsilon(G_1) : \{i,j\} \in G^* \setminus G_1\}| \leq \nu \log n / n |I_\epsilon(G_1)|.$$

Next, define the event $\mathcal{Z} := \{|I_\epsilon(G_1)| \leq n^{1 - s\mathrm{D}_+(a,b) + 2\epsilon|\log(a/b)|}\}$.

Notice that $P(\mathcal{Z}) = 1 - o(1)$ by Lemma D.7 and Markov's inequality, provided $s\mathrm{D}_+(a,b) < 99$. Following identical arguments as the proof of Lemma H.7, we arrive at

$$\mathbb{P}(\exists i \in [n], |N_{G^* \setminus_{\pi_{12}^*} G_1}(i) \cap I_\epsilon(G_1)| \geq m) = o(1)$$

when $m > \lceil (s\mathrm{D}_+(a,b) - 2\epsilon|\log a/b|)^{-1} \rceil$. If $\epsilon \leq \frac{s\mathrm{D}_+(a,b)}{4|\log(a/b)|}$, then it suffices to set $m = 2\lceil (s\mathrm{D}_+(a,b))^{-1} \rceil + 1$. $\qquad \square$

**Lemma H.10.** *Suppose that* $a, b, \epsilon > 0$ *satisfy the following conditions:*

$$(1 - (1-s)^K)\mathrm{D}_+(a,b) > 1 + 2\epsilon|\log a/b|, \quad 0 < \epsilon \leq \frac{s\mathrm{D}_+(a,b)}{4|\log a/b|}.$$

*With high probability, the algorithm correctly labels all vertices in* $\{i \in [n] \setminus M^*\}$.

*Proof.* Compare the neighborhood majority in $H$ corresponding to $\widehat{\sigma_1}$ with the true majority in $H$, where $H$ is defined in Lemma F.2:

$$|\sigma^*(i) \sum_{j \in N_H(i)} (\widehat{\sigma_1}(j) - \sigma^*(j))| \leq |N_H(i) \cap I_\epsilon| \leq |N_{G^*}(i) \cap I_\epsilon(G_1)|$$

$$\leq |N_{G^* \setminus G_1}(i) \cap I_\epsilon(G_1)| + |N_{G_1}(i) \cap I_\epsilon(G_1)| \leq 2\lceil \mathrm{D}_+(a,b)^{-1} \rceil + 2\lceil (s\mathrm{D}_+(a,b))^{-1} \rceil \leq \epsilon \log n / 2.$$

The first inequality uses Lemma D.5 that the set of errors are contained in $I_\epsilon(G_1)$. The last inequality is due to Lemma D.8, H.9. Notice that $\mathsf{maj}_H(i) \geq \epsilon \log n$ for $i \in M^*$. Hence, $\sigma^*(i) \sum_{j \in N_H(i)} \widehat{\sigma_1}(j) \geq \mathsf{maj}_H(i) - |\sigma^*(i) \sum_{j \in N_H(i)} (\widehat{\sigma_1}(j) - \sigma^*(j))| \geq \epsilon \log n / 2 > 0$, which implies that the sign of neighborhood majorities are equal to the truth community label for any $i \in M^*$, with probability $1 - o(1)$. Then we can convert $H$ to $\widetilde{G}\{M\}$, the vertices in $M$ are correctly labeled with probability $1 - o(1)$. $\qquad \square$

Using an identical proof, we can argue that the algorithm correctly labels all "good" vertices with probability $1 - o(1)$.

## H.5 Exact recovery for "bad" vertices

**Lemma H.11.** *Suppose that $a, b, \epsilon > 0$ satisfy the following conditions:*

$$(1 - (1-s)^K)\mathrm{D}_+(a,b) > 1 + 2\epsilon|\log a/b|, \ 0 < \epsilon \le \frac{s\mathrm{D}_+(a,b)}{4|\log a/b|},$$

$$s(1 - (1-s)^{K-1})\mathrm{T}_c(a,b) + s(1-s)^{K-1}\mathrm{D}_+(a,b) > 1.$$

*With high probability, the algorithm correctly labels all "bad" vertices.*

*Proof.* For vertex $i$ that are "bad", denote $\psi := \{j \in [K] : i \text{ cannot be matched through } \widehat{\mu}_{1j}, j \neq 1\}$. Denote $F_b$ as the vertex set of all the "bad" vertices that have the same $\psi$ with vertex $i$. Denote $M^* := \cap_{i \in [K]} M_{1i}^*$. define $H_i := (G_1 \setminus_{\pi_{12}^*} G_2 \setminus_{\pi_{13}^*} G_3 \dots \setminus_{\pi_{1K}^*} G_K)\{M \cup \{i\}\}$. Let $E_i$ be the event that i has a majority of at most $\epsilon' \log n$ in the graph $H_i$. Let $\widehat{\sigma}$ be the labeling after the step. For brevity, define a "nice" event based on the previous results. Define the event $\mathcal{H}$, which holds if and only if:

- $F_{ij} = F_{ij}^*$;

- $\widehat{\sigma}(i) = \sigma^*(i)$ for all $i \in M^*$;

- The event $\mathcal{F}$ holds;

- $|F_b| \le n^{1 - s(1-(1-s)^{K-1})\mathrm{T}_c(a,b)+\delta}$.

By Lemma E.5, H.4, D.4, H.10, the event $\mathcal{H}$ holds with probability $1 - o(1)$. Furthermore, define $E_i^* := \mathsf{maj}_{H_i}(i) \le \epsilon' \log n$, we have that

$$\mathbb{P}(\cup_{i \in [n]}(\{i \in F_b\} \cap E_i))$$

$$\le \mathbb{P}((\cup_{i \in [n]}(\{i \in F_b^*\} \cap E_i^*)) \cap \mathcal{H}) + \mathbb{P}(\mathcal{H}^c)$$

$$\le \sum_{i=1}^{n} \mathbb{P}(\{i \in F_b^*\} \cap E_i^* \cap \{F_b^* \le n^{1 - s(1-(1-s)^{K-1})\mathrm{T}_c(a,b)+\delta}\} \cap \mathcal{F}) + o(1).$$

By the tower rule, rewrite the term in the right hand side as:

$$\mathbb{E}\left[\mathbb{P}\left(E_i^* \mid \pi^*, \sigma^*, \mathcal{E}, F_b^*\right)\mathbf{1}_{i \in F_b^*}\mathbf{1}_{\{|F_b^*| \le n^{1 - s(1-(1-s)^{K-1})\mathrm{T}_c(a,b)+\delta}\} \cap \mathcal{F}}\right]. \tag{H.12}$$

Now look at $\mathbb{P}\left(E_i^* \mid \pi^*, \sigma^*, \mathcal{E}, F_b^*\right)$. Conditional on $\mathcal{E}, \sigma^*, \pi^*$, $\mathsf{maj}_{H_i}(i) :\overset{d}{=} Y - Z$, where $Y, Z$ are independent with:

$$Y \sim \mathrm{Bin}(|j \in M^* : \{i,j\} \in \mathcal{E}_{100\dots0} \cap \mathcal{E}^+(\sigma^*)|, a\log n/n),$$

$$Z \sim \mathrm{Bin}(|j \in M^* : \{i,j\} \in \mathcal{E}_{100\dots0} \cap \mathcal{E}^-(\sigma^*)|, b\log n/n).$$

By the Definition D.3 of the event $\mathcal{F}$, we know that $|j \in M^* : \{i,j\} \in \mathcal{E}_{100\dots0} \cap \mathcal{E}^-(\sigma^*)| = (1 - o(1))s(1-s)^{K-1}n/2$ and $|j \in M^* : \{i,j\} \in \mathcal{E}_{100..0} \cap \mathcal{E}^+(\sigma^*)| = (1 - o(1))s(1-s)^{K-1}n/2$. Lemma D.1 implies that

$$\mathbb{P}(E_i^*|\pi^*, \sigma^*, \mathcal{E}, F_b^*)\mathbf{1}_{i \in F_b^*}\mathbf{1}_{\{|F_b^*| \le n^{1-s(1-(1-s)^{K-1})\mathrm{T}_c(a,b)+\delta}\} \cap \mathcal{F}}$$

$$\le n^{-s(1-s)^{K-1}\mathrm{D}_+(a,b)+\epsilon'\log(a/b)/2+o(1)}.$$

Follow (H.12) and take a union bound, we have that

$$\sum_{i=1}^{n} \mathbb{P}(\{i \in F_b^*\} \cap E_i^* \cap \{F_b^* \le n^{1 - s(1-(1-s)^{K-1})\mathrm{T}_c(a,b)+\delta}\} \cap \mathcal{F}) + o(1)$$

$$\le n^{-s(1-s)^{K-1}\mathrm{D}_+(a,b)+\epsilon'\log(a/b)/2+o(1)}\mathbb{E}[|F_b^*|\mathbf{1}_{F_b^* \le n^{1-s(1-(1-s)^{K-1})}}]$$

$$\le n^{1 - s(1-(1-s)^{K-1})\mathrm{T}_c(a,b)-s(1-s)^{K-1}\mathrm{D}_+(a,b)+\epsilon'\log(a/b)/2+\delta}.$$

Under the condition $s(1-(1-s)^{K-1})\mathrm{T_c}(a,b) + s(1-s)^{K-1}\mathrm{D_+}(a,b) > 1$, we can choose $\epsilon', \delta$ small enough so that the right hand side is $o(1)$. $\mathsf{maj}_{H_i}(i) > \epsilon' \log n$ for $i \in F_b^*$, by Lemma E.5, $\mathsf{maj}_{\widehat{H}_i}(i) > \epsilon' \log n$ for $i \in F_b$.

Note that $i$ cannot be matched for all $\widehat{\mu}_{1i}, i \in \psi$. Hence $i$ has at most 12 neighbors in the graph $(G_1 \wedge_{\pi_{1i}^*} G_i)$. Therefore for any $i \in F_b$ has at least $\epsilon' \log n - 12|\psi|$ majority in $G_1 \setminus_{\widehat{\mu}_{1j}, j \notin \psi} G_j\{M \cup \{i\}\}$ with high probability. Hence, we can correctly label all vertices in $F_b$ with high probability.

Use the same arguments for all types of "bad" vertices, we can correctly label all "bad" vertices. $\square$

# I  Proof of impossibility for $K$ graphs

We study the MAP (maximum a posterior) estimator for the communities in $G_1$. Even with the additional information provided, including all the correct community labels in $G_2$, the true matching $\pi_{23}^*, \pi_{24}^*, \ldots, \pi_{2K}^*$ and most of the true matching $\pi_{12}^*$, the MAP estimator fails to exactly recovery communities with probability bounded away form 0 if the condition (G.1) holds. The proof can be derived by generalizing proof of impossibility for three graphs. The only difference is that we are considering $K$ correlated SBM $G_1, G_2, \ldots, G_K$. Since we know the true matching $\pi_{i,j}^*, i, j \in \{2, 3, \ldots, K\}$, we can consider $H := G_2 \vee G_3 \ldots \vee G_K \sim \mathrm{SBM}(n, (1-(1-s)^{K-1})a \log n/n, (1-(1-s)^{K-1})b \log n/n)$. Denote $R_{ij}$ the singleton in $G_i \wedge G_j$. Then $R = R_{12} \wedge R_{13} \ldots \wedge R_{1K}$ is the singleton set in $G_1 \wedge H$. The proof follows the same arguments with more involved notation, and hence we omit the details. Here we point out the differences of the proof for $K$ graphs.

Define $R_\pi := R(\pi, A, B^2, ..., B^K) := \{i \in [n] : \forall j \in [n], A_{i,j}D_{\pi(i)\pi(j)} = 0, D = \max(B^2, ...., B^K)\}$, here $B^i$ is the adjacent matrix of $G_k$. Similar to Definition G.2, we can define $S_\pi = S(\pi, A, B^2, .., B^K)$. Let $G_\delta$ be the event that the following inequalities all hold:

$$n^{1-s(1-(1-s)^{K-1})T_c(a,b)-\delta} \le |R^* \cap V_1^+|, |R^* \cap V_1^-|, |\bar{R}^* \cap V_1^+|, |\bar{R}^* \cap V_1^+|$$
$$\le n^{1-s(1-(1-s)^{K-1})T_c(a,b)+\delta}.$$

We can prove similar versions of Lemma G.7 and G.8, with $(2s^2 - s^3)$ replaced by $s(1-(1-s)^{K-1})$ and $s(1-s)^2$ replaced by $s(1-s)^{K-1}$. We can have same versions of Lemma G.9, Corollary G.12. We can similarly define $\mu^+(\pi)_{i_1 i_2 \ldots i_K}$ and $\mu^-(\pi)_{i_1 i_2 \ldots i_K}$ for all $i_1, \ldots, i_K \in \{0, 1\}$, and $\nu^+(\pi)$ and $\nu^-(\pi)$. When deriving the posterior distribution of $\pi_{12}^*$, similar to Lemma G.14, the information of $A, B^2, \ldots, B^K, \sigma_2^*, S^*, \pi_{12}^*[[n] \setminus S^*], \boldsymbol{\pi}^* := \{\pi_{ij}^*, i, j \in \{2, 3, \ldots, K\}\}$ are given. Note that for $\pi \in \mathcal{A}^*$, we have that $\mu^+(\pi)_{1i_2\ldots i_K}$ and $\mu^-(\pi)_{1i_2\ldots i_K}$ are constant for all $i_2, \ldots, i_K \in \{0, 1\}$ except for $\mu^+(\pi)_{10\ldots 0}$ and $\mu^-(\pi)_{10\ldots 0}$. We can derive an analogue of Lemma G.14 with $p_{100}$ replaced by $p_{100\ldots 0}$, $p_{000}$ replaced by $p_{000\ldots 0}$ and similar for $q$. Then we have analogous versions of Lemmas G.15, G.16, and G.17, with $p_{100}$ replaced by $p_{100\ldots 0}$, $p_{000}$ replaced by $p_{000\ldots 0}$, and similar for $q$. Note that $\frac{p_{100\ldots 0}q_{000\ldots 0}}{p_{000\ldots 0}q_{100\ldots 0}} = (1 + o(1))\frac{a}{b}$. The impossibility proof for Theorem 2 follows.

# J  Proofs for exact graph matching

## J.1  Exact graph matching for $K$ graphs

*Proof.* Through the 13-core matching in Algorithm 1, we obtain $\widetilde{\pi} := \{\widehat{\mu}_{ij}, i, j \in [K]\}$, where $\widehat{\mu}_{ij}$ is the 13-core matching between the graph $G_i$ and $G_j$.

Recall Definition H.2, we can directly infer that the "good" vertices are those which can be matched for $K$ graphs through a path across 13-core estimators $\widetilde{\pi}$. We can define a new estimator $\widehat{\pi}$ for those "good" vertices using the combination of 13-core estimator through the path that connects all $K$ graphs. The path is defined as in Lemma H.3. For any "good" vertex, such path exists and we can define the estimator $\widehat{\pi}$ for that vertex.

Note that, by Lemma E.5, with high probability $\widehat{\mu}_{ij} = \pi_{ij}^*$. Hence if for all the matched vertices, they will be matched correctly. If the number of "bad" vertices approaches zero, it indicates that all vertices are correctly matched. Consequently, exact graph matching can be achieved through the 13-core matching algorithm. By Lemma H.4, H.6, we can quantify the size of "bad" vertices: for every $\delta > 0$ we have that $|\cap_{1 \le i \le L, L+1 \le j \le K} F_{ij}^*| \le n^{1-s(1-(1-s)^{K-1})\mathrm{T_c}(a,b)+\delta}$. When $1 - s(1 - $

$(1-s)^{K-1})\mathrm{T_c}(a,b) > 1$, that is, when the condition (3.5) holds, the number of "bad" vertices goes to zero when $n$ goes to infinity, with high probability. Thus all vertices can be correctly matched and exact graph matching for $K$ graphs is possible with high probability. $\square$

## J.2 Impossibility of exact graph matching for $K$ graphs

*Proof.* Now we consider the graph matching problem with additional information provided, including the true correspondences $\pi_{23}^*, \ldots, \pi_{2K}^*$ and the community label $\sigma^*$. Then we obtain the union graph $H := G_2 \vee_{\pi_{23}^*} G_3 \vee \ldots \vee_{\pi_{2K}^*} G_K$. We now prove the impossibility by contradictory. Suppose that there exists an estimator $\widehat{\pi}$ which can exactly match $G_1, G_2, ..., G_K$, note that $H \sim \mathrm{SBM}(n, \frac{(1-(1-s)^{K-1})a\log n}{n}, \frac{(1-(1-s)^{K-1})b\log n}{n})$. One key point, is that, we can subsample $H_2', H_3', ..., H_K'$ from $H$. To be more specific, consider the following parameter: $r_{i_1,i_2,...,i_K} = \frac{s^{\sum_{j=1}^K i_j}(1-s)^{K-\sum_{j=1}^K i_j}}{1-(1-s)^{K-1}}$ where $i_1, i_2, \ldots, i_K \in \{0,1\}$ and $\sum_{j=1}^K i_j > 0$. Here $\sum r_{i_1,i_2,...,i_K} = 1$. Then for any vertex pair $(i,j)$:

1. If $(i,j)$ is not an edge in $H$, then $(i,j)$ is not an edge in $H_2', H_3', \ldots, H_K'$.

2. If $(i,j)$ is an edge in $H$, with probability $r_{i_1,i_2,...,i_K}$, $(i,j)$ is an edge in the graphs $\{H_{i_j}'\}$ where $i_j = 1$ in $r_{i_1,i_2,...,i_K}$ and $(i,j)$ is not an edge in the graphs $\{H_{i_j}'\}$ where $i_j = 0$ in $r_{i_1,i_2,...,i_K}$.

Following the subsampling described as above, we can simulate $H_2', \ldots, H_K'$ from $H$. Note that by construction, $(G_1, G_2', G_3', \ldots, G_K')$ has the same distribution as $(G_1, H_2', \ldots, H_K')$. Then after independent permutations, we can obtain $(H_3, H_4, \ldots, H_K)$ by relabeling the vertex index in $(H_3', H_4', \ldots, H_K')$. Note that $H_2 = H_2'$. Then $(G_1, G_2, \ldots, G_K)$ has the same distribution as $(G_1, H_2, H_3, \ldots, H_K)$. Since the estimator $\widehat{\pi}$ can exactly match $(G_1, G_2, \ldots, G_K)$ with high probability, it can also exactly match $(G_1, H_2, H_3, \ldots, H_k)$ with high probability. Naturally, it can exactly match vertices in $G_1$ and $H_2$, since $H$ and $H_2$ share the same vertex index then we can have an estimator that exactly match $G_1$ and $H$ given $G_1$ and $H$, where $G_1, H$ are correlated SBMs, independently subsampling from the parent graph $G$ with probability $s_1 = s$ for $G_1$ and $s_2 = 1 - (1-s)^{K-1}$ for $H$. However, [14, Theorem 1] proves that suppose $(G_1, G_2) \sim \mathrm{CSBM}(n, \frac{a\log n}{n}, \frac{b\log n}{n}, s_1, s_2)$ subsampling from $G \sim \mathrm{SBM}(n, \frac{a\log n}{n}, \frac{b\log n}{n})$ with probability $s_1$ and $s_2$, respectively, if $s_1 s_2 \mathrm{T_c}(a,b) < 1$, then exact graph matching between $G_1$ and $G_2$ is impossible. Directly applying [14, Theorem 1] we have that exact graph matching between $G_1$ and $H$ is impossible if $s(1 - (1-s)^{K-1}) < 1$. This is a contradiction, and hence exact graph matching for $G_1, \ldots, G_K$ is impossible. $\square$

