# OpenReview forum: "Harnessing Multiple Correlated Networks for Exact Community Recovery"
_NeurIPS.cc/2024/Conference — NeurIPS 2024 poster_

### Official Review · Reviewer_b7qh · 2024-07-03

**Soundness:** 4
**Presentation:** 4
**Contribution:** 4
**Rating:** 8
**Confidence:** 3

**Summary:**

The paper studies the problem of exact community detection in correlated stochastic block models (with two symmetric communities). More precisely, a graph is sampled from an SBM with parameters p and q and each edge in the graph is then kept with probability q. This downsampling is performed K times independently at random to obtain K different graphs. Each graph is then randomly permuted. From these K correlated sample we want to recover the communities of the model.

The authors provide a necessary and sufficient condition on the parameters p,q,k, and s for exact recovery.
In particular, they show there exists a regime of p,q,s where K-1 graphs are not enough to recover exactly the communities, but K samples are enough. In particular, there exists a regime in which K samples might not even be enough to solve graph matching perfectly (i.e., recover the random permutation). Therefore, it is necessary to aggregate imperfect matching information with techniques for community detection.

The case for K=2 has been solved by Gaudio, Rácz, and Sridhar. This paper extends their result for any fixed K>=2.

**Strengths:**

I think the problem is a natural and interesting one. The extension from the case of two to more correlated graphs appears highly nontrivial, since there are many ways in which one could try and resolve potentially conflicting information arising from matching different pairs of graphs.

The paper is also well written, both in explaining the background and motivation to the problem, and also when giving a flavour of the techniques used in the analysis.

**Weaknesses:**

Perhaps the main weakness of the paper is that the analysis is not algorithmic, in the sense that there is not an efficient algorithm able to recover the communities. This is, however, more an invitation to future work rather than a flaw of the paper itself.

The other point I'd like to raise is that the proofs in this paper are fairly lengthy and involved. Is NeurIPS the best venue for these kind of papers? I believe the paper will be interesting to the NeurIPS community and that's why I recommend acceptance, but in a perfect world I'd like results of this kind to be also checked for correctness (disclaimer: I did not read the appendix).

**Questions:**

Just a few questions that are essentially out of curiosity:
1. Can some of these techniques be used to analyse the sparse case? Of course we wouldn't aim for exact recovery, but rather partial recovery, and we would need to use different algorithms to obtain a good community labelling of the individual graphs.
1. I guess the limit of the recovery probability depends on K and that's the reason K needs to be constant? How bad is its dependency on K? Can K diverge at least slightly?
1. I know this is not the point of this paper, but I wonder if there is a practical application of the techniques developed in the paper. It'd be quite interesting to try and apply them to obtain an algorithm that work in practice, even without theoretical guarantees.

**Limitations:**

Yes.

---

> ### Author Rebuttal · Authors · 2024-08-07
>
> Thank you for your review and comments!
>
> We fully agree with the reviewer that the main open problem that our work leaves open is to develop efficient algorithms for the setting studied in our paper (whenever this is possible). The quest for efficient algorithms has been a major driving force behind the recent surge of papers on random graph matching (see Section 5), culminating in the breakthrough works [27] and [28] on correlated Erdos--Renyi random graphs. These works give promise that it is possible to develop efficient algorithms in our setting (perhaps with some conditions on the correlation parameter $s$). Our work lays the groundwork by determining the information-theoretic limits, which provide targets for efficient algorithms to achieve.
>
> Q1 (sparse setting): Weak recovery in the constant average degree regime using correlated graphs is indeed a fascinating direction for future work (as already suggested in [18]). We suspect that the $k$-core analysis still gives something nontrivial in the large but constant average degree regime. However, proving this is far from immediate, due to a lack of concentration at this scale. Furthermore, determining the precise threshold for weak/partial recovery seems like a challenging problem which requires many more new ideas.
>
> Q2 ($K$ diverging with $n$): An inspection of our proofs shows that our results hold also when $K$ diverges (slowly) with $n$. We have not made an attempt to see how fast $K$ can grow with $n$ to have the results still hold, in part because the conditions (3.1) and (3.2) converge exponentially in $K$, so they are already very close to their limiting values for large constant $K$.
>
> That said, as an example, let us consider one particular aspect of the proof -- the correctness of the $k$-core estimator -- and see what we may allow $K$ to be here. Lemma F.4 says that the result holds with high probability; in other words, that the error probability is $o(1)$. A careful inspection of the proof of Lemma 4.8 in [18] actually shows that the error probability is bounded by $n^{-0.16}$ (moreover, if $k$ in the $k$-core is chosen to be a large constant, then this error probability can be made to be an arbitrarily small inverse polynomial in $n$). Since in a union bound over $K$ graphs we gain a factor of at most $K^{2}$, this shows that this lemma allows $K$ to grow polynomially in $n$. Of course, there are other aspects of the proof where the dependence on $K$ of the error probabilities have to be checked, and together these can give a full answer of how fast $K$ can grow with $n$.
>
> Q3 (practical applications): We fully agree with the reviewer and we hope that such applications will arise in the future, motivated by our (theoretical) work.

---

> > ### Comment · Reviewer_b7qh · 2024-08-08
> >
> > Thanks for your response. I agree that the fact that the threshold depends exponentially on K makes handling the non-constant case not super important, but I appreciated the explanation.

---

### Official Review · Reviewer_cuxD · 2024-07-09

**Soundness:** 4
**Presentation:** 4
**Contribution:** 3
**Rating:** 7
**Confidence:** 4

**Summary:**

Given K correlated SBMs, the authors derive information-theoretic conditions for (i) the exact recovery of the community structure and (ii) the perfect recovery of the planted alignment $\pi^*$.

**Strengths:**

The paper is well-written, and I enjoyed reading it. It generalizes [37] and [18] to multiple (K \ge 3) SBMs. The interplay between community recovery and graph alignment by combining the information from the K graphs is well-explained and well-executed.

**Weaknesses:**

No major weaknesses, the paper opens and closes the problem it intends to solve. The discussion section lays directions for interesting future works. I hesitate between 7 and 8. But for a higher grade, I would have liked to see a bit more (such as graphs with 2 communities of different sizes, or more than 2 communities, which I believe is not that much harder). In any case, the paper is a clear accept.

Minor comments:

* Since the authors consider SBM with two balanced communities and edge probabilities p&q, the term planted partition model may be more adapted.

* In the planted partition model, the key information-theoretic quantity for exact recovery is the Rényi divergence of order 1/2 between two Bernoulli distributions. The CH divergence is only needed for SBM with general connection probabilities and/or block sizes.

* Typo line 143: "for K \ge graphs", 3 is missing.

* It may be useful to define earlier "almost exact community labeling" and "partial almost exact graph matching" (which I believe are accounted first in lines 179 and 180, and not before, and define only lines 241).

**Questions:**

* Can you elaborate on why k-core matching is a good choice? In particular, going over the proof, it appears that k = 13 is used (Lemmas F.4, F.6), is the choice of k important?

* Is there a low-degree hardness conjecture for the alignment of correlated SBMs?

* What happens when $K$ grows unbounded?

* Can one conjecture what happens with k \ge 3 communities of equal size? Going over ref [43], it seems natural to replace the quantity T_c by (a+(k-1)b) / k (which by the way, multiplied by \log n, is the average degree) and replace D_+ by (\sqrt(a) - \sqrt(b) )^2 / k.

* With k=2 communities of different sizes, could one get a different story than simply replacing the D_+ with the CH divergence? (a naive thinking: the step of almost exact recovery provides 2 communities of sizes n_1 < n_2 for G_1 and of sizes n_1' < n_2' for G_2'; thus I can already map the community of size n_1 with the one of size n_1'. Does it help?).

**Limitations:**

The theoretical results and their assumptions are clearly stated. The work is purely theoretical and does not require more discussion on societal impact.

---

> ### Author Rebuttal · Authors · 2024-08-07
>
> Thank you for your review and comments!
>
> We agree with the reviewer that it is natural to consider more general settings, such as more than two communities, as well as unbalanced communities. We suspect that our methods can be extended to these settings as well. However, in the present paper we decided to keep the setting as simple as possible, to focus on the effect of more than two graphs.
>
> Thank you for your comments on the planted partition model and the Renyi divergence; we will add these terminologies to the paper. Thank you also for spotting that typo, and for suggesting to move forward some definitions; we will make these edits in the revision.
>
> Q1 (on the $k$-core matching): The general intuition behind why the $k$-core matching performs well is as follows. The $k$-core of a graph captures its "central" part, in some sense. Thus if a vertex is part of the $k$-core of the intersection graph, then it has many common connections in the two graphs, so it is likely that these can help find the correct match for this vertex.
>
> The choice of $k=13$ is not particularly important -- any large enough constant will work. The particular choice of $k=13$ comes from [18], where it arises in their Lemma 4.8 and Lemma 4.10. In particular, Lemma 4.10 is a probability tail bound, and the choice of $k \geq 13$ (and in particular $k=13$) guarantees that this vanishes sufficiently quickly. The proof of Lemma 4.10 contains a union bound that may be loose, so it may be possible that the value of $k$ can be further lowered, but this doesn't have any major importance.
>
>
> Q2 (low-degree hardness): Yes, we conjecture that there is an information-computation gap for matching correlated SBMs, and we conjecture that this is the same as for matching correlated Erdos--Renyi random graphs, as in [28]. Namely, let $\alpha \approx 0.338$ denote Otter's tree counting constant [28]. We conjecture that if graph matching is information-theoretically feasible and $s^2 > \alpha$, then there is an efficient algorithm to do so. On the other hand, if $s^2 < \alpha$, then there does not exist such an efficient algorithm. Investigating this question is beyond the scope of the current paper.
>
>
> Q3 ($K$ unbounded): As $K \to \infty$ (and assuming $s \in (0,1)$), the condition (3.1) converges to the condition that exact community recovery is possible in the parent graph $G$, while the condition (3.2) converges to the condition that $G_{1}$ is connected. The convergence in both cases is exponential in $K$, so for large constant $K$ the conditions will already be very close to the limiting conditions.
>
> An inspection of our proofs shows that our results also hold for $K$ growing slowly with $n$; we haven't tried to optimize the bounds to allow $K$ to grow as fast as possible.
>
> Q4 (3+ symmetric communities): We fully agree with the reviewer's conjecture (which generalizes the conjecture of reference [43] for $K=2$). We suspect that our methods can be extended to show this. However, in the present paper our aim was to focus on the effect of the number of graphs $K$, so we decided to keep other aspects of the setting as simple as possible. This is an interesting question for future work.
>
> Q5 (two unbalanced communities): This is a natural and intriguing question. However, we believe that unbalanced communities do not help. Note that while unbalanced communities allow the communities to be matched, what we really need is to match individual vertices, and this is not readily done simply by virtue of the communities being unbalanced. Note also that after the partial matching and the initial majority votes, all nodes that have been matched have a correct estimate of their community. However, the condition (3.2) really comes from the (number of) unmatched vertices, and the related genie-aided argument, which gives the CH divergence. Of course, the arguments above are not a proof, just some heuristics, and we leave resolving this is as an interesting open question for the future.

---

> > ### Comment · Reviewer_cuxD · 2024-08-08
> >
> > Thank you for your answers! It answers my questions very well. My overall rating remains unchanged, I recommend acceptance of the paper.

---

### Official Review · Reviewer_c8Tz · 2024-07-10

**Soundness:** 4
**Presentation:** 4
**Contribution:** 3
**Rating:** 7
**Confidence:** 4

**Summary:**

The paper studies the problem of exact community recovery from multiple ($K$) correlated graphs in 2-community balanced symmetric SBM. Prior work of [Gaudio, Ra\'cz, and Sridhar 2022] for $K=2$ case. The paper generalizes their result for any $K$ (constant) number of graphs. In particular, the main result of the paper determines a sharp information-theoretic threshold in terms of $a,b,s$ (correlated SBM parameters) and $K$ such that
1. (Theorem 1) Above the threshold, the optimal MAP estimator (not efficient though) achieves exact recovery with high probability. To show this, the main challenge is to combine the information from more than two networks when none of the pairs can be matched exactly.
2. (Theorem 2) Below the threshold, any estimator fails to exactly recover the communities with high probability.

In particular, some interesting highlights from their results are that there is a region of parameter $(a,b,s)$ such that exact recovery is (i) impossible using $K-1$ graphs but possible using $K$ graphs AND (ii) matching the vertex labels of any of the graph pair is impossible.

**Strengths:**

1. The paper provides a clean characterization of the precise information theoretic limit for an important problem in exact community recovery literature.
2. The paper is well-written with clear intuitions of interplay between exact recovery and graph matching.

**Weaknesses:**

I do not see any major weaknesses in the paper.

**Questions:**

What do authors think about the challenges in handling weak recovery in a constant degree regime using correlated graphs? For weak recovery, it would be only good enough to create a partial overlap.

The rest of the questions are to clarify my understanding of Figure 2.
1. Why pink region is not a finding of this paper (it only mentions violet)? Would it be correct to say that one of the findings is to characterize the boundary between pink and violet region, i.e. how the union of violet and pink splits into the two regions?
2. Is the only difference between the pink and yellow regions that in the former, exact recovery would have been possible if any pair of matchings were known, but in the latter one needs all pairwise matchings? Otherwise, the two regions have the same properties: e.g. no pairwise matching is possible, not possible to recover the community using any individual pairs, yet possible using three graphs. Also, does the region of parameters given by pink+yellow exactly correspond to the region mentioned in abstract lines 12-15 (and as the highlights in the summary of the review).

**Limitations:**

Yes, the authors list out both negative and positive societal impacts of their work and graph matching in general.

---

> ### Author Rebuttal · Authors · 2024-08-07
>
> Thank you for your review and comments!
>
> Weak recovery in the constant average degree regime using correlated graphs is indeed a fascinating direction for future work (as already suggested in [18]). We suspect that the $k$-core analysis still gives something nontrivial in the large but constant average degree regime. However, proving this is far from immediate, due to a lack of concentration at this scale. Furthermore, determining the precise threshold for weak/partial recovery seems like a challenging problem which requires many more new ideas.
>
> Regarding Figure 2, indeed, the reviewer is completely right: characterizing both the pink and the violet regions, and the boundary between them, are all important findings of the paper. Previously, [18] characterized the union of these two regions. We will correct the caption of Figure 2, please see the attachment to the "global" response/rebuttal for an updated version. Thank you for catching this.
>
> Regarding the second set of questions about Figure 2, we hope that the "global" response/rebuttal -- which describes the threshold for exact graph matching from $K$ graphs (not just pairwise exact graph matching) -- helps clarify things. In particular, the previously yellow region now splits into an updated yellow region and a grey region. It is indeed the case that the union of the pink and the updated yellow regions is the region described in lines 12-15 of the abstract. We will add a sentence along these lines to the text in order to aid the reader.

---

> > ### Comment · Reviewer_c8Tz · 2024-08-09
> >
> > Thanks for providing clarifications! I would be happy to see this paper getting accepted.

---

### Official Review · Reviewer_RUAG · 2024-07-12

**Soundness:** 3
**Presentation:** 3
**Contribution:** 3
**Rating:** 7
**Confidence:** 3

**Summary:**

Theoretical work showing conditions for exact community recovery in $K$ correlated $2$-community SBMs where node labels are not maintained between networks. The work extends previous work for $K=2$ which introduces new challenges and proof mechanisms to allow for $K \ge 3$ networks. Theorems 1 and 2 provide necessary and sufficient conditions for exact community recovery relating to the difficulty of the pairwise matching problem $T_c(a, b)$ and the individual exact community recovery problem $D_+(a,b)$.

**Strengths:**

This research fully answers the open question of [18] introducing novel ideas top extend existing techniques to work in the much more complicated scenario with $K \ge 3$ networks. I feel the paper was well-structured to introduce and explain the problem at hand to someone outside of the research area. Section 2 did an excellent job introducing the problem highlighting the interplay between the community recovery and graph matching. Section 3 giving the main results along with a helpful high level description of the algorithm used within the proof and Figure 2 demonstrates these results pictorially highlighting the regions where their research come into play. Section 4 gave more details of the proof, while still a bit technically in places was still useful for myself who wanted to get the vibe of the proof without delving into the details in the Appendix.

The paper is very clear in its goal, what is has achieved and possible future directions in this area.

**Weaknesses:**

While reading this paper, I was unsure whether this paper fitted the remit of NeurIPS and would be better suited in another journal rather than conference paper. I felt more comfortable about this upon noticing that [18], which supplied the problem for this work, was itself inspired by work of community recovery in correlated SBMs [37] published in NeurIPS. I recognise this is a personal bias of what I consider a NeurIPS paper.

There is some discussion of how this work relates to real-world networks, but it is unclear how much further research needs to be done before these ideas can be used for practical analysis of real graphs. A number of my questions below relate to applying these ideas to more realistic networks.

**Questions:**

How does changing the correlated SBM to allow for different subsampling probabilities $s_i$ for each network affect the theory?

Is it possible to do just exact graph matching better than $s^2 T_c(a,b) > 1$ for $K \ge 3$ graphs? Perhaps this is addresses in some of the referenced papers, but my reading of Lines 128-129 is that exact graph matching is done pairwise for any graphs $G_k$ and $G_l$ which means that all graphs can be paired. Is there any improvement by doing things other than pairwise? I believe the answer reading on later is no, at least asymptotically, but it is uncertain later whether the benefit of extra graphs is solely for graph matching and community detection together, rather than individually. This may be addressed by resolving whether the "if if" typo in Line 128 should be just a single  "if" or, in fact, "if and only if".

While it is useful to consider the extreme case, often there is some evidence of persistent node labelling across networks. For example, Joe Bloggs on Facebook is more likely to correspond to the email joe_bloggs_01@mail.com compared to a completely random node. How could this scenario be incorporated into graph matching and community detection?

Do Steps 3-5 in the algorithm still work if $a < b$? Obviously, there is a symmetry in these two parameters, but as the algorithm is written, in this scenario neighbours are more likely to be the other community rather than the same. Please can you explain the changes necessary to make this approach work when $a < b$?

What subsampling parameter $s$ should we expect in real-world networks? Figure 2 shows two possible values but I have no idea what I would expect as normal. Given networks $G_1, \ldots, G_K$ one could find a maximum likelihood estimator for $s$, perhaps by computing the true $G$ using the graph matching algorithms described here and finding the subsample rate.

Any intuition why the value $k = 13$ in the $k$-core algorithm in Line 244?

How do these ideas relate to subsampling from any arbitrary base graph $G$ rather than just a 2-community SBM? For example, if $G$ was sampled from a generalised random dot product graph (A statistical interpretation of spectral embedding: the generalised random dot product graph, Rubin-Delanchy et al), then the subsampled graphs $G_i$ would also be a GRDPG. The problem of community recovery may not always make sense in that setting, but graph matching is still very important.

An alternative way of subsampling graphs is EdgeFlip used for edge-differential privacy in networks (Sharing social network data: differentially private estimation of exponential-family random graph models, Karwa et al). Edges and non-edges are flipped in $G$ with some probability $p$. How do these techniques work using this method of generating anomalised networks? The results here could show the dangers of producing multiple anomalous versions of the same network, a potentially important result for data privacy.

**Limitations:**

The authors highlight the positive and negative impact of their results, particular for de-anomalising multiple networks.

---

> ### Author Rebuttal · Authors · 2024-08-07
>
> Thank you for your review and comments!
>
> Q1 (different subsampling probabilities): This is a very natural question, and our developed theory can fully handle this. We simply chose to keep this part of the model simple, since there are already several parameters involved in the model.
>
> Let us illustrate on Theorem 1 how the results change with different subsampling probabilities $\{s_i\}$. First, condition (3.1) becomes $(1-\prod_{i=1}^{K}(1-s_i)) D_{+}(a,b)$. This change is immediate, since $(1-\prod_{i=1}^{K}(1-s_i))$ is the probability that an edge in the parent graph survives in at least one of the subsampled graphs, so this quantity arises in the correctly matched union graph (from which (3.1) is derived).
>
> More interesting is how (3.2) changes with different subsampling probabilities. An immediate adaptation of our proof gives the sufficient condition
>
> $\left[s_1\left(1-\prod_{i=2}^{K}(1-s_i)\right)\right]T_{c}(a,b)+\left[s_{1}\prod_{i=2}^{K}(1-s_i)\right]D_{+}(a,b)>1$.
>
> However, this condition is not necessary. The tight sufficient condition turns out to be:
>
> $\min_{j \in [K]} \left(\left[s_j\left(1-\prod_{i=1, i \neq j}^{K}(1-s_i) \right)\right]T_{c}(a,b)+\left[s_{j}\prod_{i=1, i \neq j}^{K}(1-s_i)\right]D_{+}(a,b)\right)>1$.
>
> The basic idea is as follows. Suppose that $j^*$ minimizes the expression above. Then, under this condition, we can first exactly recover the communities in $G_{j^*}$ (using the same algorithm as in our proof). Subsequently, we can port this partition to $G_1$ using the ideas in Appendices G and I.
>
> In the revision, we will add a remark to the paper along the lines above.
>
> Q2 (exact graph matching threshold): This is an excellent question and indeed it is possible to do exact graph matching better than just pairwise using $K \geq 3$ graphs. The exact graph matching threshold given $K$ correlated SBMs is given by
>
> $s\left(1-\left(1-s\right)^{K-1}\right)T_{c}(a,b)=1$.
>
> We expand upon this in detail in the "global" response/rebuttal, please see there for more.
>
> Q3 ("Joe Bloggs"): This is an astute observation and there are indeed various ways to incorporate this into a model, depending on the situation. One is to consider "seeded" graph matching, where a subset of vertices in the graphs are already matched, and the task is to match the remaining vertices. This setting is sometimes closer to practical applications, and is widely studied (e.g., [33], [45]). Another possibility is to consider "attributed" graph matching, where each node has a corresponding vector of attributes, and these are correlated across the graphs. See, e.g., Zhang, Wang, Wang, and Wang (IEEE Trans. on Information Theory, 2024) and Wang, Wang, and Wang (COLT 2024). Of course, these two modeling frameworks can be combined, and others may be relevant as well, depending on the application. All of these are worth more detailed study in our setting; however, this is beyond the scope of the current paper. We will add these to the possible future directions mentioned in the paper.
>
> Q4 (when $a<b$): Yes, everything works also when $a<b$, simply by replacing "majority" with "minority" everywhere. This is correctly stated in the Algorithms stated in Appendix C. In lines 173--192 of the main text, we gave a high level overview, which is perhaps more intuitive in the assortative setting when $a>b$. We forgot to mention here that this description is for $a>b$; we will correct this in the revision. Thank you for catching this omission.
>
> Q5 (subsampling parameter in real-world networks): We suspect that the answer may vary widely across applications. As an example, in a recent work by Li, Arroyo, Pantazis, Lyzinski (IEEE TNSE, 2023), the authors develop both theory and also apply it to human connectomes. While their paper does not indicate what the correlation parameter may be in this application, their theory and simulations are for constant correlation, and they vary the correlation parameter between 0 and 1. Likewise, our theory holds for any constant correlation $s\in [0,1]$; we hope that this can thus be useful in applications, regardless of what the actual correlation parameter is.
>
> Q6 (the value of $k$ in the $k$-core): The choice of $k=13$ is not particularly important -- any large enough constant will work. The particular choice of $k=13$ comes from [18], where it arises in their Lemma 4.8 and Lemma 4.10. In particular, Lemma 4.10 is a probability tail bound, and the choice of $k\geq 13$ (and in particular $k=13$) guarantees that this vanishes sufficiently quickly. The proof of Lemma 4.10 contains a union bound that may be loose, so it may be possible that the value of $k$ can be further lowered, but this doesn't have any major importance.
>
> Q7 (subsampling from an arbitrary base graph or a GRDPG): This is an excellent direction for future research. We suspect that the $k$-core analysis should also work for understanding (the information-theoretic limits of) graph matching of correlated GRDPGs, and the results of [39] support this heuristic as well. However, this is by no means obvious, and requires its own careful analysis.
>
> The setting of an arbitrary base graph is much more challenging. This essentially corresponds to a "smoothed analysis" of graph matching, which has not been done before. This would be very interesting to consider, but it is significantly beyond the scope of this paper.
>
> Q8 (alternative sampling using EdgeFlip): This setting is also interesting, and somewhat resembles the alternative setting we describe at the end of the paper (lines 365--372). We suspect that our methods are able to handle such a setting as well. In particular, for a constant number of graphs $K$, we conjecture that the results would be quantitatively different but qualitatively similar to the ones in the current paper. However, if $K$ is large (diverging with $n$), then we expect new qualitative phenomena to appear. This is an exciting avenue for future research, which we plan to pursue.

---

> > ### Comment · Reviewer_RUAG · 2024-08-11
> >
> > Thank you for your responses to my questions, several of which I accept are beyond the scope of this paper (e.q. Q7 and 8), but were of interest to me. I have increased my score of this paper as a result.

---

### Author Rebuttal · Authors · 2024-08-07

Thanks to all reviewers for the careful reviews and many helpful comments!

We respond to each review separately in its own rebuttal. In this "global" response, we take the opportunity to discuss something that was brought up by multiple reviewers: the threshold for exact graph matching given $K$ correlated SBMs.

In the submitted manuscript (on page 4), we mention that the information-theoretic threshold for exact graph matching for two correlated SBMs is $s^{2} T_{c}(a,b) = 1$ (by [37]). So, in particular, by a union bound, above this threshold one can do pairwise exact graph matching for any constant number $K$ correlated SBMs. However, this does not imply that this is the threshold for exact graph matching given $K$ correlated SBMs (it only gives a one-sided bound). And indeed: the threshold for exact graph matching given $K$ correlated SBMs is given by a different threshold, namely

$s \left( 1 - \left( 1 - s \right)^{K-1} \right) T_{c}(a,b) = 1$.

More formally, we have the following two theorems:

Theorem: Fix constants $a,b>0$ and $s\in [0,1]$, and let $(G_{1},G_{2},\ldots,G_{K})\sim \mathrm{CSBM}(n,\frac{a\log n}{n},\frac{b\log n}{n},s)$. Suppose that $s \left( 1 - \left( 1 - s \right)^{K-1} \right) T_{c}(a,b) > 1$. Then exact graph matching is possible. That is, there exists an estimator $\widehat{\pi}=\widehat{\pi}(G_1,\ldots,G_K)$ such that $\lim_{n\to\infty}\mathbb{P}(\widehat{\pi}=\pi^*)=1$.

Theorem: Fix constants $a,b>0$ and $s\in [0,1]$, and let $(G_{1},G_{2},\ldots,G_{K})\sim \mathrm{CSBM}(n,\frac{a\log n}{n},\frac{b\log n}{n},s)$. Suppose that $s \left( 1 - \left( 1 - s \right)^{K-1} \right) T_{c}(a,b) < 1$. Then exact graph matching is impossible. That is, for every estimator $\widehat{\pi}=\widehat{\pi}(G_1,\ldots,G_K)$ we have that $\lim_{n\to\infty}\mathbb{P}(\widehat{\pi}=\pi^*)=0$.

Note, in particular, that for $K \geq 3$ there exists a regime (specifically given by $s^{2} T_{c}(a,b) < 1 < s \left( 1 - \left( 1 - s \right)^{K-1} \right) T_{c}(a,b)$) where exact graph matching is possible from $K$ correlated SBMs even though pairwise exact graph matching is impossible.

We emphasize that the proofs of these theorems (which we sketch below) are already implicit in our submitted manuscript; in fact, our proofs for community exact recovery go significantly beyond these. In our original submitted manuscript we didn't mention these theorems, in part since our focus is on exact community recovery, and in part due to length constraints. Your feedback has made us realize that we should indeed include these in the paper. We plan on adding these in the revised version of the manuscript (with the statements in the main text, using the additional allowed page, and proofs in the appendix).

It is worth highlighting this threshold in the phase diagrams as well. We have attached here an updated version of Figure 2 with this incorporated. The change is that the previously yellow region (in the submitted version) now breaks into two regions: a yellow region and a grey region. In the grey region, exact community recovery is impossible from $(G_{1}, G_{2})$, pairwise exact graph matching is also impossible, but exact graph matching given $(G_{1}, G_{2}, G_{3})$ is possible, and subsequently exact community recovery is possible from $(G_1, G_2, G_3)$. In the (updated) yellow region, exact community recovery is impossible from $(G_{1}, G_{2})$, exact graph matching given $(G_{1}, G_{2}, G_{3})$ is also impossible, yet exact community recovery is possible given $(G_1,G_2,G_3)$.

We now sketch the proofs, starting with the impossibility result. Suppose that we give the algorithm even more information, namely all the matchings between the graphs $G_2, \ldots, G_K$. Then the correctly matched union graph $G_2 \vee \ldots \vee G_K$ can be computed (and, by a simulation argument (similar to the proof of Theorem 3.4 in [37]) it suffices to only consider this union graph). Note that $G_2 \vee \ldots \vee G_K$ is an SBM with parameters $(n, \left( 1 - \left( 1 - s \right)^{K-1} \right) a \log(n)/n, \left( 1 - \left( 1 - s \right)^{K-1} \right) b \log(n)/n)$. By a result of Cullina, Singhal, Kiyavash, and Mittal (2016), it is impossible to exactly recover the matching between an aforementioned SBM and an SBM with parameters $(n, s a \log(n)/n, s b \log(n)/n)$ under the condition $s \left( 1 - \left( 1 - s \right)^{K-1} \right) T_{c}(a,b) < 1$.

For the possibility result, the key is Appendix I. Under the condition $s \left( 1 - \left( 1 - s \right)^{K-1} \right) T_{c}(a,b) > 1$, one can see that the conclusion of Lemma I.2 is that the intersection set of interest is empty, which implies that all vertices are "good" vertices. The algorithm described in Appendix I then shows how to recover the latent matchings.

---

### Decision · Program_Chairs · 2024-09-25

**Decision:**

Accept (poster)

**Comment:**

The paper studies the problem of exact community recovery in $K$ correlated 2-community stochastic block models (SBMs) where node labels are not maintained between networks. The authors extend previous work for the $K=2$ case, introducing new challenges and proof mechanisms to allow for $K \geq 3$ networks. The main results of the paper are Theorems 1 and 2, which provide necessary and sufficient conditions for exact community recovery, relating to the difficulty of the pairwise matching problem and the individual exact community recovery problem.

All reviewers agree that the paper is well-written, with clear explanations of the problem and the interplay between community recovery and graph matching. The paper successfully answers the open question, generalizing the results to multiple ($K \geq 3$) SBMs. The reviewers appreciate the paper's well-structured presentation. The questions raised by the reviewers are well addressed.

Overall, this paper is technically sound, well-presented, and provides a solid foundation for future work in this area.